# CORRELATIVE INFORMATION MAXIMIZATION BASED BIOLOGICALLY PLAUSIBLE NEURAL NETWORKS FOR CORRELATED SOURCE SEPARATION

**Bariscan Bozkurt[1,2]**    **Ates Isfendiyaroglu[3]**    **Cengiz Pehlevan[4]**    **Alper T. Erdogan[1,2]**
[1]KUIS AI Center, Koc University, Turkey    [2]EEE Department, Koc University, Turkey
[3] Uskudar American Academy
[4]John A. Paulson School of Engineering & Applied Sciences and Center for Brain Science, Harvard University, Cambridge, MA 02138, USA
{bbozkurt15, alperdogan}@ku.edu.tr  cpehlevan@seas.harvard.edu

## ABSTRACT

The brain effortlessly extracts latent causes of stimuli, but how it does this at the network level remains unknown. Most prior attempts at this problem proposed neural networks that implement independent component analysis, which works under the limitation that latent causes are mutually independent. Here, we relax this limitation and propose a biologically plausible neural network that extracts correlated latent sources by exploiting information about their domains. To derive this network, we choose the maximum correlative information transfer from inputs to outputs as the separation objective under the constraint that the output vectors are restricted to the set where the source vectors are assumed to be located. The online formulation of this optimization problem naturally leads to neural networks with local learning rules. Our framework incorporates infinitely many set choices for the source domain and flexibly models complex latent structures. Choices of simplex or polytopic source domains result in networks with piecewise-linear activation functions. We provide numerical examples to demonstrate the superior correlated source separation capability for both synthetic and natural sources.

## 1 INTRODUCTION

Extraction of latent causes, or sources, of complex stimuli sensed by sensory organs is essential for survival. Due to absence of any supervision in most circumstances, this extraction must be performed in an unsupervised manner, a process which has been named blind source separation (BSS) (Comon & Jutten, 2010; Cichocki et al., 2009).

How BSS may be achieved in visual, auditory, or olfactory cortical circuits has attracted the attention of many researchers, e.g. (Bell & Sejnowski, 1995; Olshausen & Field, 1996; Bronkhorst, 2000; Lewicki, 2002; Asari et al., 2006; Narayan et al., 2007; Bee & Micheyl, 2008; McDermott, 2009; Mesgarani & Chang, 2012; Golumbic et al., 2013; Isomura et al., 2015). Influential papers showed that visual and auditory cortical receptive fields could arise from performing BSS on natural scenes (Bell & Sejnowski, 1995; Olshausen & Field, 1996) and sounds (Lewicki, 2002). The potential ubiquity of BSS in the brain suggests that there exists generic neural circuit motifs for BSS (Sharma et al., 2000). Motivated by these observations, here, we present a set of novel biologically plausible neural network algorithms for BSS.

BSS algorithms typically derive from normative principles. The most important one is the information maximization principle, which aims to maximize the information transferred from input mixtures to separator outputs under the restriction that the outputs satisfy a specific generative assumption about sources. However, Shannon mutual information is a challenging choice for quantifying information transfer, especially for data-driven adaptive applications, due to its reliance on the joint and conditional densities of the input and output components. This challenge is eased by the independent component analysis (ICA) framework by inducing joint densities into separable forms based on the assumption of source independence (Bell & Sejnowski, 1995). In particular scenarios,

the mutual independence of latent causes of real observations may not be a plausible assumption (Träuble et al., 2021). To address potential dependence among latent components, Erdogan (2022) recently proposed the use of the second-order statistics-based *correlative (log-determinant) mutual information* maximization for BSS to eliminate the need for the independence assumption, allowing for correlated source separation.

In this article, we propose an online correlative information maximization-based biologically plausible neural network framework (CorInfoMax) for the BSS problem. Our motivations for the proposed framework are as follows:

- The correlative mutual information objective function is only dependent on the second-order statistics of the inputs and outputs. Therefore, its use avoids the need for costly higher-order statistics or joint pdf estimates,
- The corresponding optimization is equivalent to maximization of correlation, or linear dependence, between input and output, a natural fit for the linear inverse problem,
- The framework relies only on the source domain information, eliminating the need for the source independence assumption. Therefore, neural networks constructed with this framework are capable of separating correlated sources. Furthermore, the CorInfoMax framework can be used to generate neural networks for infinitely many source domains corresponding to the combination of different attributes such as sparsity, nonnegativity etc.,
- The optimization of the proposed objective inherently leads to learning with local update rules.
- CorInfoMax acts as a unifying framework to generate biologically plausible neural networks for various unsupervised data decomposition methods to obtain structured latent representations, such as nonnegative matrix factorization (NMF) (Fu et al., 2019), sparse component analysis (SCA) (Babatas & Erdogan, 2018), bounded component analysis (BCA) (Erdogan, 2013; Inan & Erdogan, 2014) and polytopic matrix factorization (PMF) (Tatli & Erdogan, 2021).

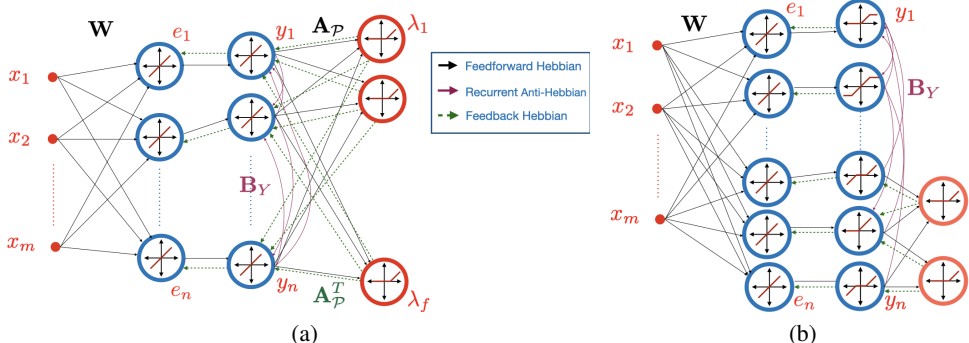

Figure 1: CorInfoMax BSS neural networks for two different canonical source domain representations. $x_i$'s and $y_i$'s represent inputs (mixtures) and (separator) outputs , respectively, $\mathbf{W}$ are feedforward weights, $e_i$'s are errors between transformed inputs and outputs, $\mathbf{B_y}$, the inverse of output autocorrelation matrix, represents lateral weights at the output. For the canonical form (a), $\lambda_i$'s are Lagrangian interneurons imposing source domain constraints, $\mathbf{A}_{\mathcal{P}}$ ($\mathbf{A}_{\mathcal{P}}^T$) represents feedforward (feedback) connections between outputs and interneurons. For the canonical form (b), interneurons on the right impose sparsity constraints on the subsets of outputs.

Figure 1 illustrates CorInfoMax neural networks for two different source domain representation choices, which are three-layer neural networks with piecewise linear activation functions.We note that the proposed CorInfoMax framework, beyond solving the BSS problem, can be used to learn structured and potentially correlated representations from data through the maximum correlative information transfer from inputs to the choice of the structured domain at the output.

## 1.1 RELATED WORK AND CONTRIBUTIONS

### 1.1.1 BIOLOGICALLY PLAUSIBLE NEURAL NETWORKS FOR BSS

There are different methods to solve the BSS problem through neural networks with local learning rules. These methods are differentiated on the basis of the observation models they assume and the normative approach they propose. We can list biologically plausible ICA networks as an example category, which are based on the exploitation of the presumed mutual independence of sources (Isomura & Toyoizumi, 2018; Bahroun et al., 2021; Lipshutz et al., 2022). There exist alternative approaches which exploit different properties of the data model to replace the independence assumption with a weaker one. As an example, Pehlevan et al. (2017a) uses the nonnegativeness property along with the biologically inspired similarity matching (SM) framework (Pehlevan et al., 2017b) to derive biologically plausible neural networks that are capable of separating uncorrelated but potentially dependent sources. Similarly, Erdogan & Pehlevan (2020) proposes bounded similarity matching (BSM) as an alternative approach that takes advantage of the magnitude bound-edness property for uncorrelated source separation. More recently, Bozkurt et al. (2022) introduced a generic biologically plausible neural network framework based on weighted similarity matching (WSM) introduced in Erdogan & Pehlevan (2020) and maximization of the output correlation determinant criterion used in the NMF, SCA, BCA and PMF methods. This new framework exploits the domain structure of sources to generate two- / three-layer biologically plausible networks that have the ability to separate potentially correlated sources. Another example of biologically plausible neural networks with correlated source separation capability is offered in Simsek & Erdogan (2019), which also uses the determinant maximization criterion for the separation of antisparse sources.

Our proposed framework differs significantly from Bozkurt et al. (2022): Bozkurt et al. (2022) uses the similarity matching criterion, which is not employed in our framework, as the main tool for generating biologically plausible networks. Therefore, the resulting network structure and learning rules are completely different from Bozkurt et al. (2022). For example, the lateral connections of the outputs in Bozkurt et al. (2022) are based on the output autocorrelation matrix, while the lateral connections for our proposed framework are based on the inverse of the output autocorrelation matrix. Unlike our framework, neurons in Bozkurt et al. (2022) have learnable gains. The feedforward weights of the networks in Bozkurt et al. (2022) correspond to a cross-correlation matrix between the inputs and outputs of a layer, whereas, for our proposed framework, the feedforward connections correspond to the linear predictor of the output from the input. The feedback connection matrix in Bozkurt et al. (2022) is the transpose of the feedforward matrix which is not the case in our proposed framework. Compared to the approach in Simsek & Erdogan (2019), our proposed framework is derived from information-theoretic grounds, and its scope is not limited to antisparse sources, but to infinitely many different source domains.

### 1.1.2 INFORMATION MAXIMIZATION FOR UNSUPERVISED LEARNING

The use of Shannon's mutual information maximization for various unsupervised learning tasks dates back to a couple of decades. As one of the pioneering applications, we can list Linsker's work on self-organizing networks, which proposes maximizing mutual information between input and its latent representation as a normative approach (Linsker, 1988). Under the Gaussian assumption, the corresponding objective simplifies to determinant maximization for the output covariance matrix. Becker & Hinton (1992) suggested maximizing mutual information between alternative latent vectors derived from the same input source as a self-supervised method for learning representations. The most well-known application of the information maximization criterion to the BSS problem is the ICA-Infomax approach by Bell & Sejnowski (1995). The corresponding algorithm maximizes the information transferred from the input to the output under the constraint that the output components are mutually independent. For potentially correlated sources, Erdogan (2022) proposed the use of a second-order statistics-based correlative (or log-determinant) mutual information measure for the BSS problem. This approach replaces the mutual independence assumption in the ICA framework with the source domain information, enabling the separation of both independent and dependent sources. Furthermore, it provides an information-theoretic interpretation for the determinant maximization criterion used in several unsupervised structured matrix factorization frameworks such as NMF (or simplex structured matrix factorization (SSMF)) (Chan et al., 2011; Lin et al., 2015; Fu et al., 2018; 2019), SCA, BCA, and PMF. More recently, Ozsoy et al. (2022)

proposed maximization of the correlative information among latent representations corresponding to different augmentations of the same input as a self-supervised learning method.

The current article offers an online optimization formulation for the batch correlative information maximization method of Erdogan (2022) that leads to a general biologically plausible neural network generation framework for the unsupervised unmixing of potentially dependent/correlated sources.

## 2 PRELIMINARIES

This section aims to provide background information for the CorInfoMax-based neural network framework introduced in Section 3. For this purpose, we first describe the BSS setting assumed throughout the article in Section 2.1. Then, in Section 2.2, we provide an essential summary of the batch CorInfoMax-based BSS approach introduced in (Erdogan, 2022).

### 2.1 BLIND SOURCE SEPARATION SETTING

_SOURCES:_ We assume a BSS setting with a finite number of $n$-dimensional source vectors, represented by the set $\mathbb{S} = \{s(1), s(2), \ldots, s(N)\} \subset \mathcal{P}$, where $\mathcal{P}$ is a particular subset of $\mathbb{R}^n$. The choice of source domain $\mathcal{P}$ determines the identifiability of the sources from their mixtures, the properties of the individual sources and their mutual relations. Structured unsupervised matrix factorization methods are usually defined by the source/latent domain, such as

i. _Normalized nonnegative sources in the NMF(SSMF) framework_: $\Delta = \{s \mid s \geq 0, \mathbf{1}^T s = 1\}$. Signal processing and machine learning applications such as hyperspectral unmixing and text mining (Abdolali & Gillis, 2021), (Fu et al., 2016).

ii. _Bounded antisparse sources in the BCA framework_: $\mathcal{B}_{\ell_\infty} = \{s \mid \|s\|_\infty \leq 1\}$. Applications include digital communication signals Erdogan (2013).

iii. _Bounded sparse sources in the SCA framework_: $\mathcal{B}_{\ell_1} = \{s \mid \|s\|_1 \leq 1\}$. Applications: modeling efficient representations of stimulus such as vision Olshausen & Field (1997) and sound Smith & Lewicki (2006).

iv _Nonnegative bounded antiparse sources in the nonnegative-BCA framework_: $\mathcal{B}_{\ell_\infty,+} = \mathcal{B}_{\ell_\infty} \cap \mathbb{R}^n_+$. Applications include natural images Erdogan (2013).

v. _Nonnegative bounded sparse sources in the nonnegative-SCA framework_: $\mathcal{B}_{\ell_1,+} = \mathcal{B}_{\ell_1} \cap \mathbb{R}^n_+$. Potential applications similar to $\Delta$ in (i).

Note that the sets in (ii)-(v) of the above list are the special cases of (convex) polytopes. Recently, Tatli & Erdogan (2021) showed that infinitely many polytopes with a certain symmetry restriction enable identifiability for the BSS problem. Each _identifiable_ polytope choice corresponds to different structural assumptions on the source components. A common canonical form to describe polytopes is to use the H-representation Grünbaum et al. (1967):

$$\mathcal{P} = \{y \in \mathbb{R}^n | A_{\mathcal{P}} y \preccurlyeq b_{\mathcal{P}}\}, \tag{1}$$

which corresponds to the intersection of half-spaces. Alternatively, similar to Bozkurt et al. (2022), we can consider a subset of polytopes, which we refer to as _feature-based_ polytopes, defined in terms of attributes (such as non-negativity and sparseness) assigned to the subsets of components:

$$\mathcal{P} = \left\{ s \in \mathbb{R}^n \mid s_i \in [-1, 1] \forall i \in \mathcal{I}_s, \, s_i \in [0, 1] \forall i \in \mathcal{I}_+, \, \|s_{\mathcal{J}_l}\|_1 \leq 1, \, \mathcal{J}_l \subseteq \mathbb{Z}_n^+, \, l \in \mathbb{Z}_L^+ \right\}, \tag{2}$$

where $\mathcal{I}_s \subseteq \mathbb{Z}_n^+$ is the set of indexes for signed sources, and $\mathcal{I}_+$ is its complement, $s_{\mathcal{J}_l}$ is the sub-vector constructed from the elements with indices in $\mathcal{J}_l$, and $L$ is the number of sparsity constraints imposed on the sub-vector level. In this article, we consider both polytope representations above.

_MIXING:_ We assume a linear generative model, that is, the source vectors are mixed through an unknown matrix $A \in \mathbb{R}^{m \times n}$, $x(i) = As(i)$, $\forall i = 1, \ldots, N$, where we consider the overdetermined case, that is, $m \geq n$ and $rank(A) = n$. We define $X = [\ x(1) \ \ldots \ x(N) \ ]$.

_SEPARATION:_ The purpose of the source separation setting is to recover the original source matrix $S$ from the mixture matrix $X$ up to some scaling and/or permutation ambiguities, that is, the separator output vectors $\{y(i)\}$ satisfy $y(i) = Wx(i) = \Pi\Lambda s(i)$, for all $i = 1, \ldots, N$, where $W \in \mathbb{R}^{n \times m}$ is the learned separator matrix, $\Pi$ is a permutation matrix, $\Lambda$ is a full-rank diagonal matrix and $y(i)$ refers to the estimate of the source of the sample index $i$.

## 2.2 Correlative Mutual Information Maximization for BSS

Erdogan (2022) proposes maximizing the (correlative) information flow from the mixtures to the separator outputs, while the outputs are restricted to lie in their presumed domain $\mathcal{P}$. The corresponding batch optimization problem is given by

$$\underset{\boldsymbol{Y} \in \mathbb{R}^{n \times N}}{\text{maximize}} \quad I_{\text{LD}}^{(\epsilon)}(\boldsymbol{X}, \boldsymbol{Y}) = \frac{1}{2} \log \det(\hat{\boldsymbol{R}}_{\boldsymbol{y}} + \epsilon \boldsymbol{I}) - \frac{1}{2} \log \det(\hat{\boldsymbol{R}}_{\boldsymbol{e}} + \epsilon \boldsymbol{I}) \tag{3a}$$

$$\text{subject to} \quad \boldsymbol{Y}_{:,i} \in \mathcal{P}, i = 1, \ldots, N, \tag{3b}$$

where the objective function $I_{\text{LD}}^{(\epsilon)}(\boldsymbol{X}, \boldsymbol{Y})$ is the log-determinant (LD) mutual information[1] between the mixture and the separator output vectors (see Appendix A.1 and Erdogan (2022) for more information), $\hat{\boldsymbol{R}}_{\boldsymbol{y}}$ is the sample autocorrelation, i.e., $\hat{\boldsymbol{R}}_{\boldsymbol{y}} = \frac{1}{N} \boldsymbol{Y} \boldsymbol{Y}^T$, (or autocovariance, i.e., $\hat{\boldsymbol{R}}_{\boldsymbol{y}} = \frac{1}{N} \boldsymbol{Y}(\boldsymbol{I}_N - \frac{1}{N} \mathbf{1}_N \mathbf{1}_N^T) \boldsymbol{Y}^T$) matrix for the separator output vector, $\hat{\boldsymbol{R}}_{\boldsymbol{e}}$ equals $\hat{\boldsymbol{R}}_{\boldsymbol{y}} - \hat{\boldsymbol{R}}_{\boldsymbol{xy}}^T (\hat{\boldsymbol{R}}_{\boldsymbol{x}} + \epsilon \boldsymbol{I})^{-1} \hat{\boldsymbol{R}}_{\boldsymbol{yx}}$ where $\hat{\boldsymbol{R}}_{\boldsymbol{xy}}$ is the sample cross-correlation, i.e., $\hat{\boldsymbol{R}}_{\boldsymbol{xy}} = \frac{1}{N} \boldsymbol{X} \boldsymbol{Y}^T$ (or cross-covariance $\hat{\boldsymbol{R}}_{\boldsymbol{xy}} = \frac{1}{N} \boldsymbol{X}(\boldsymbol{I}_N - \frac{1}{N} \mathbf{1}_N \mathbf{1}_N^T) \boldsymbol{Y}^T$) matrix between mixture and output vectors, and $\hat{\boldsymbol{R}}_{\boldsymbol{x}}$ is the sample autocorrelation (or autocovariance) matrix for the mixtures. As discussed in Appendix A.1, for sufficiently small $\epsilon$, $\hat{\boldsymbol{R}}_{\boldsymbol{e}}$ is the sample autocorrelation (covariance) matrix of the error vector corresponding to the best linear (affine) minimum mean square error (MMSE) estimate of the separator output vector $\boldsymbol{y}$, from the mixture vector $\boldsymbol{x}$. Under the assumption that the original source samples are sufficiently scattered in $\mathcal{P}$, (Fu et al., 2019; Tatli & Erdogan, 2021), i.e., they form a maximal LD-entropy subset of $\mathcal{P}$, (Erdogan, 2022), then the optimal solution of (3) recovers the original sources up to some permutation and sign ambiguities, for sufficiently small $\epsilon$.

The biologically plausible CorInfoMax BSS neural network framework proposed in this article is obtained by replacing batch optimization in (3) with its online counterpart, as described in Section 3.

## 3 Method: Biologically Plausible Neural Networks for Correlative Information Maximization

### 3.1 Online Optimization Setting for LD-Mutual Information Maximization

We start our online optimization formulation for CorInfoMax by replacing the output and error sample autocorrelation matrices in (3a) with their weighted versions

$$\hat{\boldsymbol{R}}_{\boldsymbol{y}}^{\zeta_{\boldsymbol{y}}}(k) = \frac{1 - \zeta_{\boldsymbol{y}}}{1 - \zeta_{\boldsymbol{y}}^k} \sum_{i=1}^{k} \zeta_{\boldsymbol{y}}^{k-i} \boldsymbol{y}(i) \boldsymbol{y}(i)^T \quad \hat{\boldsymbol{R}}_{\boldsymbol{e}}^{\zeta_{\boldsymbol{e}}}(k) = \frac{1 - \zeta_{\boldsymbol{e}}}{1 - \zeta_{\boldsymbol{e}}^k} \sum_{i=1}^{k} \zeta_{\boldsymbol{e}}^{k-i} \boldsymbol{e}(i) \boldsymbol{e}(i)^T, \tag{4}$$

where $0 \ll \zeta_{\boldsymbol{y}} < 1$ is the forgetting factor, $\boldsymbol{W}(i)$ is the best linear MMSE estimator matrix (to estimate $\boldsymbol{y}$ from $\boldsymbol{x}$), and $\boldsymbol{e}(k) = \boldsymbol{y}(i) - \boldsymbol{W}(i) \boldsymbol{x}(i)$ is the corresponding error vector. Therefore, we can define the corresponding online CorInfoMax optimization problem as

$$\underset{\boldsymbol{y}(k) \in \mathbb{R}^n}{\text{maximize}} \quad \mathcal{J}(\boldsymbol{y}(k)) = \frac{1}{2} \log \det(\hat{\boldsymbol{R}}_{\boldsymbol{y}}^{\zeta_{\boldsymbol{y}}}(k) + \epsilon \boldsymbol{I}) - \frac{1}{2} \log \det(\hat{\boldsymbol{R}}_{\boldsymbol{e}}^{\zeta_{\boldsymbol{e}}}(k) + \epsilon \boldsymbol{I}) \tag{5a}$$

$$\text{subject to} \quad \boldsymbol{y}_k \in \mathcal{P}. \tag{5b}$$

Note that the above formulation assumes knowledge of the best linear MMSE matrix $\boldsymbol{W}(i)$, whose update is formulated as a solution to an online regularized least squares problem,

$$\underset{\boldsymbol{W}(i) \in \mathbb{R}^{m \times n}}{\text{maximize}} \quad \mu_{\boldsymbol{W}} \|\boldsymbol{y}(i) - \boldsymbol{W}(i) \boldsymbol{x}(i)\|_2^2 + \|\boldsymbol{W}(i) - \boldsymbol{W}(i-1)\|_F^2. \tag{6a}$$

### 3.2 Description of the Network Dynamics for Sparse Sources

We now show that the gradient-ascent-based maximization of the online CorInfoMax objective in (5) corresponds to the neural dynamics of a multilayer recurrent neural network with local learning

---

[1] In this article, we use "correlative mutual information" and "LD-mutual information" interchangeably.

rules. Furthermore, the presumed source domain $\mathcal{P}$ determines the output activation functions and additional inhibitory neurons. For an illustrative example, in this section, we concentrate on the sparse special case in Section 2.1, that is, $\mathcal{P} = \mathcal{B}_{\ell_1}$. We can write the corresponding Lagrangian optimization setting as

$$\underset{\lambda \geq 0}{\text{minimize}} \; \underset{\boldsymbol{y}(k) \in \mathbb{R}^n}{\text{maximize}} \qquad \mathcal{L}(\boldsymbol{y}(k), \lambda(k)) = \mathcal{J}(\boldsymbol{y}(k)) - \lambda(k)(\|\boldsymbol{y}(k)\|_1 - 1). \qquad (7)$$

To derive network dynamics, we use the proximal gradient update (Parikh et al., 2014) for $\boldsymbol{y}(k)$ with the expression (A.14) for $\nabla_{\boldsymbol{y}(k)} \mathcal{J}(\boldsymbol{y}(k))$, derived in Appendix B, and the projected gradient descent update for $\lambda(k)$ using $\nabla_\lambda \mathcal{L}(\boldsymbol{y}(k), \lambda(k)) = 1 - \|\boldsymbol{y}(k; \nu+1)\|_1$, leading to the following iterations:

$$\boldsymbol{e}(k; \nu) = \boldsymbol{y}(k; \nu) - \boldsymbol{W}(k)\boldsymbol{x}(k) \qquad (8)$$

$$\nabla_{\boldsymbol{y}(k)} \mathcal{J}(\boldsymbol{y}(k; \nu)) = \gamma_{\boldsymbol{y}}(k) \boldsymbol{B}_{\boldsymbol{y}}^{\zeta_{\boldsymbol{y}}}(k-1) \boldsymbol{y}(k; \nu) - \gamma_{\boldsymbol{e}}(k) \boldsymbol{B}_{\boldsymbol{e}}^{\zeta_{\boldsymbol{e}}}(k-1) \boldsymbol{e}(k; \nu), \qquad (9)$$

$$\boldsymbol{y}(k; \nu+1) = ST_{\lambda(k; \nu)} \big( \boldsymbol{y}(k; \nu) + \eta_{\boldsymbol{y}}(\nu) \nabla_{\boldsymbol{y}(k)} \mathcal{J}(\boldsymbol{y}(k; \nu)) \big) \qquad (10)$$

$$\lambda(k; \nu+1) = \text{ReLU}\Big( \lambda(k; \nu) - \eta_\lambda(\nu)(1 - \|\boldsymbol{y}(k; \nu+1)\|_1) \Big), \qquad (11)$$

where $\boldsymbol{B}_{\boldsymbol{y}}^{\zeta_{\boldsymbol{y}}}(k)$ and $\boldsymbol{B}_{\boldsymbol{e}}^{\zeta_{\boldsymbol{e}}}(k)$ are inverses of $\boldsymbol{R}_{\boldsymbol{y}}^{\zeta_{\boldsymbol{y}}}(k-1)$ and $\boldsymbol{R}_{\boldsymbol{e}}^{\zeta_{\boldsymbol{e}}}(k-1)$ respectively, $\gamma_{\boldsymbol{y}}(k)$ and $\gamma_{\boldsymbol{e}}(k)$ are provided in (A.11) and (A.13), $\nu \in \mathbb{N}$ is the iteration index, $\eta_{\boldsymbol{y}}(\nu)$ and $\eta_\lambda(\nu)$ are the learning rates for the output and $\lambda$, respectively, at iteration $\nu$, $\text{ReLU}(\cdot)$ is the rectified linear unit, and $ST_\lambda(.)$ is the soft-thresholding nonlinearity defined as $ST_\lambda(\boldsymbol{y})_i = \begin{cases} 0 & |y_i| \leq \lambda, \\ y_i - sign(y_i)\lambda & \text{otherwise} \end{cases}$ . We represent the values in the final iteration $\nu_{final}$, with some abuse of notation, with $\boldsymbol{y}(k) = \boldsymbol{y}(k; \nu_{final})$ and $\boldsymbol{e}(k) = \boldsymbol{e}(k; \nu_{final})$.

The neural dynamic iterations in (8)-(11) define a recurrent neural network, where $\boldsymbol{W}(k)$ represents feedforward synaptic connections from the input $\boldsymbol{x}(k)$ to the error $\boldsymbol{e}(k)$, $\boldsymbol{B}_{\boldsymbol{e}}^{\zeta_{\boldsymbol{e}}}(k)$ represents feedforward connections from the error $\boldsymbol{e}(k)$ to the output $\boldsymbol{y}(k)$, and $\boldsymbol{B}_{\boldsymbol{y}}^{\zeta_{\boldsymbol{y}}}(k)$ corresponds to lateral synaptic connections among the output components. Next, we examine the learning rules for this network.

Update of inverse correlation matrices $\boldsymbol{B}_{\boldsymbol{y}}^{\zeta_{\boldsymbol{y}}}(k)$ and $\boldsymbol{B}_{\boldsymbol{e}}^{\zeta_{\boldsymbol{e}}}(k)$: We can obtain the update expressions by applying matrix inversion lemma to $(\hat{\boldsymbol{R}}_{\boldsymbol{y}}^{\zeta_{\boldsymbol{y}}}(k) + \epsilon \boldsymbol{I})^{-1}$ and $(\hat{\boldsymbol{R}}_{\boldsymbol{e}}^{\zeta_{\boldsymbol{e}}}(k) + \epsilon \boldsymbol{I})^{-1}$, as derived in Appendix B to obtain (A.10) and (A.12)

$$\boldsymbol{B}_{\boldsymbol{y}}^{\zeta_{\boldsymbol{y}}}(k+1) = \frac{1 - \zeta_{\boldsymbol{y}}^k}{\zeta_{\boldsymbol{y}} - \zeta_{\boldsymbol{y}}^k}(\boldsymbol{B}_{\boldsymbol{y}}^{\zeta_{\boldsymbol{y}}}(k) - \gamma_{\boldsymbol{y}}(k)\boldsymbol{B}_{\boldsymbol{y}}^{\zeta_{\boldsymbol{y}}}(k)\boldsymbol{y}(k)\boldsymbol{y}(k)^T \boldsymbol{B}_{\boldsymbol{y}}^{\zeta_{\boldsymbol{y}}}(k)), \qquad (12)$$

$$\boldsymbol{B}_{\boldsymbol{e}}^{\zeta_{\boldsymbol{e}}}(k+1) = \frac{1 - \zeta_{\boldsymbol{e}}^k}{\zeta_{\boldsymbol{e}} - \zeta_{\boldsymbol{e}}^k}(\boldsymbol{B}_{\boldsymbol{e}}^{\zeta_{\boldsymbol{e}}}(k) - \gamma_{\boldsymbol{e}}(k)\boldsymbol{B}_{\boldsymbol{e}}^{\zeta_{\boldsymbol{e}}}(k)\boldsymbol{e}(k)\boldsymbol{e}(k)^T \boldsymbol{B}_{\boldsymbol{e}}^{\zeta_{\boldsymbol{e}}}(k)). \qquad (13)$$

However, note that (12), and (13) violate biological plausibility, since the multiplier $\gamma_{\boldsymbol{y}}$ and $\gamma_{\boldsymbol{e}}$ depend on all output and error components contrasting the locality. Furthermore, the update in (13) is not local since it is only a function of the feedforward signal $\boldsymbol{z}_{\boldsymbol{e}}(k) = \boldsymbol{B}_{\boldsymbol{e}}^{\zeta_{\boldsymbol{e}}}(k)\boldsymbol{e}(k)$ entering into output neurons: the update of $[\boldsymbol{B}_{\boldsymbol{e}}^{\zeta_{\boldsymbol{e}}}]_{ij}$, the synaptic connection between the output neuron $i$ and the error neuron $j$ requires $[\boldsymbol{z}_{\boldsymbol{e}}]_j$ which is a signal input to the output neuron $j$. To modify the updates (12) and (13) into a biologically plausible form, we make the following observations and assumptions:

- $\hat{\boldsymbol{R}}_{\boldsymbol{e}}^{\zeta_{\boldsymbol{e}}}(k) + \epsilon \boldsymbol{I} \approx \epsilon \boldsymbol{I} \Rightarrow \boldsymbol{B}_{\boldsymbol{e}}^{\zeta_{\boldsymbol{e}}}(k+1) \approx \frac{1}{\epsilon}\boldsymbol{I}$, which is a reasonable assumption, as we expect the error $\boldsymbol{e}(k)$ to converge near zero in the noiseless linear observation model,

- If $\zeta_{\boldsymbol{y}}$ is close enough to 1, and the time step $k$ is large enough, $\gamma_{\boldsymbol{y}}(k)$ is approximately $\frac{1 - \zeta_{\boldsymbol{y}}}{\zeta_{\boldsymbol{y}}}$.

Therefore, we modify the update equation of $\boldsymbol{B}_{\boldsymbol{y}}^{\zeta_{\boldsymbol{y}}}(k+1)$ in (12) as

$$\boldsymbol{B}_{\boldsymbol{y}}^{\zeta_{\boldsymbol{y}}}(k+1) = \frac{1}{\zeta_{\boldsymbol{y}}}(\boldsymbol{B}_{\boldsymbol{y}}^{\zeta_{\boldsymbol{y}}}(k) - \frac{1 - \zeta_{\boldsymbol{y}}}{\zeta_{\boldsymbol{y}}}\boldsymbol{B}_{\boldsymbol{y}}^{\zeta_{\boldsymbol{y}}}(k)\boldsymbol{y}(k)\boldsymbol{y}(k)^T \boldsymbol{B}_{\boldsymbol{y}}^{\zeta_{\boldsymbol{y}}}(k)). \qquad (14)$$

Feed-forward synaptic connections $\boldsymbol{W}(k)$: The solution of online optimization in (6) is given by

$$\boldsymbol{W}(k+1) = \boldsymbol{W}(k) + \mu_{\boldsymbol{W}}(k)\boldsymbol{e}(k)\boldsymbol{x}(k)^T, \qquad (15)$$

where $\mu_{\boldsymbol{W}}(k)$ is the step size corresponding to the adaptive least-mean-squares (LMS) update based on the MMSE criterion (Sayed, 2003). **Algorithm 1** below summarizes the Sparse CorInfoMax output and learning dynamics:

---

**Algorithm 1** Sparse CorInfoMax Algorithm

---

**Input**: Streaming data $\{\boldsymbol{x}(k) \in \mathbb{R}^m\}_{k=1}^N$, **Output**: $\{\boldsymbol{y}(k) \in \mathbb{R}^n\}_{k=1}^N$.

1: Initialize $\zeta_{\boldsymbol{y}}, \zeta_{\boldsymbol{e}}, \mu(1)_{\boldsymbol{W}}, \boldsymbol{W}(1), \boldsymbol{B}_{\boldsymbol{y}}^{\zeta_{\boldsymbol{y}}}(1), \boldsymbol{B}_{\boldsymbol{e}}^{\zeta_{\boldsymbol{e}}}(1)$.
2: **for** k = 1, 2, …, N **do**
3:     **run neural output dynamics until convergence:**

$$\boldsymbol{e}(k; \nu) = \boldsymbol{y}(k; \nu) - \boldsymbol{W}(k)\boldsymbol{x}(k)$$

$$\nabla_{\boldsymbol{y}(k)} \mathcal{J}(\boldsymbol{y}(k; \nu)) = \gamma_{\boldsymbol{y}}(k) \boldsymbol{B}_{\boldsymbol{y}}^{\zeta_{\boldsymbol{y}}}(k-1)\boldsymbol{y}(k; \nu) - \gamma_{\boldsymbol{e}}(k) \boldsymbol{B}_{\boldsymbol{e}}^{\zeta_{\boldsymbol{e}}}(k-1)\boldsymbol{e}(k; \nu),$$

$$\boldsymbol{y}(k; \nu + 1) = ST_{\lambda(k; \nu)}\left(\boldsymbol{y}(k; \nu) + \eta_{\boldsymbol{y}}(\nu)\nabla_{\boldsymbol{y}(k)} \mathcal{J}(\boldsymbol{y}(k; \nu))\right)$$

$$\lambda(k; \nu + 1) = \text{ReLU}(\lambda(k; \nu) - \eta_\lambda(\nu)(1 - \|\boldsymbol{y}(k; \nu + 1)\|_1)),$$

4:     Update feedforward synapses: $\boldsymbol{W}(k+1) = \boldsymbol{W}(k) + \mu_{\boldsymbol{W}}(k)\boldsymbol{e}(k)\boldsymbol{x}(k)^T$
5:     Update lateral synapses: $\boldsymbol{B}_{\boldsymbol{y}}^{\zeta_{\boldsymbol{y}}}(k+1) = \frac{1}{\zeta_{\boldsymbol{y}}}(\boldsymbol{B}_{\boldsymbol{y}}^{\zeta_{\boldsymbol{y}}}(k) - \frac{1-\zeta_{\boldsymbol{y}}}{\zeta_{\boldsymbol{y}}} \boldsymbol{B}_{\boldsymbol{y}}^{\zeta_{\boldsymbol{y}}}(k)\boldsymbol{y}(k)\boldsymbol{y}(k)^T \boldsymbol{B}_{\boldsymbol{y}}^{\zeta_{\boldsymbol{y}}}(k))$
6: **end for**

---

Neural Network Realizations: Figure 2a shows the three-layer realization of the sparse CorInfoMax neural network based on the network dynamics expressions in (8)- (11) and the approximation $\boldsymbol{B}_{\boldsymbol{e}}^{\zeta_{\boldsymbol{e}}}(k) \approx \frac{1}{\epsilon}\boldsymbol{I}$. The first layer corresponds to the error ($\boldsymbol{e}(k)$) neurons, and the second layer corresponds to the output ($\boldsymbol{y}(k)$) neurons with soft thresholding activation functions. If we substitute (8) and $\boldsymbol{B}_{\boldsymbol{e}}^{\zeta_{\boldsymbol{e}}}(k+1) = \frac{1}{\epsilon}\boldsymbol{I}$ in (9), we obtain $\nabla_{\boldsymbol{y}(k)} J(\boldsymbol{y}(k; \nu)) = \boldsymbol{M}_{\boldsymbol{y}}^{\zeta_{\boldsymbol{y}}}(k)\boldsymbol{y}(k; \nu) + \frac{\gamma_{\boldsymbol{e}}}{\epsilon}\boldsymbol{W}(k)\boldsymbol{x}(k)$ where $\boldsymbol{M}_{\boldsymbol{y}}^{\zeta_{\boldsymbol{y}}}(k) = \gamma_{\boldsymbol{y}} \boldsymbol{B}_{\boldsymbol{y}}^{\zeta_{\boldsymbol{y}}}(k) - \frac{\gamma_{\boldsymbol{e}}}{\epsilon}\boldsymbol{I}$. Therefore, this gradient expression and (10)- (11) correspond to the two-layer network shown in Figure 2b.

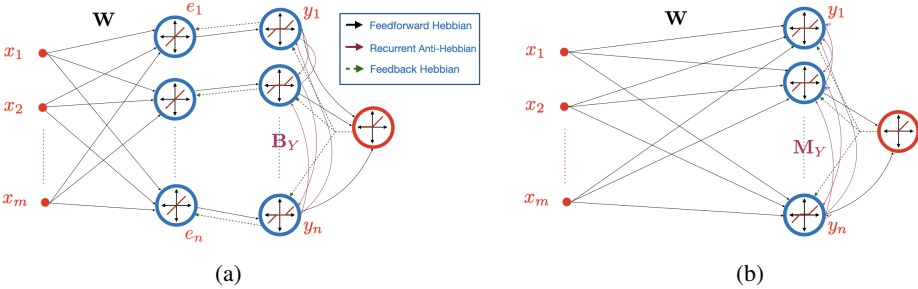

(a)                                      (b)

Figure 2: Sparse CorInfoMax Network: (a) three-layer (b) two-layer. $x_i$'s and $y_i$'s represent inputs (mixtures) and (separator) outputs , respectively, **W** are feedforward weights, $e_i$'s in the three-layer implementation (on the left) are errors between transformed inputs and outputs, $\mathbf{B_Y}$ in (a), the inverse of output autocorrelation matrix, represents lateral weights at the output. $\mathbf{M_Y}$ in (b), represents the output lateral weights, which is a diagonally modified form of the inverse of output correlation. In both representations, the rightmost interneuron impose the sparsity constraint.

The neural network derivation examples for other source domains are provided in Appendix C.

### 3.3   DESCRIPTION OF THE NETWORK DYNAMICS FOR A CANONICAL POLYTOPE REPRESENTATION

In this section, we consider the optimization problem specified in (5) for a generic polytope with H-representation in (1). We can write the corresponding online optimization setting in Lagrangian form as

$$\underset{\substack{\boldsymbol{\lambda}(k) \succcurlyeq 0 \quad \boldsymbol{y}(k) \in \mathbb{R}^n}}{\text{minimize maximize}} \quad \mathcal{L}(\boldsymbol{y}(k), \boldsymbol{\lambda}(k)) = \mathcal{J}(\boldsymbol{y}(k)) - \boldsymbol{\lambda}(k)^T(\boldsymbol{A}_{\mathcal{P}}\boldsymbol{y}(k) - \boldsymbol{b}_{\mathcal{P}}), \quad (16)$$

which is a Min-Max problem. For the recursive update dynamics of the network output and the Lagrangian variable, we obtain the derivative of the objective in (16) with respect to $\boldsymbol{y}(k)$ and $\boldsymbol{\lambda}(k)$ as

$$\nabla_{\boldsymbol{y}(k)}\mathcal{L}(\boldsymbol{y}(k;\nu),\boldsymbol{\lambda}(k;\nu)) = \gamma_{\boldsymbol{y}}\boldsymbol{B}_{\boldsymbol{y}}^{\zeta_{\boldsymbol{y}}}(k-1)\boldsymbol{y}(k;\nu) - \gamma_{\boldsymbol{e}}\boldsymbol{B}_{\boldsymbol{e}}^{\zeta_{e}}(k-1)\boldsymbol{e}(k;\nu) - \boldsymbol{A}_{\mathcal{P}}^{T}\boldsymbol{\lambda}(k;\nu), \quad (17)$$

$$\nabla_{\boldsymbol{\lambda}(k)}\mathcal{L}(\boldsymbol{y}(k;\nu)) = -\boldsymbol{A}_{\mathcal{P}}\boldsymbol{y}(k;\nu) + \boldsymbol{b}_{\mathcal{P}}. \quad (18)$$

Recursive update dynamics for $\boldsymbol{y}(k)$ and $\boldsymbol{\lambda}(k)$: To solve the optimization problem in (16), the projected gradient updates on the $\boldsymbol{y}(k)$ and $\boldsymbol{\lambda}(k)$ lead to the following neural dynamic iterations:

$$\boldsymbol{y}(k;\nu+1) = \boldsymbol{y}(k;\nu) + \eta_{\boldsymbol{y}}(\nu)\nabla_{\boldsymbol{y}(k)}\mathcal{L}(\boldsymbol{y}(k;\nu),\boldsymbol{\lambda}(k;\nu)), \quad (19)$$

$$\boldsymbol{\lambda}(k,\nu+1) = \text{ReLU}\left(\boldsymbol{\lambda}(k,\nu) - \eta_{\boldsymbol{\lambda}}(\nu)(\boldsymbol{b}_{\mathcal{P}} - \boldsymbol{A}_{\mathcal{P}}\boldsymbol{y}(k;\nu))\right), \quad (20)$$

where $\eta_{\boldsymbol{\lambda}}(\nu)$ denotes the learning rate for $\boldsymbol{\lambda}$ at iteration $\nu$. These iterations correspond to a recurrent neural network for which we can make the following observations: i) output neurons use linear activation functions since $\boldsymbol{y}(k)$ is unconstrained in (16), ii) the network contains $f$ interneurons corresponding to the Lagrangian vector $\boldsymbol{\lambda}$, where $f$ is the number of rows of $\boldsymbol{A}_{\mathcal{P}}$ in (16), or the number of $(n-1)$-faces of the corresponding polytope, iii) the nonnegativity of $\boldsymbol{\lambda}$ implies ReLU activation functions for interneurons. The neural network architecture corresponding to the neural dynamics in (19)- (20) is shown in Figure 1a, which has a layer of $f$ interneurons to impose the polytopic constraint in (1). The updates of $\boldsymbol{W}(k)$ and $\boldsymbol{B}_{\boldsymbol{y}}^{\zeta_{\boldsymbol{y}}}(k)$ follow the equations provided in Section 3.2. Although the architecture in Figure 1a allows implementation of arbitrary polytopic source domains; $f$ can be a large number. Alternatively, it is possible to consider the subset of polytopes in (2), which are described by individual properties and the relations of source components. Appendix C.5 derives the network dynamics for this *feature-based* polytope representation, and Figure 1b illustrates its particular realization. The number of interneurons in this case is equivalent to the number of sparsity constraints in (2), which can be much less than the number of faces of the polytope.

## 4    NUMERICAL EXPERIMENTS

In this section, we illustrate different domain selections for sources and compare the proposed CorInfoMax framework with existing batch algorithms and online biologically plausible neural network approaches. We demonstrate the correlated source separation capability of the proposed framework for both synthetic and natural sources. Additional experiments and details about their implementations are available in Appendix D.

### 4.1    SYNTHETICALLY CORRELATED SOURCE SEPARATION WITH ANTISPARSE SOURCES

To illustrate the correlated source separation capability of the online CorInfoMax framework for both nonnegative and signed antisparse sources, i.e. $\boldsymbol{s}(i) \in \mathcal{B}_{\ell_{\infty},+} \, \forall i$ and $\boldsymbol{s}(i) \in \mathcal{B}_{\ell_{\infty}} \, \forall i$, respectively, we consider a BSS setting with $n = 5$ sources and $m = 10$ mixtures. The 5-dimensional sources are generated using the Copula-T distribution with 4 degrees of freedom. We control the correlation level of the sources by adjusting a Toeplitz distribution parameter matrix with a first row of $[1 \quad \rho \quad \rho \quad \rho \quad \rho]$ for $\rho \in [0, 0.8]$. In each realization, we generate $N = 5 \times 10^{5}$ samples for each source and mix them through a random matrix $\boldsymbol{A} \in \mathbb{R}^{10 \times 5}$ whose entries are drawn from an i.i.d. standard normal distribution. Furthermore, we use an i.i.d. white Gaussian noise (WGN) corresponding to the signal-to-noise ratio (SNR) level of 30dB to corrupt the mixture signals. We use antisparse CorInfoMax network in Section C.1 and nonnegative CorInfoMax network in Appendix C.2 for these experiments. To compare, we also performed these experiments with biologically plausible algorithms: online BCA (Simsek & Erdogan, 2019), WSM (Bozkurt et al., 2022), NSM (Pehlevan et al., 2017a), BSM (Erdogan & Pehlevan, 2020), and batch altgorithms: ICA-Infomax (Bell & Sejnowski, 1995), LD-InfoMax (Erdogan, 2022), PMF (Tatli & Erdogan, 2021).

Figure 3 shows the signal-to-interference-plus-noise ratio (SINR) versus correlation level $\rho$ curves of different algorithms for nonnegative antisparse and antisparse source separation experiments. We observe that the proposed CorInfoMax approach achieves relatively high SINR results despite increasing $\rho$ in both cases. Although the WSM curve has a similar characteristic, its performance falls behind that of CorInfoMax. Moreover, the LD-InfoMax and PMF algorithms typically achieve the best results, as expected, due to their batch learning settings. Furthermore, the performance

of NSM, BSM, and ICA-InfoMax degrades with increasing source correlation because these approaches assume uncorrelated or independent sources.

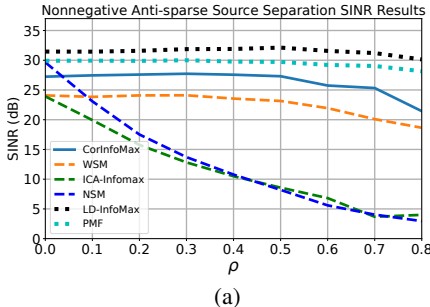 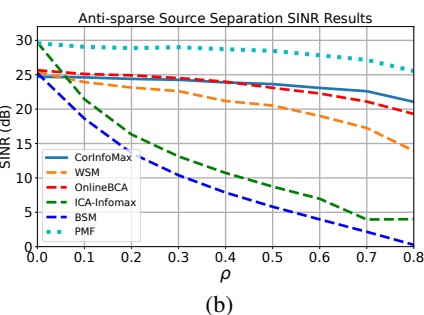

(a)                                    (b)

Figure 3: The SINR performances of CorInfoMax (ours), LD-InfoMax, PMF, ICA-InfoMax, NSM, and BSM, averaged over $100$ realizations, (y-axis) with respect to the correlation factor $\rho$ (x-axis). SINR vs. $\rho$ curves for (a) nonnegative antisparse ($\mathcal{B}_{\ell_\infty,+}$), (b) antisparse ($\mathcal{B}_{\ell_\infty}$) source domains.

## 4.2 VIDEO SEPARATION

To provide a visual example and illustrate a real naturally correlated source scenario, we consider the following video separation setup: $3$ videos of $10$ seconds are mixed to generate $5$ mixture videos. The average (across frames) and maximum Pearson correlation coefficients for these three sources are $\rho_{12}^{\text{average}} = -0.1597, \rho_{13}^{\text{average}} = -0.1549, \rho_{23}^{\text{average}} = 0.3811$ and $\rho_{12}^{\text{maximum}} = 0.3139, \rho_{13}^{\text{maximum}} = 0.2587, \rho_{23}^{\text{maximum}} = 0.5173$, respectively. We use a random mixing matrix $\boldsymbol{A} \in \mathbb{R}^{5 \times 3}$ with positive entries (to ensure nonnegative mixtures so that they can be displayed as proper images without loss of generality), which is provided in Appendix D.3.3. Since the image pixels are in the set $[0, 1]$, we use the nonnegative antisparse CorInfoMax network to separate the original videos. The demo video (which is available in supplementary files and whose link is provided in the footnote [2]) visually demonstrates the separation process by the proposed approach over time. The first and second rows of the demo are the $3$ source videos and $3$ of the $5$ mixture videos, respectively. The last row contains the source estimates obtained by the CorInfoMax network during its unsupervised learning process. We observe that the output frames become visually better as time progresses and start to represent individual sources. In the end, the CorInfoMax network is trained to a stage of near-perfect separation, with peak signal-to-noise ratio (PSNR) levels of $35.60$dB, $48.07$dB, and $44.58$dB for each source, respectively. Further details for this experiment can be found in the Appendix D.3.3.

## 5 CONCLUSION

In this article, we propose an information-theoretic framework for generating biologically plausible neural networks that are capable of separating both independent and correlated sources. The proposed CorInfoMax framework can be applied to infinitely many source domains, enabling a diverse set of source characterizations. In addition to solving unsupervised linear inverse problems, CorInfoMax networks have the potential to generate structured embeddings from observations based on the choice of source domains. In fact, as a future extension, we consider representation frameworks that learn desirable source domain representations by adapting the output-interneuron connections in Figure 1a. Finally, the proposed unsupervised framework and its potential supervised extensions can be useful for neuromorphic systems that are bound to use local learning rules.

In terms of limitations, we can list the computational complexity for simulating such networks in conventional computers, mainly due to the loop-based recurrent output computation. However, as described in Appendix D.7, the neural networks generated by the proposed framework have computational loads similar to the existing biologically plausible BSS neural networks.

---

[2]https://figshare.com/s/a3fb926f273235068053

## 6 REPRODUCIBILITY

To ensure the reproducibility of our results, we provide

i. Detailed mathematical description of the algorithms for different source domains and their neural network implementations in Section 3.2, Appendix C.1, Appendix C.2, Appendix C.3, Appendix C.4 and Appendix C.5,

ii. Detailed information on the simulation settings of the experiments in Section 4 in the main article, and Appendix D,

iii. Full list of hyperparameter sets used in these experiments in Table 3, Table 4 in Appendix D.4,

iv. Ablation studies on hyperparameters in Appendix D.5,

v. Algorithm descriptions for special source domains in pseudo-code format in Appendix D.1,

vi. Python scripts and notebooks for individual experiments to replicate the reported results in the supplementary zip file as well as in https://github.com/BariscanBozkurt/Bio-Plausible-CorrInfoMax.

## 7 ETHICS STATEMENT

Related to the algorithmic framework we propose in this article, we see no immediate ethical concerns. In addition, the datasets that we use have no known or reported ethical issues, to the best of our knowledge.

## 8 ACKNOWLEDGMENTS AND DISCLOSURE OF FUNDING

This work was supported by KUIS AI Center Research Award. CP was supported by an NSF Award (DMS-2134157) and the Intel Corporation through the Intel Neuromorphic Research Community.

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

# A  APPENDIX

## A.1  INFORMATION THEORETIC DEFINITIONS

In this section, we review the logarithm-determinant (LD) entropy measure and mutual information for the BSS setting introduced in Section 2.1 based on Erdogan (2022). Note that in this article, we refer to LD-mutual information synonymously as correlative mutual information. For a finite set of vectors $\mathbb{X} = \{\boldsymbol{x}(1), \boldsymbol{x}(2), \dots, \boldsymbol{x}(N)\} \subset \mathbb{R}^m$ with a sample covariance matrix $\hat{\boldsymbol{R}}_{\boldsymbol{x}} = \dfrac{1}{N}\boldsymbol{X}\boldsymbol{X}^T - \dfrac{1}{N^2}\boldsymbol{X}\mathbf{1}\mathbf{1}^T\boldsymbol{X}^T$, where $\boldsymbol{X}$ is defined as $\boldsymbol{X} = [\ \boldsymbol{x}(1)\ \ \boldsymbol{x}(2)\ \ \dots\ \ \boldsymbol{x}(N)\ ]$, the deterministic LD-entropy is defined in Erdogan (2022) as

$$H(\boldsymbol{X})_{\text{LD}}^{(\epsilon)} = \frac{1}{2}\log\det(\hat{\boldsymbol{R}}_{\boldsymbol{x}} + \epsilon\boldsymbol{I}) + \frac{m}{2}\log(2\pi e) \tag{A.1}$$

where $\epsilon > 0$ is a small number to keep the expression away from $-\infty$. If the sample covariance $\hat{\boldsymbol{R}}_{\boldsymbol{x}}$ in this expression is replaced with the true covariance, then $H(\boldsymbol{x})_{\text{LD}}^{(0)}$ coincides with the Shannon differential entropy for a Gaussian vector $\boldsymbol{x}$. Moreover, a deterministic joint LD-entropy of two sets of vectors $\mathbb{X} \subset \mathbb{R}^m$ and $\mathbb{Y} \subset \mathbb{R}^n$ can be defined as

$$\begin{aligned}
H(\boldsymbol{X}, \boldsymbol{Y})_{\text{LD}}^{(\epsilon)} &= \frac{1}{2}\log\det(\hat{\boldsymbol{R}}_{\begin{bmatrix}\boldsymbol{x}\\\boldsymbol{y}\end{bmatrix}} + \epsilon\boldsymbol{I}) + \frac{m+n}{2}\log\det(2\pi e) \\
&= \frac{1}{2}\log\det\left(\begin{bmatrix}\hat{\boldsymbol{R}}_{\boldsymbol{x}} + \epsilon\boldsymbol{I} & \hat{\boldsymbol{R}}_{\boldsymbol{xy}}\\ \hat{\boldsymbol{R}}_{\boldsymbol{yx}} & \hat{\boldsymbol{R}}_{\boldsymbol{y}} + \epsilon\boldsymbol{I}\end{bmatrix}\right) + \frac{m+n}{2}\log\det(2\pi e) \\
&= \frac{1}{2}\log\left(\det(\hat{\boldsymbol{R}}_{\boldsymbol{x}} + \epsilon\boldsymbol{I})\det(\hat{\boldsymbol{R}}_{\boldsymbol{y}} + \epsilon\boldsymbol{I} - \hat{\boldsymbol{R}}_{\boldsymbol{xy}}^T(\hat{\boldsymbol{R}}_{\boldsymbol{x}} + \epsilon\boldsymbol{I})^{-1}\hat{\boldsymbol{R}}_{\boldsymbol{yx}})\right) \\
&\quad + \frac{m+n}{2}\log\det(2\pi e) \\
&= \frac{1}{2}\log\det(\hat{\boldsymbol{R}}_{\boldsymbol{x}} + \epsilon\boldsymbol{I}) + \frac{m}{2}\log(2\pi e) + \frac{1}{2}\log\det(\hat{\boldsymbol{R}}_{\boldsymbol{e}} + \epsilon\boldsymbol{I}) + \frac{n}{2}\log(2\pi e) \\
&= H(\boldsymbol{X})_{\text{LD}}^{(\epsilon)} + H(\boldsymbol{Y}|_L\boldsymbol{X})_{\text{LD}}^{(\epsilon)}
\end{aligned} \tag{A.2}$$

where $\hat{\boldsymbol{R}}_{\boldsymbol{e}} = \hat{\boldsymbol{R}}_{\boldsymbol{y}} - \hat{\boldsymbol{R}}_{\boldsymbol{xy}}^T(\hat{\boldsymbol{R}}_{\boldsymbol{x}} + \epsilon\boldsymbol{I})^{-1}\hat{\boldsymbol{R}}_{\boldsymbol{yx}}$, and $\hat{\boldsymbol{R}}_{\boldsymbol{xy}} = \frac{1}{N}\boldsymbol{X}\boldsymbol{Y}^T = \hat{\boldsymbol{R}}_{\boldsymbol{yx}}^T$. In (A.2), the notation $\boldsymbol{Y}|_L\boldsymbol{X}$ is used to signify $H(\boldsymbol{Y}|_L\boldsymbol{X})_{\text{LD}}^{(\epsilon)} \neq H(\boldsymbol{Y}|\boldsymbol{X})_{\text{LD}}^{(\epsilon)}$ as the latter requires the use of $\hat{\boldsymbol{R}}_{\boldsymbol{y}|\boldsymbol{x}}$ instead of $\hat{\boldsymbol{R}}_{\boldsymbol{e}}$. Moreover, $H(\boldsymbol{Y}|_L\boldsymbol{X})_{\text{LD}}^{(\epsilon)}$ corresponds to the log-determinant of the error sample covariance of the best linear minimum mean squared estimate (MMSE) of $\boldsymbol{y}$ from $\boldsymbol{x}$. To verify that in the zero-mean and noiseless case, consider the MMSE estimate $\hat{\boldsymbol{y}} = \boldsymbol{W}\boldsymbol{x}$ for which the solution is given by $\boldsymbol{W} = \boldsymbol{R}_{\boldsymbol{yx}}\boldsymbol{R}_{\boldsymbol{x}}^{-1} = \boldsymbol{R}_{\boldsymbol{xy}}^T\boldsymbol{R}_{\boldsymbol{x}}^{-1}$ (Kailath et al., 2000). Then $\boldsymbol{R}_{\hat{\boldsymbol{y}}} = \mathbb{E}[\hat{\boldsymbol{y}}\hat{\boldsymbol{y}}^T] = \mathbb{E}[\boldsymbol{R}_{\boldsymbol{xy}}^T\boldsymbol{R}_{\boldsymbol{x}}^{-1}\boldsymbol{x}\boldsymbol{x}^T\boldsymbol{R}_{\boldsymbol{x}}^{-1}\boldsymbol{R}_{\boldsymbol{xy}}] = \boldsymbol{R}_{\boldsymbol{xy}}^T\boldsymbol{R}_{\boldsymbol{x}}^{-1}\boldsymbol{R}_{\boldsymbol{xy}}$. Therefore, if the error is defined as $\boldsymbol{e} = \boldsymbol{y} - \hat{\boldsymbol{y}}$, its covariance matrix can be found as desired, i.e., $\boldsymbol{R}_{\boldsymbol{e}} = \boldsymbol{R}_{\boldsymbol{y}} - \boldsymbol{R}_{\hat{\boldsymbol{y}}} = \boldsymbol{R}_{\boldsymbol{y}} - \boldsymbol{R}_{\boldsymbol{xy}}^T\boldsymbol{R}_{\boldsymbol{x}}^{-1}\boldsymbol{R}_{\boldsymbol{xy}}$.

The LD-mutual information for $\boldsymbol{X}$ and $\boldsymbol{Y}$ can be defined based on the equations (A.1) and (A.2) as

$$\begin{aligned}
I^{(\epsilon)}(\boldsymbol{X}, \boldsymbol{Y}) &= H_{\text{LD}}^{(\epsilon)}(\boldsymbol{Y}) - H_{\text{LD}}^{(\epsilon)}(\boldsymbol{Y}|_L\boldsymbol{X}) = H_{\text{LD}}^{(\epsilon)}(\boldsymbol{X}) - H_{\text{LD}}^{(\epsilon)}(\boldsymbol{X}|_L\boldsymbol{Y}) \\
&= \frac{1}{2}\log\det(\hat{\boldsymbol{R}}_{\boldsymbol{y}} + \epsilon\boldsymbol{I}) - \frac{1}{2}\log\det(\hat{\boldsymbol{R}}_{\boldsymbol{y}} - \hat{\boldsymbol{R}}_{\boldsymbol{xy}}^T(\hat{\boldsymbol{R}}_{\boldsymbol{x}} + \epsilon\boldsymbol{I})^{-1}\hat{\boldsymbol{R}}_{\boldsymbol{yx}} + \epsilon\boldsymbol{I}) + C \\
&= \frac{1}{2}\log\det(\hat{\boldsymbol{R}}_{\boldsymbol{x}} + \epsilon\boldsymbol{I}) - \frac{1}{2}\log\det(\hat{\boldsymbol{R}}_{\boldsymbol{x}} - \hat{\boldsymbol{R}}_{\boldsymbol{yx}}^T(\hat{\boldsymbol{R}}_{\boldsymbol{y}} + \epsilon\boldsymbol{I})^{-1}\hat{\boldsymbol{R}}_{\boldsymbol{xy}} + \epsilon\boldsymbol{I}) + C,
\end{aligned} \tag{A.3}$$

where $C = \dfrac{m}{2}\log(2\pi e) + \dfrac{n}{2}\log(2\pi e)$ is a constant.

# B  GRADIENT DERIVATIONS FOR ONLINE OPTIMIZATION OBJECTIVE

Assuming that the mapping $\boldsymbol{W}(k)$ changes slowly over time, the current output $\boldsymbol{y}(k)$ can be implicitly defined by the projected gradient ascent with neural dynamics. To derive the corresponding neural dynamics for the output $\boldsymbol{y}(k)$, we need to calculate the gradient of the objective 5a with respect

to $\boldsymbol{y}(k)$. First, we consider the derivative of both $\log \det(\hat{\boldsymbol{R}}_{\boldsymbol{y}}^{\zeta_y}(k) + \epsilon \boldsymbol{I})$ and $\log \det(\hat{\boldsymbol{R}}_{\boldsymbol{e}}^{\zeta_e}(k) + \epsilon \boldsymbol{I})$ with respect to $\boldsymbol{y}(k)$.

$$\frac{\partial \log \det(\hat{\boldsymbol{R}}_{\boldsymbol{y}}^{\zeta_y}(k) + \epsilon \boldsymbol{I})}{\partial y_i(k)} = Tr\left(\nabla_{(\hat{\boldsymbol{R}}_{\boldsymbol{y}}^{\zeta_y}(k)+\epsilon \boldsymbol{I})} \log \det(\hat{\boldsymbol{R}}_{\boldsymbol{y}}^{\zeta_y}(k) + \epsilon \boldsymbol{I}) \frac{\partial \hat{\boldsymbol{R}}_{\boldsymbol{y}}^{\zeta_y}(k)}{\partial y_i(k)}\right),$$

where $y_i(k)$ is the $i$-th element of the vector $\boldsymbol{y}(k)$, and

$$\nabla_{(\hat{\boldsymbol{R}}_{\boldsymbol{y}}^{\zeta_y}(k)+\epsilon \boldsymbol{I})} \log \det(\hat{\boldsymbol{R}}_{\boldsymbol{y}}^{\zeta_y}(k) + \epsilon \boldsymbol{I}) = (\hat{\boldsymbol{R}}_{\boldsymbol{y}}^{\zeta_y}(k) + \epsilon \boldsymbol{I})^{-1}, \tag{A.4}$$

$$\frac{\partial \hat{\boldsymbol{R}}_{\boldsymbol{y}}^{\zeta_y}(k)}{\partial y_i(k)} = \frac{1 - \zeta_y}{1 - \zeta_y^k}(\boldsymbol{y}(k)\boldsymbol{e}^{(i)T} + \boldsymbol{e}^{(i)}\boldsymbol{y}(k)^T). \tag{A.5}$$

In (A.5), $\boldsymbol{e}^{(i)}$ denotes the standard basis vector with a 1 at position $i$ and should not be confused with the error vector $\boldsymbol{e}(k)$. Combining (A.4) and (A.5), we obtain the following result:

$$\frac{\partial \log \det(\hat{\boldsymbol{R}}_{\boldsymbol{y}}^{\zeta_y}(k) + \epsilon \boldsymbol{I})}{\partial y_i(k)} = 2\frac{1 - \zeta_y}{1 - \zeta_y^k}\boldsymbol{e}^{(i)T}(\hat{\boldsymbol{R}}_{\boldsymbol{y}}^{\zeta_y}(k) + \epsilon \boldsymbol{I})^{-1}\boldsymbol{y}(k),$$

which leads to

$$\nabla_{\boldsymbol{y}(k)} \log \det(\hat{\boldsymbol{R}}_{\boldsymbol{y}}^{\zeta_y}(k) + \epsilon \boldsymbol{I}) = 2\frac{1 - \zeta_y}{1 - \zeta_y^k}(\hat{\boldsymbol{R}}_{\boldsymbol{y}}^{\zeta_y}(k) + \epsilon \boldsymbol{I})^{-1}\boldsymbol{y}(k). \tag{A.6}$$

If we apply the same procedure to obtain the gradient of $\log \det(\hat{\boldsymbol{R}}_{\boldsymbol{e}}^{\zeta_e}(k) + \epsilon \boldsymbol{I})$ with respect to $\boldsymbol{e}(k)$, we obtain

$$\nabla_{\boldsymbol{e}(k)} \log \det(\hat{\boldsymbol{R}}_{\boldsymbol{e}}^{\zeta_e}(k) + \epsilon \boldsymbol{I}) = 2\frac{1 - \zeta_e}{1 - \zeta_e^k}(\hat{\boldsymbol{R}}_{\boldsymbol{e}}^{\zeta_e}(k) + \epsilon \boldsymbol{I})^{-1}\boldsymbol{e}(k).$$

Using the composition rule, we can obtain the gradient of $\log \det(\hat{\boldsymbol{R}}_{\boldsymbol{e}}^{\zeta_e}(k) + \epsilon \boldsymbol{I})$ with respect to $\boldsymbol{y}(k)$ as follows:

$$\nabla_{\boldsymbol{y}(k)} \log \det(\hat{\boldsymbol{R}}_{\boldsymbol{e}}^{\zeta_e}(k) + \epsilon \boldsymbol{I}) = \underbrace{\frac{\partial \boldsymbol{e}(k)}{\partial \boldsymbol{y}(k)}}_{\boldsymbol{I}_n} \nabla_{\boldsymbol{e}(k)} \log \det(\hat{\boldsymbol{R}}_{\boldsymbol{e}}^{\zeta_e}(k) + \epsilon \boldsymbol{I})$$

$$= 2\frac{1 - \zeta_e}{1 - \zeta_e^k}(\hat{\boldsymbol{R}}_{\boldsymbol{e}}^{\zeta_e}(k) + \epsilon \boldsymbol{I})^{-1}\boldsymbol{e}(k). \tag{A.7}$$

Finally, combining the results from (A.6) and (A.7), we obtain the derivative of the objective function $\mathcal{J}(\boldsymbol{y}(k))$ with respect to $\boldsymbol{y}(k)$

$$\nabla_{\boldsymbol{y}(k)} \mathcal{J}(\boldsymbol{y}(k)) = \frac{1}{2}\nabla_{\boldsymbol{y}(k)} \log \det(\hat{\boldsymbol{R}}_{\boldsymbol{y}}^{\zeta_y}(k) + \epsilon \boldsymbol{I}) - \frac{1}{2}\nabla_{\boldsymbol{y}(k)} \log \det(\hat{\boldsymbol{R}}_{\boldsymbol{e}}^{\zeta_e}(k) + \epsilon \boldsymbol{I})$$

$$= \frac{1 - \zeta_y}{1 - \zeta_y^k}(\hat{\boldsymbol{R}}_{\boldsymbol{y}}^{\zeta_y}(k) + \epsilon \boldsymbol{I})^{-1}\boldsymbol{y}(k) - \frac{1 - \zeta_e}{1 - \zeta_e^k}(\hat{\boldsymbol{R}}_{\boldsymbol{e}}^{\zeta_e}(k) + \epsilon \boldsymbol{I})^{-1}\boldsymbol{e}(k) \tag{A.8}$$

For further simplification of (A.8), we define the recursions for $\hat{\boldsymbol{R}}_{\boldsymbol{y}}^{\zeta_y}(k)^{-1}$ and $\hat{\boldsymbol{R}}_{\boldsymbol{e}}^{\zeta_e}(k)^{-1}$ based on the recursive definitions of the corresponding correlation matrices. Based on the definition in (4), we can write

$$\hat{\boldsymbol{R}}_{\boldsymbol{y}}^{\zeta_y}(k) + \epsilon \boldsymbol{I} = \frac{1 - \zeta_y^{k-1}}{1 - \zeta_y^k}\zeta_y(\hat{\boldsymbol{R}}_{\boldsymbol{y}}^{\zeta_y}(k-1) + \epsilon \boldsymbol{I}) + \frac{1 - \zeta_y}{1 - \zeta_y^k}\boldsymbol{y}(k)\boldsymbol{y}(k)^T + \frac{1 - \zeta_y}{1 - \zeta_y^k}\epsilon \boldsymbol{I}$$

$$\approx \frac{1 - \zeta_y^{k-1}}{1 - \zeta_y^k}\zeta_y(\hat{\boldsymbol{R}}_{\boldsymbol{y}}^{\zeta_y}(k-1) + \epsilon \boldsymbol{I}) + \frac{1 - \zeta_y}{1 - \zeta_y^k}\boldsymbol{y}(k)\boldsymbol{y}(k)^T \tag{A.9}$$

Using the assumption in (A.9), we take the inverse of both sides and apply the matrix inversion lemma (similar to its use in the derivation of the RLS algorithm Kailath et al. (2000)) to obtain

$$
(\hat{\boldsymbol{R}}_{\boldsymbol{y}}^{\zeta_{\boldsymbol{y}}}(k) + \epsilon \boldsymbol{I})^{-1} = \frac{1 - \zeta_{\boldsymbol{y}}^{k}}{\zeta_{\boldsymbol{y}} - \zeta_{\boldsymbol{y}}^{k}} \Big( (\hat{\boldsymbol{R}}_{\boldsymbol{y}}^{\zeta_{\boldsymbol{y}}}(k-1) + \epsilon \boldsymbol{I})^{-1}
$$
$$
- \gamma_{\boldsymbol{y}}(k)(\hat{\boldsymbol{R}}_{\boldsymbol{y}}^{\zeta_{\boldsymbol{y}}}(k-1) + \epsilon \boldsymbol{I})^{-1} \boldsymbol{y}(k) \boldsymbol{y}(k)^T (\hat{\boldsymbol{R}}_{\boldsymbol{y}}^{\zeta_{\boldsymbol{y}}}(k-1) + \epsilon \boldsymbol{I})^{-1} \Big), \quad \text{(A.10)}
$$

where

$$
\gamma_{\boldsymbol{y}}(k) = \left( \frac{\zeta_{\boldsymbol{y}} - \zeta_{\boldsymbol{y}}^{k}}{1 - \zeta_{\boldsymbol{y}}} + \boldsymbol{y}(k)^T (\hat{\boldsymbol{R}}_{\boldsymbol{y}}^{\zeta_{\boldsymbol{y}}}(k-1) + \epsilon \boldsymbol{I})^{-1} \boldsymbol{y}(k) \right)^{-1}. \quad \text{(A.11)}
$$

We apply the same procedure to obtain the inverse of $(\hat{\boldsymbol{R}}_{\boldsymbol{e}}^{\zeta_{\boldsymbol{e}}}(k) + \epsilon \boldsymbol{I})^{-1}$:

$$
(\hat{\boldsymbol{R}}_{\boldsymbol{e}}^{\zeta_{\boldsymbol{e}}}(k) + \epsilon \boldsymbol{I})^{-1} = \frac{1 - \zeta_{\boldsymbol{e}}^{k}}{\zeta_{\boldsymbol{e}} - \zeta_{\boldsymbol{e}}^{k}} \Big( (\hat{\boldsymbol{R}}_{\boldsymbol{e}}^{\zeta_{\boldsymbol{e}}}(k-1) + \epsilon \boldsymbol{I})^{-1}
$$
$$
- \gamma_{\boldsymbol{e}}(k)(\hat{\boldsymbol{R}}_{\boldsymbol{e}}^{\zeta_{\boldsymbol{e}}}(k-1) + \epsilon \boldsymbol{I})^{-1} \boldsymbol{e}(k) \boldsymbol{e}(k)^T (\hat{\boldsymbol{R}}_{\boldsymbol{e}}^{\zeta_{\boldsymbol{e}}}(k-1) + \epsilon \boldsymbol{I})^{-1} \Big), \quad \text{(A.12)}
$$

where

$$
\gamma_{\boldsymbol{e}}(k) = \left( \frac{\zeta_{\boldsymbol{e}} - \zeta_{\boldsymbol{e}}^{k}}{1 - \zeta_{\boldsymbol{e}}} + \boldsymbol{e}(k)^T (\hat{\boldsymbol{R}}_{\boldsymbol{e}}^{\zeta_{\boldsymbol{e}}}(k-1) + \epsilon \boldsymbol{I})^{-1} \boldsymbol{e}(k) \right)^{-1}. \quad \text{(A.13)}
$$

Note that plugging (A.10) into the first part of (A.8) yields the following simplification:

$$
\frac{1 - \zeta_{\boldsymbol{y}}}{1 - \zeta_{\boldsymbol{y}}^{k}} (\hat{\boldsymbol{R}}(k) + \epsilon \boldsymbol{I})^{-1} \boldsymbol{y}(k) = \frac{1 - \zeta_{\boldsymbol{y}}}{1 - \zeta_{\boldsymbol{y}}^{k}} \frac{1 - \zeta_{\boldsymbol{y}}^{k}}{\zeta_{\boldsymbol{y}} - \zeta_{\boldsymbol{y}}^{k}} \Big( (\hat{\boldsymbol{R}}(k-1) + \epsilon \boldsymbol{I})^{-1}
$$
$$
- \gamma_{\boldsymbol{y}}(k)(\hat{\boldsymbol{R}}(k-1) + \epsilon \boldsymbol{I})^{-1} \boldsymbol{y}(k) \boldsymbol{y}(k)^T (\hat{\boldsymbol{R}}(k-1) + \epsilon \boldsymbol{I})^{-1} \Big) \boldsymbol{y}(k)
$$
$$
= \frac{1 - \zeta_{\boldsymbol{y}}}{\zeta_{\boldsymbol{y}} - \zeta_{\boldsymbol{y}}^{k}} \Big( (\hat{\boldsymbol{R}}(k-1) + \epsilon \boldsymbol{I})^{-1}
$$
$$
- \frac{(\hat{\boldsymbol{R}}(k-1) + \epsilon \boldsymbol{I})^{-1} \boldsymbol{y}(k) \boldsymbol{y}(k)^T (\hat{\boldsymbol{R}}(k-1) + \epsilon \boldsymbol{I})^{-1}}{\frac{\zeta_{\boldsymbol{y}} - \zeta_{\boldsymbol{y}}^{k}}{1 - \zeta_{\boldsymbol{y}}} + \boldsymbol{y}(k)^T (\hat{\boldsymbol{R}}(k-1) + \epsilon \boldsymbol{I})^{-1} \boldsymbol{y}(k)} \Big) \boldsymbol{y}(k)
$$
$$
= \frac{1 - \zeta_{\boldsymbol{y}}}{\zeta_{\boldsymbol{y}} - \zeta_{\boldsymbol{y}}^{k}} \left( \frac{\frac{\zeta_{\boldsymbol{y}} - \zeta_{\boldsymbol{y}}^{k}}{1 - \zeta_{\boldsymbol{y}}} (\hat{\boldsymbol{R}}(k-1) + \epsilon \boldsymbol{I})^{-1} \boldsymbol{y}(k)}{\frac{\zeta_{\boldsymbol{y}} - \zeta_{\boldsymbol{y}}^{k}}{1 - \zeta_{\boldsymbol{y}}} + \boldsymbol{y}(k)^T (\hat{\boldsymbol{R}}(k-1) + \epsilon \boldsymbol{I})^{-1} \boldsymbol{y}(k)} \right)
$$
$$
+ \frac{1 - \zeta_{\boldsymbol{y}}}{\zeta_{\boldsymbol{y}} - \zeta_{\boldsymbol{y}}^{k}} \left( \frac{(\hat{\boldsymbol{R}}(k-1) + \epsilon \boldsymbol{I})^{-1} \boldsymbol{y}(k) \boldsymbol{y}(k)^T (\hat{\boldsymbol{R}}(k-1) + \epsilon \boldsymbol{I})^{-1} \boldsymbol{y}(k)}{\frac{\zeta_{\boldsymbol{y}} - \zeta_{\boldsymbol{y}}^{k}}{1 - \zeta_{\boldsymbol{y}}} + \boldsymbol{y}(k)^T (\hat{\boldsymbol{R}}(k-1) + \epsilon \boldsymbol{I})^{-1} \boldsymbol{y}(k)} \right)
$$
$$
- \frac{1 - \zeta_{\boldsymbol{y}}}{\zeta_{\boldsymbol{y}} - \zeta_{\boldsymbol{y}}^{k}} \left( \frac{(\hat{\boldsymbol{R}}(k-1) + \epsilon \boldsymbol{I})^{-1} \boldsymbol{y}(k) \boldsymbol{y}(k)^T (\hat{\boldsymbol{R}}(k-1) + \epsilon \boldsymbol{I})^{-1} \boldsymbol{y}(k)}{\frac{\zeta_{\boldsymbol{y}} - \zeta_{\boldsymbol{y}}^{k}}{1 - \zeta_{\boldsymbol{y}}} + \boldsymbol{y}(k)^T (\hat{\boldsymbol{R}}(k-1) + \epsilon \boldsymbol{I})^{-1} \boldsymbol{y}(k)} \right)
$$
$$
= \gamma_{\boldsymbol{y}}(k)(\hat{\boldsymbol{R}}_{\boldsymbol{y}}^{\zeta_{\boldsymbol{y}}}(k-1) + \epsilon \boldsymbol{I})^{-1} \boldsymbol{y}(k).
$$

A similar simplification can be obtained for (A.12), and incorporating these simplifications into (A.8) yields

$$
\nabla_{\boldsymbol{y}(k)} \mathcal{J}(\boldsymbol{y}(k)) = \gamma_{\boldsymbol{y}}(k) \boldsymbol{B}_{\boldsymbol{y}}^{\zeta_{\boldsymbol{y}}}(k) \boldsymbol{y}(k) - \gamma_{\boldsymbol{e}}(k) \boldsymbol{B}_{\boldsymbol{e}}^{\zeta_{\boldsymbol{e}}}(k) \boldsymbol{e}(k), \quad \text{(A.14)}
$$

where we denote $(\hat{\boldsymbol{R}}_{\boldsymbol{y}}^{\zeta_{\boldsymbol{y}}}(k) + \epsilon \boldsymbol{I})^{-1}$ and $(\hat{\boldsymbol{R}}_{\boldsymbol{e}}^{\zeta_{\boldsymbol{e}}}(k) + \epsilon \boldsymbol{I})^{-1}$ by $\boldsymbol{B}_{\boldsymbol{y}}^{\zeta_{\boldsymbol{y}}}(k+1)$ and $\boldsymbol{B}_{\boldsymbol{e}}^{\zeta_{\boldsymbol{e}}}(k+1)$ for simplicity, respectively.

## C  SUPPLEMENTARY ON THE NETWORK STRUCTURES FOR THE EXAMPLE DOMAINS

We can generalize the procedure for obtaining CorInfoMax BSS networks in Section 3.2 for other source domains. The choice of source domain would affect the structure of the output layer and the potential inclusion of additional interneurons. Table 1 summarizes the output dynamics for special source domains provided in Section 2.1, for which the derivations are provided in the following subsections.

Table 1: Example source domains and the corresponding CorInfoMax network dynamics.

| Source Domain | Output Dynamics | Output Activation |
|---|---|---|
| $\mathcal{P} = \Delta$ | $\nabla_{\boldsymbol{y}(k)} J(\boldsymbol{y}(k;\nu)) = \gamma_{\boldsymbol{y}} \boldsymbol{B}_{\boldsymbol{y}}^{\zeta_{\boldsymbol{y}}}(k-1)\boldsymbol{y}(k;\nu) - \gamma_{\boldsymbol{e}} \boldsymbol{B}_{\boldsymbol{e}}^{\zeta_{\boldsymbol{e}}}(k-1)\boldsymbol{e}(k;\nu),$ $\boldsymbol{y}(k;\nu+1) = \text{ReLU}\left(\boldsymbol{y}(k;\nu) + \eta_{\boldsymbol{y}}(\nu)\nabla_{\boldsymbol{y}(k)} J(\boldsymbol{y}(k;\nu)) - \lambda(\nu)\right),$ $\lambda(k;\nu+1) = \lambda(k;\nu) - \eta_{\lambda}(\nu)\left(1 - \left(\sum_{i=1}^{n} y_i(k;\nu+1)\right)\right).$ | ReLU(x) |
| $\mathcal{P} = \mathcal{B}_{\ell_\infty}$ | $\nabla_{\boldsymbol{y}(k)} J(\boldsymbol{y}(k;\nu)) = \gamma_{\boldsymbol{y}} \boldsymbol{B}_{\boldsymbol{y}}^{\zeta_{\boldsymbol{y}}}(k-1)\boldsymbol{y}(k;\nu) - \gamma_{\boldsymbol{e}} \boldsymbol{B}_{\boldsymbol{e}}^{\zeta_{\boldsymbol{e}}}(k-1)\boldsymbol{e}(k;\nu),$ $\boldsymbol{y}(k;\nu+1) = \sigma_1\left(\boldsymbol{y}(k;\nu) + \eta_{\boldsymbol{y}}(\nu)\nabla_{\boldsymbol{y}(k)} J(\boldsymbol{y}(k;\nu))\right),$ | $\sigma_1(x)$ |
| $\mathcal{P} = \mathcal{B}_{\ell_\infty,+}$ | $\nabla_{\boldsymbol{y}(k)} J(\boldsymbol{y}(k;\nu)) = \gamma_{\boldsymbol{y}} \boldsymbol{B}_{\boldsymbol{y}}^{\zeta_{\boldsymbol{y}}}(k)\boldsymbol{y}(k;\nu) - \gamma_{\boldsymbol{e}} \boldsymbol{B}_{\boldsymbol{e}}^{\zeta_{\boldsymbol{e}}}(k)\boldsymbol{e}(k;\nu),$ $\boldsymbol{y}(k;\nu+1) = \sigma_+\left(\boldsymbol{y}(k;\nu) + \eta_{\boldsymbol{y}}(\nu)\nabla_{\boldsymbol{y}(k)} J(\boldsymbol{y}(k;\nu))\right),$ | $\sigma_+(x)$ |
| $\mathcal{P} = \mathcal{B}_{\ell_1,+}$ | $\nabla_{\boldsymbol{y}(k)} J(\boldsymbol{y}(k;\nu)) = \gamma_{\boldsymbol{y}} \boldsymbol{B}_{\boldsymbol{y}}^{\zeta_{\boldsymbol{y}}}(k-1)\boldsymbol{y}(k;\nu) - \gamma_{\boldsymbol{e}} \boldsymbol{B}_{\boldsymbol{e}}^{\zeta_{\boldsymbol{e}}}(k-1)\boldsymbol{e}(k;\nu),$ $\boldsymbol{y}(k;\nu+1) = \text{ReLU}\left(\boldsymbol{y}(k;\nu) + \eta_{\boldsymbol{y}}(\nu)\nabla_{\boldsymbol{y}(k)} J(\boldsymbol{y}(k;\nu))\right),$ $\lambda(k;\nu+1) = \text{ReLU}\left(\lambda(k;\nu) - \eta_{\lambda}(\nu)\left(1 - \left(\sum_{i=1}^{n} y_i(k;\nu+1)\right)\right)\right).$ | ReLU(x) |

### C.1  DESCRIPTION OF THE NETWORK DYNAMICS FOR ANTISPARSE SOURCES

We consider the source domain $\mathcal{P} = \mathcal{B}_{\ell_\infty}$ which corresponds to antisparse sources. Similar to the sparse CorInfoMax example in Section 3.2, we derive the network corresponding to the antisparse CorInfoMax network through the projected gradient ascent method. Since projection onto the $\mathcal{P} = \mathcal{B}_{\ell_\infty}$ is an elementwise clipping operation, unlike the sparse CorInfoMax case, we do not require any interneurons related to the projection operation.

Recursive update dynamics for $\boldsymbol{y}(k)$: Based on the gradient of (5a) with respect to $\boldsymbol{y}(k)$ in (A.14), derived in Appendix B, we can write the corresponding projected gradient ascent iterations for (5) as

$$\boldsymbol{e}(k;\nu) = \boldsymbol{y}(k;\nu) - \boldsymbol{W}(k)\boldsymbol{x}(k),$$

$$\nabla_{\boldsymbol{y}(k)} \mathcal{J}(\boldsymbol{y}(k;\nu)) = \gamma_{\boldsymbol{y}} \boldsymbol{B}_{\boldsymbol{y}}^{\zeta_{\boldsymbol{y}}}(k)\boldsymbol{y}(k;\nu) - \gamma_{\boldsymbol{e}} \boldsymbol{B}_{\boldsymbol{e}}^{\zeta_{\boldsymbol{e}}}(k)\boldsymbol{e}(k;\nu),$$

$$\boldsymbol{y}(k;\nu+1) = \sigma_1\left(\boldsymbol{y}(k;\nu) + \eta_{\boldsymbol{y}}(\nu)\nabla_{\boldsymbol{y}(k)} \mathcal{J}(\boldsymbol{y}(k;\nu))\right),$$

where $\boldsymbol{B}_{\boldsymbol{y}}^{\zeta_{\boldsymbol{y}}}(k)$ and $\boldsymbol{B}_{\boldsymbol{e}}^{\zeta_{\boldsymbol{e}}}(k)$ are inverses of $\boldsymbol{R}_{\boldsymbol{y}}^{\zeta_{\boldsymbol{y}}}(k-1)$ and $\boldsymbol{R}_{\boldsymbol{e}}^{\zeta_{\boldsymbol{e}}}(k-1)$ respectively, $\nu \in \mathbb{N}$ is the index of neural dynamic iterations, $\sigma_1(.)$ is the projection onto the selected domain $\mathcal{B}_{\ell_\infty}$, which is the elementwise clipping function defined as $\sigma_1(\boldsymbol{y})_i = \begin{cases} y_i & -1 \leq y_i \leq 1, \\ \text{sign}(y_i) & \text{otherwise.} \end{cases}$.

The corresponding realization of the neural network is shown in Figure 4.

### C.2  DESCRIPTION OF THE NETWORK DYNAMICS FOR NONNEGATIVE ANTISPARSE SOURCES

We consider the source domain $\mathcal{P} = \mathcal{B}_{\ell_\infty,+}$. The treatment for this case is almost the same as the signed antisparse case in Appendix C.1. The only difference is that the projection to the source domain is performed by applying elementwise nonnegative clipping function to individual outputs.

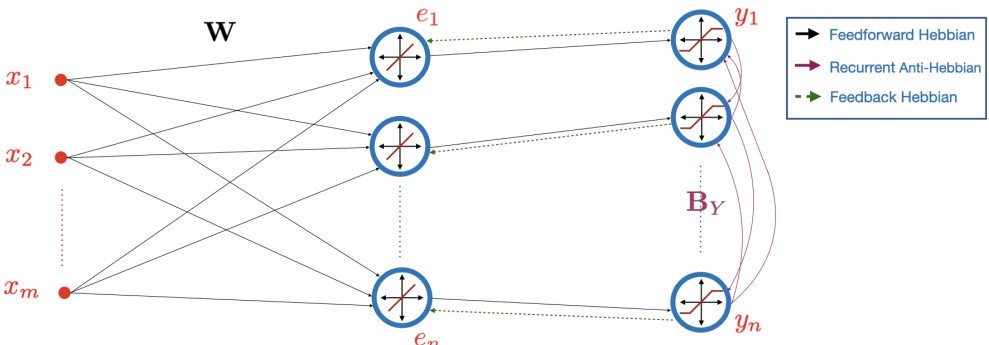

Figure 4: Two-Layer antisparse CorInfoMax Network. $x_i$'s and $y_i$'s represent inputs (mixtures) and (separator) outputs , respectively, $\mathbf{W}$ represents feedforward weights, $e_i$'s are errors between transformed inputs and outputs, $\mathbf{B_Y}$, the inverse of output autocorrelation matrix, represents lateral weights at the output. The output nonlinearities are clipping functions.

Therefore, we can write the neural network dynamics for the nonnegative antisparse case as

$$\boldsymbol{e}(k;\nu) = \boldsymbol{y}(k;\nu) - \boldsymbol{W}(k)\boldsymbol{x}(k),$$

$$\nabla_{\boldsymbol{y}(k)}\mathcal{J}(\boldsymbol{y}(k;\nu)) = \gamma_{\boldsymbol{y}}\boldsymbol{B}_{\boldsymbol{y}}^{\zeta_{\boldsymbol{y}}}(k)\boldsymbol{y}(k;\nu) - \gamma_{\boldsymbol{e}}\boldsymbol{B}_{\boldsymbol{e}}^{\zeta_{\boldsymbol{e}}}(k)\boldsymbol{e}(k;\nu),$$

$$\boldsymbol{y}(k;\nu+1) = \sigma_+\left(\boldsymbol{y}(k;\nu) + \eta_{\boldsymbol{y}}(\nu)\nabla_{\boldsymbol{y}(k)}\mathcal{J}(\boldsymbol{y}(k;\nu))\right),$$

where the nonnegative clipping function is defined as

$$\sigma_+(\boldsymbol{y})_i = \left\{ \begin{array}{cc} 0 & y_i \leq 0, \\ y_i & 0 \leq y_i \leq 1, \\ 1 & y_i \geq 1. \end{array} \right.$$

The corresponding neural network realization is shown in Figure 5.

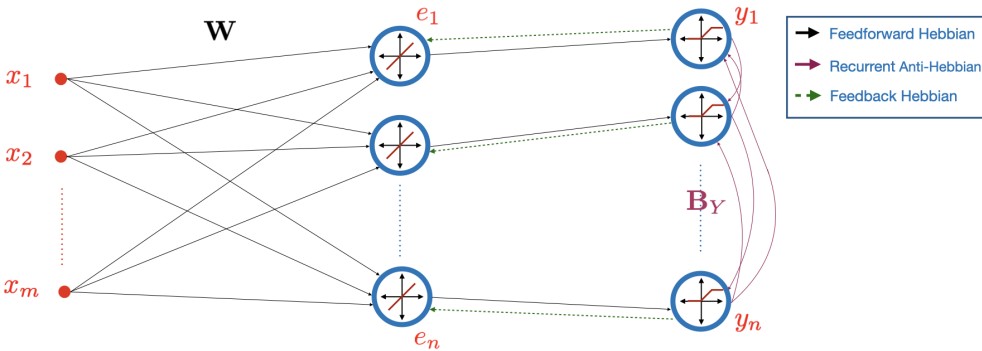

Figure 5: Two-Layer nonnegative antisparse CorInfoMax Network. $x_i$'s and $y_i$'s represent inputs (mixtures) and (separator) outputs , respectively, $\mathbf{W}$ represents feedforward weights, $e_i$'s are errors between transformed inputs and outputs, $\mathbf{B_Y}$, the inverse of output autocorrelation matrix, represents lateral weights at the output. The output nonlinearities are nonnegative clipping functions.

### C.3 Description of the Network Dynamics for Nonnegative Sparse Sources

For the nonnegative sparse CorInfoMax network in Section 3.2, the only change compared to its sparse counterpart is the replacement of the soft-thresholding activation functions at the output layer with the rectified linear unit. Accordingly, we can state the dynamics of output and inhibitory neurons as

$$\nabla_{\boldsymbol{y}(k)} \mathcal{J}(\boldsymbol{y}(k;\nu)) = \gamma_{\boldsymbol{y}} \boldsymbol{B}_{\boldsymbol{y}}^{\zeta_y}(k-1)\boldsymbol{y}(k;\nu) - \gamma_{\boldsymbol{e}} \boldsymbol{B}_{\boldsymbol{e}}^{\zeta_e}(k-1)\boldsymbol{e}(k;\nu),$$

$$\boldsymbol{y}(k;\nu+1) = \text{ReLU}\left(\boldsymbol{y}(k;\nu) + \eta_{\boldsymbol{y}}(\nu)\nabla_{\boldsymbol{y}(k)}\mathcal{J}(\boldsymbol{y}(k;\nu)) - \lambda(k;\nu)\right),$$

$$\nabla_{\lambda(k)}\mathcal{L}(\boldsymbol{y}(k;\nu)) = 1 - \left(\sum_{i=1}^{n} y_i(k;\nu+1)\right),$$

$$\lambda(k;\nu+1) = \text{ReLU}\left(\lambda(k;\nu) - \eta_\lambda(\nu)\nabla_{\lambda(k)}\mathcal{L}(\boldsymbol{y}(k;\nu))\right).$$

The corresponding network structure is illustrated in Figure 6.

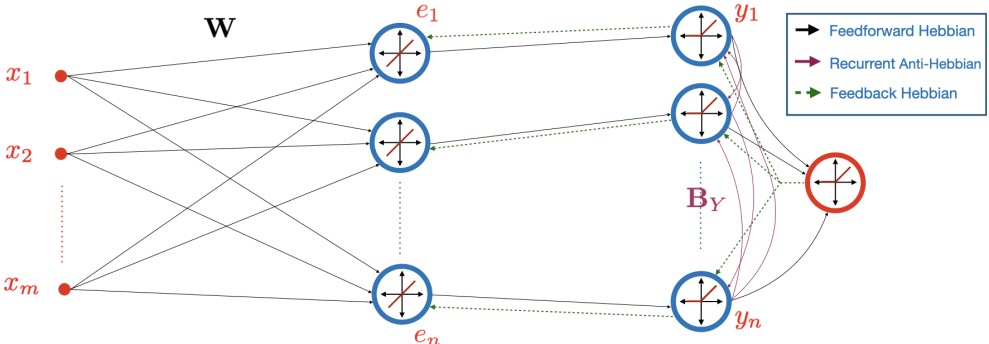

Figure 6: Three-Layer nonnegative sparse CorInfoMax Network. $x_i$'s and $y_i$'s represent inputs (mixtures) and (separator) outputs , respectively, $\mathbf{W}$ represents feedforward weights, $e_i$'s are errors between transformed inputs and outputs, $\mathbf{B_Y}$, the inverse of output autocorrelation matrix, represents lateral weights at the output. The output nonlinearities are ReLU functions. The leftmost interneuron imposes sparsity constraints on the outputs through inhibition.

## C.4  DESCRIPTION OF THE NETWORK DYNAMICS FOR UNIT SIMPLEX SOURCES

For the BSS setting for the unit simplex set $\Delta$, which is used for nonnegative matrix factorization, we consider the following optimization problem,

$$\underset{\boldsymbol{y}(k) \in \mathbb{R}^n}{\text{maximize}} \quad \mathcal{J}(\boldsymbol{y}(k)) \tag{A.15a}$$

$$\text{subject to} \quad ||\boldsymbol{y}(k)||_1 = 1, \quad \boldsymbol{y}(k) \succcurlyeq 0 \tag{A.15b}$$

for which the Lagrangian-based Min-Max problem can be stated as

$$\underset{\lambda}{\text{minimize}} \ \underset{\boldsymbol{y}(k) \in \mathbb{R}^n}{\text{maximize}} \quad \overbrace{\mathcal{J}(\boldsymbol{y}(k)) - \lambda(k)(||\boldsymbol{y}(k)||_1 - 1)}^{\mathcal{L}(\boldsymbol{y}(k),\lambda(k))}.$$

In this context, contrary to the optimization problem defined in (7), we do not require that the Lagrangian variable $\lambda$ be nonnegative, due to the equality constraint in (A.15b). Hence, we write the network dynamics for a simplex source as

$$\nabla_{\boldsymbol{y}(k)} \mathcal{J}(\boldsymbol{y}(k;\nu)) = \gamma_{\boldsymbol{y}} \boldsymbol{B}_{\boldsymbol{y}}^{\zeta_y}(k-1)\boldsymbol{y}(k;\nu) - \gamma_{\boldsymbol{e}} \boldsymbol{B}_{\boldsymbol{e}}^{\zeta_e}(k-1)\boldsymbol{e}(k;\nu),$$

$$\boldsymbol{y}(k;\nu+1) = \text{ReLU}\left(\boldsymbol{y}(k;\nu) + \eta_{\boldsymbol{y}}(\nu)\nabla_{\boldsymbol{y}(k)}\mathcal{J}(\boldsymbol{y}(k;\nu)) - \lambda(k;\nu)\right),$$

$$\nabla_{\lambda(k)}\mathcal{L}(\boldsymbol{y}(k;\nu)) = 1 - \left(\sum_{i=1}^{n} y_i(k;\nu+1)\right),$$

$$\lambda(k;\nu+1) = \lambda(k;\nu) - \eta_\lambda(\nu)\nabla_{\lambda(k)}\mathcal{L}(\boldsymbol{y}(k;\nu)).$$

Figure 7 demonstrates the network structure for simplex sources, which is identical to nonnegative sparse CorInfoMax network except that the linear activation replaces the ReLU activation of the inhibitory neuron.

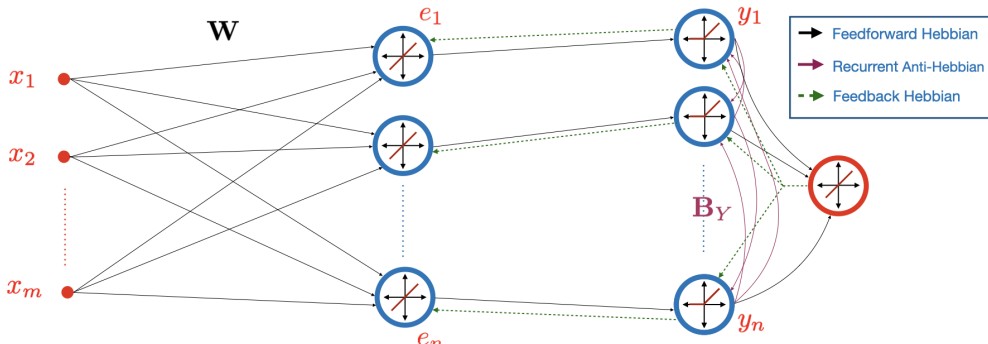

Figure 7: Three-Layer CorInfoMax Network for unit simplex sources. $x_i$'s and $y_i$'s represent inputs (mixtures) and (separator) outputs , respectively, $\mathbf{W}$ represents feedforward weights, $e_i$'s are errors between transformed inputs and outputs, $\mathbf{B_Y}$, the inverse of output autocorrelation matrix, represents lateral weights at the output. The output nonlinearities are ReLU functions. The leftmost interneuron imposes sparsity constraints on the outputs through inhibition.

## C.5  DESCRIPTION OF THE NETWORK DYNAMICS FOR FEATURE BASED SPECIFIED POLYTOPES

In this section, we consider the source separation setting where source samples are from a polytope represented in the form of (2). We expand the derivation in Bozkurt et al. (2022) (see Appendix D.6 in the reference) to obtain a neural network solution to the BSS problem for any identifiable polytope that can be expressed in the form of (2). Accordingly, we consider the following optimization problem

$$\underset{\boldsymbol{y}(k) \in \mathbb{R}^n}{\text{maximize}} \qquad \mathcal{J}(\boldsymbol{y}(k)) \tag{A.16a}$$

$$\text{subject to} \qquad -\mathbf{1} \preccurlyeq \boldsymbol{y}(k)_{\mathcal{I}_s} \preccurlyeq \mathbf{1}, \tag{A.16b}$$

$$\mathbf{0} \preccurlyeq \boldsymbol{y}(k)_{\mathcal{I}_+} \preccurlyeq \mathbf{1}, \tag{A.16c}$$

$$\|\boldsymbol{y}(k)_{\mathcal{J}_l}\|_1 \leq 1 \quad \forall l = 1, \dots, L \tag{A.16d}$$

We write the online optimization setting in a Lagrangian Min-Max setting as follows:

$$\underset{\lambda_l(k) \geq 0}{\text{minimize}} \quad \underset{\substack{\boldsymbol{y}(k) \in \mathbb{R}^n \\ -\mathbf{1} \preccurlyeq \boldsymbol{y}(k)_{\mathcal{I}_s} \preccurlyeq \mathbf{1} \\ \mathbf{0} \preccurlyeq \boldsymbol{y}(k)_{\mathcal{I}_+} \preccurlyeq \mathbf{1}}}{\text{maximize}} \quad \overbrace{\mathcal{J}(\boldsymbol{y}(k)) - \sum_{l=1}^{L} \lambda_l(k)(\|\boldsymbol{y}(k)_{\mathcal{J}_l}\|_1 - 1)}^{\mathcal{L}(\boldsymbol{y}(k), \lambda_1(k), \dots, \lambda_L(k))}.$$

The proximal operator corresponding to the Lagrangian term can be written as

$$\text{prox}_\lambda(\boldsymbol{y}) = \underset{\boldsymbol{q} \text{ s.t. } \boldsymbol{q}_{\mathcal{I}_+} \succcurlyeq \mathbf{0}}{\arg\min} \left( \frac{1}{2}\|\boldsymbol{y} - \boldsymbol{q}\|_2^2 + \sum_{l=1}^{L} \lambda_l \|\boldsymbol{q}_{\mathcal{J}_l}\|_1 \right). \tag{A.17}$$

Let $\boldsymbol{q}^*$ be the output of the proximal operator defined in (A.17). From the first order optimality condition,

- If $j \notin \mathcal{I}_+$, then $q_j^* - y_j + \sum\limits_{\substack{l \in \mathcal{J}_l \\ \text{s.t. } j \in \mathcal{J}_l}} \lambda_l sign(y_j) = 0$. Therefore, $q_j^* = y_j - \sum\limits_{\substack{l \in \mathcal{J}_l \\ \text{s.t. } j \in \mathcal{J}_l}} \lambda_l sign(y_j)$.

- If $j \in \mathcal{I}_+$, then $q_j^* = y_j - \sum\limits_{\substack{l \in \mathcal{J}_l \\ \text{s.t. } j \in \mathcal{J}_l}} \lambda_l$.

As a result, defining $\mathcal{I}_a = (\cap_l \mathcal{J}_l)^{\complement}$ as the set of dimension indices which do not appear in the sparsity constraints, we can write the corresponding output dynamics as

$$\nabla_{\boldsymbol{y}(k)} \mathcal{J}(\boldsymbol{y}(k; \nu)) = \gamma_{\boldsymbol{y}} \boldsymbol{B}_{\boldsymbol{y}}^{\zeta_y}(k-1) \boldsymbol{y}(k; \nu) - \gamma_{\boldsymbol{e}} \boldsymbol{B}_{\boldsymbol{e}}^{\zeta_e}(k-1) \boldsymbol{e}(k; \nu),$$
$$\bar{\boldsymbol{y}}(k; \nu+1) = \boldsymbol{y}(k; \nu) + \eta_{\boldsymbol{y}}(\nu) \nabla_{\boldsymbol{y}(k)} \mathcal{J}(\boldsymbol{y}(k; \nu)) \tag{A.18}$$
$$y_j(k; \nu+1) = ST_{\alpha_j(k,\nu)}(\bar{y}_j(k; \nu+1)) \text{ where } \alpha_j(k; \nu) = \sum_{\substack{l \in \mathcal{J}_l \\ \text{s.t. } j \in \mathcal{J}_l}} \lambda_l(k; \nu) \quad \forall j \in \mathcal{I}_s \cap \mathcal{I}_a^{\complement}$$
$$y_j(k; \nu+1) = \text{ReLU}\left(\bar{y}_j(k; \nu+1) - \sum_{\substack{l \in \mathcal{J}_l \\ \text{s.t. } j \in \mathcal{J}_l}} \lambda_l(k; \nu)\right) \quad \forall j \in \mathcal{I}_+ \cap \mathcal{I}_a^{\complement}$$
$$y_j(k; \nu+1) = \sigma_1(\bar{y}_j(k; \nu)) \quad \forall j \in \mathcal{I}_s \cap \mathcal{I}_a,$$
$$y_j(k; \nu+1) = \sigma_+(\bar{y}_j(k; \nu)) \quad \forall j \in \mathcal{I}_+ \cap \mathcal{I}_a$$

For inhibitory neurons corresponding to Lagrangian variables $\lambda_1, \ldots, \lambda_L$, we obtain the update dynamics based on the derivative of $\mathcal{L}(\boldsymbol{y}(k; \nu), \lambda_1(k; \nu), \ldots, \lambda_L(k; \nu))$ as

$$\frac{d\mathcal{L}(\boldsymbol{y}(k), \lambda_1(k), \ldots, \lambda_L(k))}{d\lambda_l(k)}\bigg|_{\lambda_l(k;\nu)} = 1 - \|[\boldsymbol{y}(k; \nu+1)]_{\mathcal{J}_l}\|_1 \quad \forall l,$$
$$\bar{\lambda}_l(k; \nu+1) = \lambda_l(k; \nu)$$
$$- \eta_{\lambda_l}(\nu) \frac{d\mathcal{L}(\boldsymbol{y}(k), \lambda_1(k), \ldots, \lambda_L(k))}{d\lambda_l(k)}\bigg|_{\lambda_l(k;\nu)},$$
$$\lambda_l(k, \nu+1) = \text{ReLU}\left(\bar{\lambda}_l(k; \nu+1)\right).$$

In Appendix D.2.4, we demonstrate an example setting in which the underlying domain is defined as

$$\mathcal{P}_{ex} = \left\{ \boldsymbol{s} \in \mathbb{R}^5 \;\middle|\; \begin{array}{l} s_1, s_2, s_4 \in [-1, 1], s_3, s_5 \in [0, 1], \\ \left\| \begin{bmatrix} s_1 \\ s_2 \\ s_5 \end{bmatrix} \right\|_1 \le 1, \left\| \begin{bmatrix} s_2 \\ s_3 \\ s_4 \end{bmatrix} \right\|_1 \le 1 \end{array} \right\}.$$

We summarize the neural dynamics for this specific example:

$$y_1(k; \nu+1) = ST_{\lambda_1(k,\nu)}(\bar{y}_1(k; \nu+1)),$$
$$y_2(k; \nu+1) = ST_{\lambda_1(k,\nu)+\lambda_2(k,\nu)}(\bar{y}_2(k; \nu+1)),$$
$$y_3(k; \nu+1) = \text{ReLU}(\bar{y}_3(k; \nu+1) - \lambda_2(k; \nu)),$$
$$y_4(k; \nu+1) = ST_{\lambda_2(k,\nu)}(\bar{y}_4(k; \nu+1)),$$
$$y_5(k; \nu+1) = \text{ReLU}(\bar{y}_5(k; \nu+1) - \lambda_1(k; \nu)),$$
$$\lambda_1(k; \nu+1) = \lambda_1(k; \nu) - \eta_{\lambda_1}(\nu)(1 - |y_1(k; \nu+1)| - |y_2(k; \nu+1)| - y_5(k; \nu+1)),$$
$$\lambda_2(k; \nu+1) = \lambda_2(k; \nu) - \eta_{\lambda_1}(\nu)(1 - |y_2(k; \nu+1)| - y_3(k; \nu+1) - |y_4(k; \nu+1)|),$$

where $\bar{\boldsymbol{y}}(k; \nu)$ is defined as in (A.18).

# D SUPPLEMENTARY ON NUMERICAL EXPERIMENTS

In this section, we provide more details on the algorithmic view of the proposed approach and the numerical experiments presented. In addition, we provide more examples.

### D.1 ONLINE CORINFOMAX ALGORITHM IMPLEMENTATIONS FOR SPECIAL SOURCE DOMAINS

Algorithm 2 summarizes the dynamics of the CorInfoMax network and learning rules. For each of the domain choices, the recurrent and feedforward weight updates follow (14) and (15), respectively, and the learning step is indicated in the $4^{\text{th}}$ and $5^{\text{th}}$ lines in the pseudo-code. The line $3^{\text{rd}}$ expresses the recursive neural dynamics to obtain the output of the network, and its implementation differs for different domain choices. Based on the derivations in Section 3 and Appendix C, Algorithm 3, 4, 5, and 6 summarizes the neural dynamic iterations for some example domains. For example, Algorithm 5 indicates the procedure to obtain the output $\boldsymbol{y}(k)$ of the antisparse CorInfoMax network at time step $k$ corresponding to the mixture vector $\boldsymbol{x}(k)$. As it is an optimization process, we introduce two variables for implementation in digital hardware: 1) numerical convergence tolerance $\epsilon_t$, and 2) maximum number of (neural dynamic) iterations $\nu_{\max}$. We run the proposed neural dynamic iterations until either a convergence happens, i.e., $\|\boldsymbol{y}(k;\nu) - \boldsymbol{y}(k;\nu - 1)\|/\|\boldsymbol{y}(k;\nu)\| > \epsilon_t$, or the loop counter reaches a predetermined maximum number of iterations, that is, $\nu = \nu_{\max}$. Differently from the antisparse and nonnegative antisparse networks, the other CorInfoMax neural networks include additional inhibitory neurons due to the Lagrangian Min-Max settings, and the activation of these neurons are coupled with the network's output. Therefore, inhibitory neurons are updated in neural dynamics based on the gradient of the Lagrangian objective.

---

**Algorithm 2** Online CorInfoMax pseudo-code

---

**Input**: A streaming data of $\{\boldsymbol{x}(k) \in \mathbb{R}^m\}_{k=1}^N$.
**Output**: $\{\boldsymbol{y}(k) \in \mathbb{R}^n\}_{k=1}^N$.

1: Initialize $\zeta_{\boldsymbol{y}}, \zeta_{\boldsymbol{e}}, \mu(1)_{\boldsymbol{W}}, \boldsymbol{W}(1), \boldsymbol{B}_{\boldsymbol{y}}^{\zeta_{\boldsymbol{y}}}(1), \boldsymbol{B}_{\boldsymbol{e}}^{\zeta_e}(1)$, and select $\mathcal{P}$.
2: **for** k = 1, 2, ..., N **do**
3:    **run neural dynamics (in Algorithms 3 to 5 below according to $\mathcal{P}$) until convergence** .
4:    $\boldsymbol{W}(k+1) = \boldsymbol{W}(k) + \mu_{\boldsymbol{W}}(k)\boldsymbol{e}(k)\boldsymbol{x}(k)^T$
5:    $\boldsymbol{B}_{\boldsymbol{y}}^{\zeta_{\boldsymbol{y}}}(k+1) = \dfrac{1}{\zeta_{\boldsymbol{y}}}(\boldsymbol{B}_{\boldsymbol{y}}^{\zeta_{\boldsymbol{y}}}(k) - \dfrac{1-\zeta_{\boldsymbol{y}}}{\zeta_{\boldsymbol{y}}}\boldsymbol{B}_{\boldsymbol{y}}^{\zeta_{\boldsymbol{y}}}(k)\boldsymbol{y}(k)\boldsymbol{y}(k)^T\boldsymbol{B}_{\boldsymbol{y}}^{\zeta_{\boldsymbol{y}}}(k))$
6:    Adjust $\mu_{\boldsymbol{W}}(k+1)$ if necessary.
7: **end for**

---

---

**Algorithm 3** Online CorInfoMax neural dynamic iterations: sources in unit simplex

---

1: Initialize $\nu_{\max}, \epsilon_t, \eta_{\boldsymbol{y}}(1), \eta_{\lambda}(1), \lambda(1)(= 0$ in general$)$ and $\nu = 1$
2: **while** $(\|\boldsymbol{y}(k;\nu) - \boldsymbol{y}(k;\nu - 1)\|/\|\boldsymbol{y}(k;\nu)\| > \epsilon_t)$ and $\nu < \nu_{\max}$ **do**
3:    $\nabla_{\boldsymbol{y}(k)}\mathcal{J}(\boldsymbol{y}(k;\nu)) = \gamma_{\boldsymbol{y}}\boldsymbol{B}_{\boldsymbol{y}}^{\zeta_{\boldsymbol{y}}}(k)\boldsymbol{y}(k;\nu) - \gamma_{\boldsymbol{e}}\boldsymbol{B}_{\boldsymbol{e}}^{\zeta_e}(k)\boldsymbol{e}(k;\nu)$
4:    $\boldsymbol{y}(k;\nu+1) = \text{ReLU}\left(\boldsymbol{y}(k;\nu) + \eta_{\boldsymbol{y}}(\nu)\nabla_{\boldsymbol{y}(k)}\mathcal{J}(\boldsymbol{y}(k;\nu)) - \lambda(\nu)\right)$
5:    $\nabla_{\lambda(k)}\mathcal{L}(\boldsymbol{y}(k;\nu)) = 1 - \|\boldsymbol{y}(k;\nu+1)\|_1$
6:    $\lambda(k;\nu+1) = \lambda(k;\nu) - \eta_{\lambda}(\nu)\nabla_{\lambda(k)}\mathcal{L}(\boldsymbol{y}(k;\nu))$
7:    $\nu = \nu + 1$, and adjust $\eta_{\boldsymbol{y}}(\nu), \eta_{\lambda}(\nu)$ if necessary.
8: **end while**

---

---

**Algorithm 4** Online CorInfoMax neural dynamic iterations: sparse sources

---

1: Initialize $\nu_{\max}, \epsilon_t, \eta_{\boldsymbol{y}}(1), \eta_{\lambda}(1), \lambda(1)(= 0$ in general$)$ and $\nu = 1$
2: **while** $(\|\boldsymbol{y}(k;\nu) - \boldsymbol{y}(k;\nu - 1)\|/\|\boldsymbol{y}(k;\nu)\| > \epsilon_t)$ and $\nu < \nu_{\max}$ **do**
3:    $\nabla_{\boldsymbol{y}(k)}\mathcal{J}(\boldsymbol{y}(k;\nu)) = \gamma_{\boldsymbol{y}}\boldsymbol{B}_{\boldsymbol{y}}^{\zeta_{\boldsymbol{y}}}(k)\boldsymbol{y}(k;\nu) - \gamma_{\boldsymbol{e}}\boldsymbol{B}_{\boldsymbol{e}}^{\zeta_e}(k)\boldsymbol{e}(k;\nu)$
4:    $\boldsymbol{y}(k;\nu+1) = ST_{\lambda(\nu)}\left(\boldsymbol{y}(k;\nu) + \eta_{\boldsymbol{y}}(\nu)\nabla_{\boldsymbol{y}(k)}\mathcal{J}(\boldsymbol{y}(k;\nu)) - \lambda(\nu)\right)$
5:    $\nabla_{\lambda(k)}\mathcal{L}(\boldsymbol{y}(k;\nu)) = 1 - \|\boldsymbol{y}(k;\nu+1)\|_1$
6:    $\lambda(k;\nu+1) = \text{ReLU}\left(\lambda(k;\nu) - \eta_{\lambda}(\nu)\nabla_{\lambda(k)}\mathcal{L}(\boldsymbol{y}(k;\nu))\right)$
7:    $\nu = \nu + 1$, and adjust $\eta_{\boldsymbol{y}}(\nu), \eta_{\lambda}(\nu)$ if necessary.
8: **end while**

---

---

**Algorithm 5** Online CorInfoMax neural dynamic iterations: antisparse sources

---

1: Initialize $\nu_{\max}$, $\epsilon_t$, $\eta_{\boldsymbol{y}}(1)$ and $\nu = 1$
2: **while** $(\|\boldsymbol{y}(k;\nu) - \boldsymbol{y}(k;\nu-1)\|/\|\boldsymbol{y}(k;\nu)\| > \epsilon_t)$ and $\nu < \nu_{\max}$ **do**
3:     $\nabla_{\boldsymbol{y}(k)}\mathcal{J}(\boldsymbol{y}(k;\nu)) = \gamma_{\boldsymbol{y}}\boldsymbol{B}_{\boldsymbol{y}}^{\zeta_{\boldsymbol{y}}}(k)\boldsymbol{y}(k;\nu) - \gamma_{\boldsymbol{e}}\boldsymbol{B}_{\boldsymbol{e}}^{\zeta_e}(k)\boldsymbol{e}(k;\nu)$
4:     $\boldsymbol{y}(k;\nu+1) = \sigma_1\left(\boldsymbol{y}(k;\nu) + \eta_{\boldsymbol{y}}(\nu)\nabla_{\boldsymbol{y}(k)}\mathcal{J}(\boldsymbol{y}(k;\nu))\right)$
5:     $\nu = \nu + 1$, and adjust $\eta_{\boldsymbol{y}}(\nu)$ if necessary.
6: **end while**

---

**Algorithm 6** Online CorInfoMax neural dynamic iterations for Canonical Form

---

1: Initialize $\nu_{\max}$, $\epsilon_t$, $\eta_{\boldsymbol{y}}(1)$, $\eta_{\boldsymbol{\lambda}}(1)$, $\boldsymbol{\lambda}(1)(=\boldsymbol{0}$ in general$)$ and $\nu = 1$
2: **while** $(\|\boldsymbol{y}(k;\nu) - \boldsymbol{y}(k;\nu-1)\|/\|\boldsymbol{y}(k;\nu)\| > \epsilon_t)$ and $\nu < \nu_{\max}$ **do**
3:     $\nabla_{\boldsymbol{y}(k)}\mathcal{J}(\boldsymbol{y}(k;\nu)) = \gamma_{\boldsymbol{y}}\boldsymbol{B}_{\boldsymbol{y}}^{\zeta_{\boldsymbol{y}}}(k)\boldsymbol{y}(k;\nu) - \gamma_{\boldsymbol{e}}\boldsymbol{B}_{\boldsymbol{e}}^{\zeta_e}(k)\boldsymbol{e}(k;\nu) - \boldsymbol{A}_{\mathcal{P}}^T\boldsymbol{\lambda}(\nu)$
4:     $\boldsymbol{y}(k;\nu+1) = \boldsymbol{y}(k;\nu) + \eta_{\boldsymbol{y}}(\nu)\nabla_{\boldsymbol{y}(k)}\mathcal{J}(\boldsymbol{y}(k;\nu))$
5:     $\nabla_{\boldsymbol{\lambda}(k)}\mathcal{L}(\boldsymbol{y}(k;\nu)) = -\boldsymbol{A}_{\mathcal{P}}\boldsymbol{y}(k;\nu) + \boldsymbol{b}_{\mathcal{P}}$
6:     $\boldsymbol{\lambda}(k;\nu+1) = \mathrm{ReLU}\left(\boldsymbol{\lambda}(k;\nu) - \eta_{\boldsymbol{\lambda}}(\nu)\nabla_{\boldsymbol{\lambda}(k)}\mathcal{L}(\boldsymbol{y}(k;\nu))\right)$
7:     $\nu = \nu + 1$, and adjust $\eta_{\boldsymbol{y}}(\nu), \eta_{\boldsymbol{\lambda}}(\nu)$ if necessary.
8: **end while**

---

## D.2 Additional Numerical Experiments for Special Source Domains

### D.2.1 Sparse Source Separation

In this section, we illustrate blind separation of sparse sources, i.e. $\boldsymbol{s}(i) \in \mathcal{B}_{\ell_1} \, \forall i$. We consider $n = 5$ sources and $m = 10$ mixtures. For each source, we generate $5 \times 10^5$ samples in each realization of the experiments. We examine two different experimental factors: 1) output SINR performance as a function of mixture SNR levels, and 2) output SINR performance for different distribution selections for the entries of the mixing matrix.

For the first scenario with different mixture SNR levels: the sources are mixed through a random matrix $\boldsymbol{A} \in \mathbb{R}^{10 \times 5}$ whose entries are drawn from i.i.d. standard normal distribution, and the mixtures are corrupted by WGN with 30dB SNR. We compare our approach with the WSM (Bozkurt et al., 2022), LD-InfoMax (Erdogan, 2022) and PMF (Tatli & Erdogan, 2021) algorithms and visualize the results in Figure 8. We also use the portion of the mixtures to train batch LD-InfoMax and PMF algorithms. Figure 8a illustrates the SINR performances of these algorithms for different input noise levels. The SINR results of CorInfoMax, LD-InfoMax, and PMF are noticeably close to each other, which is almost equal to the input SNR. Figure 8b illustrates the SINR convergence of the sparse CorInfoMax network for the 30dB mixture SNR level as a function of update iterations. Based on this figure, we can conclude that the proposed CorInfoMax network converges robustly and smoothly.

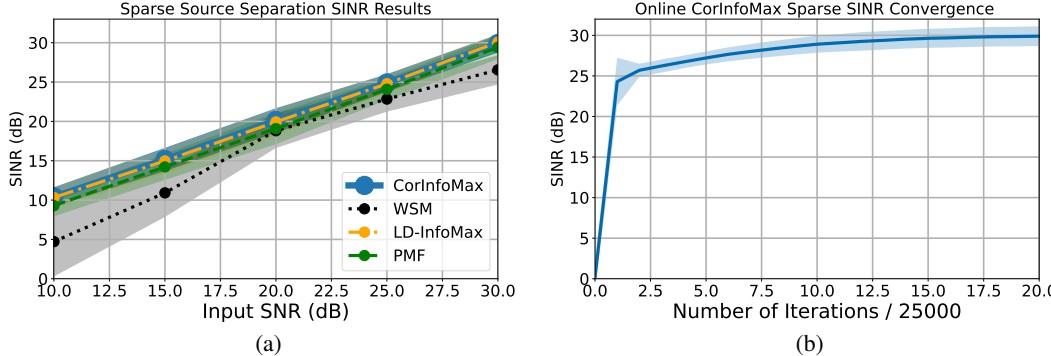

(a)                                                    (b)

Figure 8: SINR performances of CorInfoMax (ours), LD-InfoMax, PMF, and WSM (averaged over 50 realizations) for sparse sources: (a) output SINR results (vertical axis) with respect to input SNR levels (horizontal axis), (b) SINR (vertical axis) convergence plot as a function of iterations (horizontal axis) of sparse CorInfoMax for the 30 dB SNR level (mean solid line and standard deviation envelopes).

In the second experimental setting, we examine the effect of the distribution choice for generating the random mixing matrix. Figure 9 illustrates the box plots of SINR results for CorInfoMax, LD-InfoMax, and PMF for different distribution selections to generate the mixing matrix, which are $\mathcal{N}(0,1)$, $\mathcal{U}[-1,1]$, $\mathcal{U}[-2,2]$, $L(0,1)$. where $\mathcal{N}$ is normal distribution, $\mathcal{U}$ is uniform distribution, and $L$ is the Laplace distribution. It is observable that the performance of CorInfoMax is robust against the different distribution selections of the unknown mixing matrix, while its performance is on par with the batch algorithms LD-InfoMax and PMF.

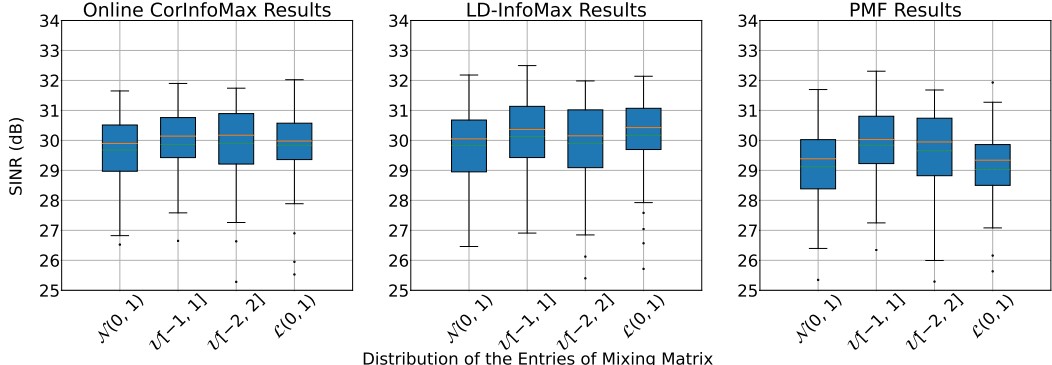

Figure 9: SINR performances of CorInfoMax (ours), LD-InfoMax, and PMF with different distribution selections for the mixing matrix entries and for sparse sources. The horizontal axis represents different distribution choices, and the vertical axis represents the SINR levels.

### D.2.2 NONNEGATIVE SPARSE SOURCE SEPARATION

We replicate the first experimental setup in Appendix D.2.1 for the nonnegative sparse source separation: evaluate the SINR performance of the nonnegative sparse CorInfoMax network for different levels of mixture SNR, compared to the batch LD-InfoMax algorithm and the biologically plausible WSM neural network. In these experiments, $n = 5$ uniform sources in $\mathcal{B}_{\ell_1,+}$ are randomly mixed to generate $m = 10$ mixtures. Figure 10a illustrates the averaged output SINR performances of each algorithm with a standard deviation envelope for different input noise levels. In Figure 10b, we observe the SINR convergence behavior of nonnegative sparse CorInfoMax as a function of update iterations. Note that CorInfoMax outperforms the biologically plausible neural network WSM for all input SNR levels, and its convergence is noticeably stable.

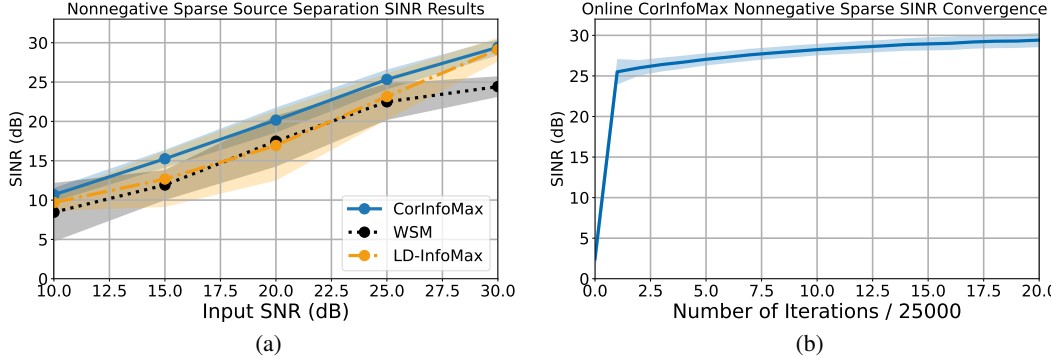

(a)                                                                 (b)

Figure 10: SINR performances of CorInfoMax (ours), LD-InfoMax, and WSM for nonnegative sparse sources: (a) the output SINR (vertical axis) results with respect to the input SNR levels (horizontal axis), (b) the SINR (vertical axis) convergence plot as a function of iterations (horizontal axis) of nonnegative sparse CorInfoMax for the 30dB SNR level (mean solid line and standard deviation envelopes).

### D.2.3    SIMPLEX SOURCE SEPARATION

We repeat both experimental settings in Appendix D.2.1 for the blind separation of simplex sources using the CorInfoMax network in Figure 7. Figure 11a shows the output SINR results of both the online CorInfoMax and the batch LD-InfoMax approaches for different mixture SNR levels. Even though simplex CorInfoMax is not as successful as the other examples (e.g., sparse CorInfoMax), in terms of the closeness of its performance to the batch algorithms, it still has satisfactory source separation capability. Similarly to the sparse network examples, its SINR convergence is fast and smooth, as illustrated in Figure 11b.

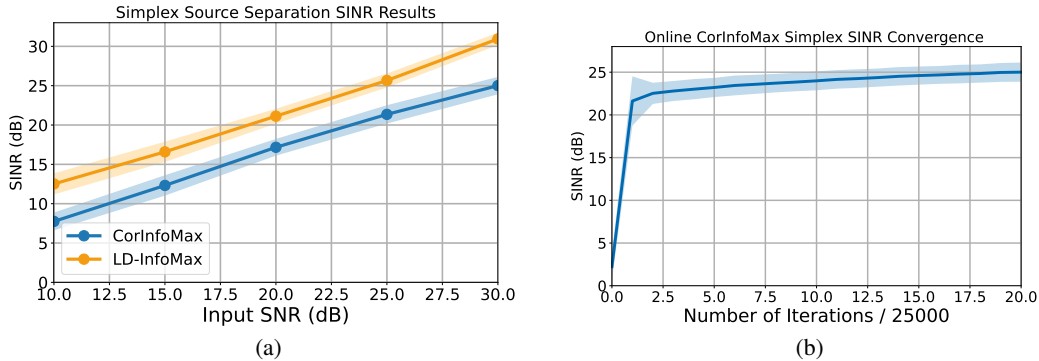

(a)                                                                 (b)

Figure 11: SINR performances of CorInfoMax (ours) and LD-InfoMax (averaged over 50 realizations) for unit simplex sources: (a) the output SINR (vertical axis) results with respect to the input SNR levels (horizontal axis), (b) SINR (vertical axis) convergence plot as a function of iterations (horizontal axis) of simplex-CorInfoMax for 30dB SNR level (mean solid line and standard deviation envelopes).

Figure 12 shows the box plots of the SINR performances for both CorInfoMax and LD-InfoMax approaches with respect to different distribution selections for generating the random mixing matrix. Based on this figure, we can conclude that the simplex CorInfoMax network significantly maintains its performance for different distributions for the entries of the mixing matrix.

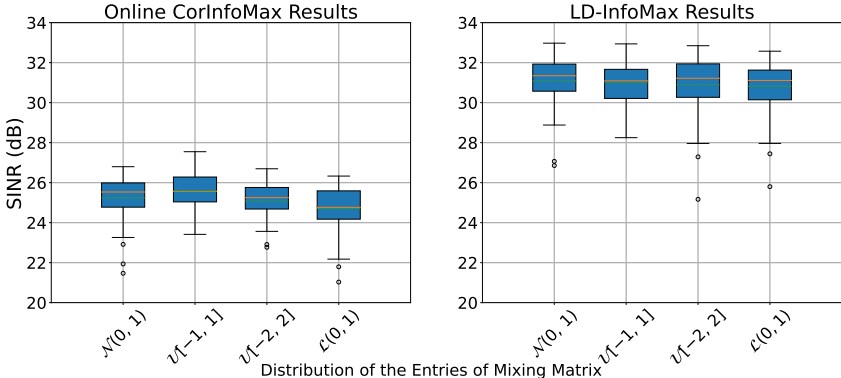

Figure 12: SINR performances of CorInfoMax (ours) and LD-InfoMax with different distribution selections for the mixing matrix entries (averaged over 50 realizations) and unit simplex sources. The horizontal axis represents different distribution choices, and the vertical axis represents the SINR levels.

### D.2.4 SOURCE SEPARATION FOR POLYTOPES WITH MIXED LATENT ATTRIBUTES

In this section, we demonstrate the source separation capability of CorInfoMax on an identifiable polytope with mixed features, which is a special case of *feature-based* polytopes in (2). We focus on the polytope

$$\mathcal{P}_{ex} = \left\{ \boldsymbol{s} \in \mathbb{R}^5 \ \middle| \ \begin{array}{l} s_1, s_2, s_4 \in [-1, 1], s_3, s_5 \in [0, 1], \\ \left\| \left[ \begin{array}{c} s_1 \\ s_2 \\ s_5 \end{array} \right] \right\|_1 \leq 1, \left\| \left[ \begin{array}{c} s_2 \\ s_3 \\ s_4 \end{array} \right] \right\|_1 \leq 1 \end{array} \right\}, \tag{A.19}$$

whose identifiability property is verified by the identifiable polytope characterization algorithm presented in Bozkurt & Erdogan (2022). We experiment with both approaches discussed in Section 3.3 and Appendix C.5. For the feature-based polytope setting introduced in Appendix C.5, the output dynamics corresponding to $\mathcal{P}_{ex}$ is also summarized as an example. This polytope can also be represented as the intersection of 10 half-spaces, that is, $\mathcal{P}_{ex} = \{ \boldsymbol{s} \in \mathbb{R}^n | \boldsymbol{A}_{\mathcal{P}} \boldsymbol{s} \preccurlyeq \boldsymbol{b}_{\mathcal{P}} \}$ where $\boldsymbol{A}_{\mathcal{P}} \in \mathbb{R}^{10 \times 5}$ and $\boldsymbol{b}_{\mathcal{P}} \in \mathbb{R}^{10}$. Therefore, using $\boldsymbol{A}_{\mathcal{P}}$ and $\boldsymbol{b}_{\mathcal{P}}$, we can also employ the neural network illustrated in Figure 1a with 10 inhibitory neurons.

For this BSS setting, we generated the sources uniformly within this 5 dimensional polytope $\mathcal{P}_{ex}$ where the sample size is $5 \times 10^5$. The source vectors are mixed through a random matrix $\boldsymbol{A} \in \mathbb{R}^{10 \times 5}$ with standard normal entries. Figure 13a and 13b show the SINR convergence of CorInfoMax networks based on the feature-based polytope representation in (2) and the H-representation in (1) respectively, for the mixture SNR level of 30dB. Moreover, Figure 13c and 13d illustrate their SINR convergence curves for the SNR level of 40dB. In Table 2, we compare both approaches with the batch algorithms LD-InfoMax and PMF.

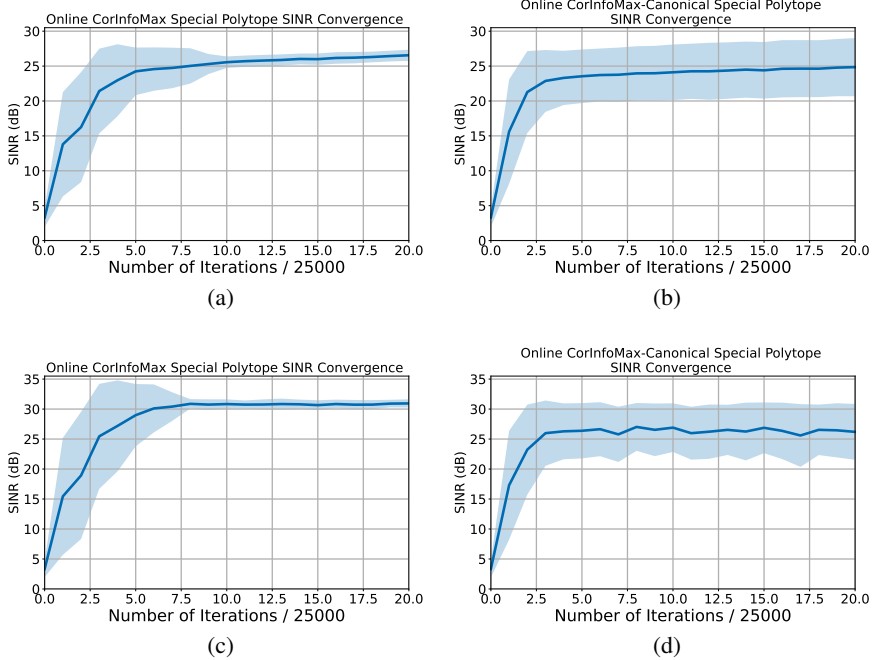

Figure 13: SINR performances of CorInfoMax networks on $\mathcal{P}_{\text{ex}}$ with mean-solid line with standard deviation envelope (averaged over 50 realization) for the polytope with mixed latent attributes: (a) SINR convergence curve for CorInfoMax feature-based polytope formulation with 30dB mixture SNR, (b) SINR convergence curve for CorInfoMax canonical formulation with 30dB mixture SNR, (c) SINR convergence curve for CorInfoMax feature-based polytope formulation with 40dB mixture SNR, (d) SINR convergence curve for CorInfoMax canonical formulation with 40dB SNR.

Table 2: Source separation averaged SINR results on $\mathcal{P}_{\text{ex}}$ for the CorInfoMax (ours), CorInfoMax Canonical (ours) LD-InfoMax, and PMF algorithms (averaged for 50 realizations).

| Algorithm | CorInfoMax | CorInfoMax Canonical | LD-InfoMax | PMF |
|---|---|---|---|---|
| SINR (\w 30dB SNR) | 26.55 | 24.85 | 30.28 | 27.68 |
| SINR (\w 40dB SNR) | 30.93 | 26.19 | 38.50 | 31.48 |

## D.3 APPLICATIONS

We present several potential applications of the proposed approach, for which we illustrate the usage of antisparse, nonnegative antisparse and sparse CorInfoMax neural networks. This section demonstrates sparse dictionary learning, source separation for digital communication signals with 4-PAM modulation scheme, and video separation.

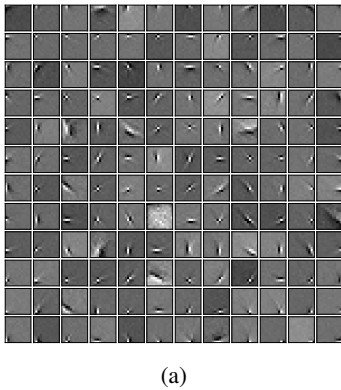
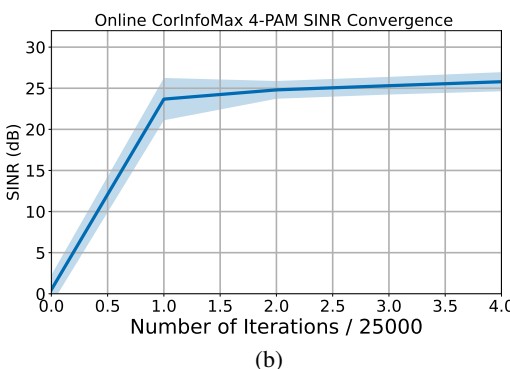

(a)                                                       (b)

Figure 14: (a) Sparse dictionary learned by sparse CorInfoMax from natural image patches, (b) The SINR (vertical axis) convergence curve for the 4-PAM digital communication example as a function of iterations (horizontal axis) which is averaged over 100 realizations: mean-solid line with standard deviation envelope.

### D.3.1 SPARSE DICTIONARY LEARNING

We consider sparse dictionary learning for natural images, to model receptive fields in the early stages of visual processing (Olshausen & Field, 1997). In this experiment, we used $12 \times 12$ pre-whitened image patches as input to the sparse CorInfoMax network. The image patches are obtained from the website http://www.rctn.org/bruno/sparsenet. The inputs are vectorized to the shape $144 \times 1$ before feeding to the neural network illustrated in Figure 2. Figure 14a illustrates the dictionary learned by the sparse CorInfoMax network.

### D.3.2 DIGITAL COMMUNICATION EXAMPLE: 4-PAM MODULATION SCHEME

One successful application of antisparse source modeling to solve BSS problems is digital communication systems Cruces (2010); Erdogan (2013). In this section, we verify that the antisparse CorInfoMax network in Figure 4 can separate 4-pulse-amplitude-modulation (4-PAM) signals with domain $\{-3, -1, 1, 3\}$ (with a uniform probability distribution). We consider that 5 digital communication (4-PAM) sources with $10^5$ samples are transmitted and then mixed through a Gaussian channel to produce 10 mixtures. The mixtures can represent signals received at some base station antennas in a multipath propagation environment. Furthermore, the mixtures are corrupted by WGN that corresponds to SNR level of 30dB. We feed the mixtures to the antisparse CorInfoMax network illustrated as input. For 100 different realization of the experimental setup, Figure 14b illustrates the SINR convergence as a function of update iterations. We note that the proposed approach distinguishably converges fast and each realization of the experiments resulted in a zero symbol error rate.

### D.3.3 VIDEO SEPARATION

We provide more details on the video separation experiment discussed in Section 4.2. Three source videos we used in this experiment are from the website https://www.pexels.com/, which are free to download and use. Videos are mixed linearly through a randomly selected nonnegative $5 \times 3$ matrix

$$\boldsymbol{A} = \begin{bmatrix} 1.198 & 1.020 & 1.273 \\ 0.367 & 1.364 & 0.901 \\ 0.100 & 0.869 & 0.627 \\ 1.257 & 0.859 & 0.015 \\ 0.957 & 0.789 & 0.592 \end{bmatrix},$$

to generate 5 mixture videos. Figures 15a and 15b illustrate the last RGB frames of the original and mixture videos, respectively. To train the nonnegative antisparse CorInfoMax network in Figure 5 for the separation of the videos, we followed the procedure below.

- In each iteration of the algorithm, we randomly choose a pixel location and select pixels from one of the color channels of all mixture videos to form a mixture vector of size $5 \times 1$,
- We sample 20 mixture vectors from each frame and perform 20 algorithm iterations per frame using these samples.

The demonstration video contains three rows of frames: the first row contains the source frames, the second row contains three of the five mixture frames, and the last row contains the network outputs for these mixture frames. Demo video is located at (https://figshare.com/s/a3fb926f273235068053). It is also included in supplementary files. If we use the separator matrix of the CorInfoMax network, which is an estimate for the left inverse of $A$, to predict the original videos after training, we obtain PSNR values of $35.60$dB, $48.07$dB, and $44.58$dB for the videos, which are calculated as the average PSNR levels of the frames. Figure 15c illustrates the final output frames of the CorInfoMax.

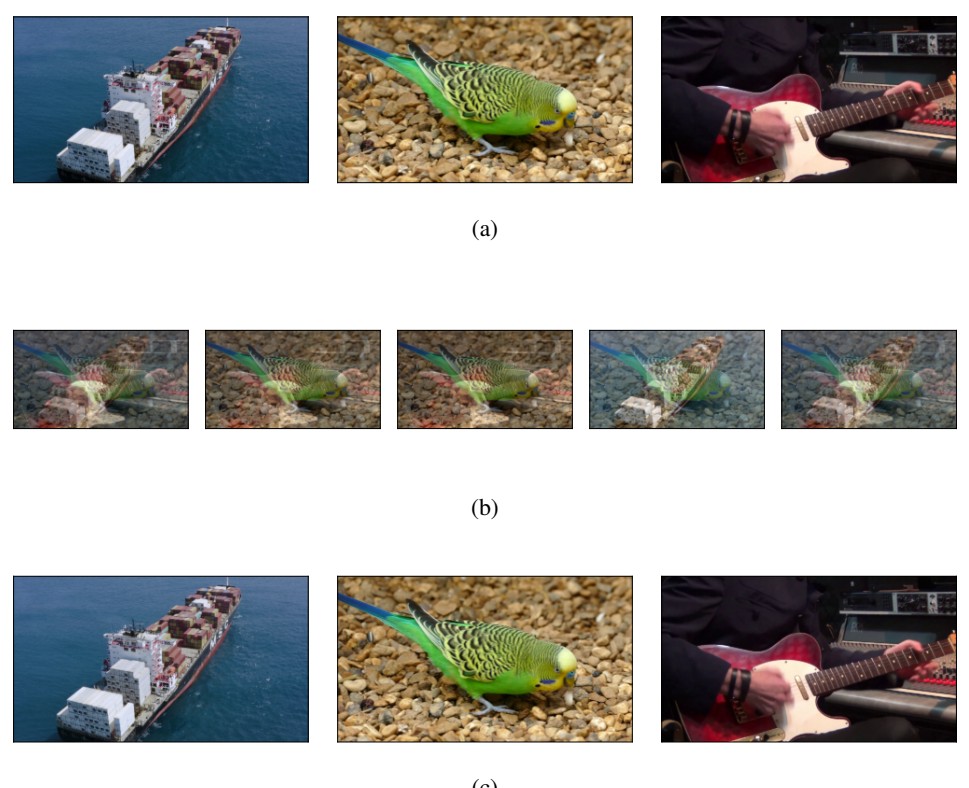

(a)

(b)

(c)

Figure 15: Video separation example: the final frames of (a) sources, (b) mixtures, and (c) the outputs of the nonnegative antisparse CorInfoMax network.

### D.3.4 IMAGE SEPARATION

Related to the example in the previous section, we consider an example on the blind separation of correlated images from their linear mixtures. In this experiment, we consider 3 RGB natural scenes with sizes $454 \times 605$ that are illustrated in Figure 16a. The Pearson correlation coefficient for these sources are $\rho_{12} = 0.076, \rho_{13} = 0.262, \rho_{23} = 0.240$, respectively. The image sources are mixed through a random matrix $A \in \mathbb{R}^{5 \times 3}$ whose entries are drawn from i.i.d. standard normal distribution. Moreover, the mixtures are corrupted with WGN corresponding to 40dB SNR. The

mixture images are demonstrated in Figure 16b, and the mixing matrix for this particular example is

$$\boldsymbol{A} = \begin{bmatrix} -0.363 & 0.650 & 1.757 \\ 1.100 & 1.568 & 1.487 \\ -1.266 & 0.032 & -0.417 \\ -0.822 & 0.643 & 1.260 \\ -0.023 & -0.752 & 0.661 \end{bmatrix}.$$

We experiment with the proposed antisparse CorInfomax and biologically plausible NSM and WSM networks, and batch ICA-InfoMax and LD-InfoMax frameworks to separate the original sources, and Figures 16c-16g show the corresponding outputs. We note that the residual interference effects in the output images of ICA-InfoMax algorithm is remarkably perceivable, and the resulting PSNR values are 18.56dB, 20.52dB, and 21.06dB, respectively. The NSM algorithm's outputs are visually better whereas some interference effects are still noticable, and the resulting PSNR values are 25.30dB, 26.49dB, 26.45dB. The visual interference effects in the WSM algorithm's outputs are barely visible, and, therefore, they achieve higher PSNR values of 27.99dB, 29.71dB, and 31.92dB. The batch LD-InfoMax algorithm achieves the best PSNR performances which are 33.60dB, 31.99dB, and 33.62dB. Finally, we note that our proposed CorInfomax method outperforms other biologically plausible neural networks and the batch ICA-InfoMax algorithm, while its performance is on par with the batch LD-InfoMax algorithm, and its output PSNR values are 32.45dB, 29.72dB, and 32.37dB. We note that ICA and NSM algorithms assume independent and uncorrelated sources, respectively, so their performances are remarkably affected by the correlation level of the sources. Typically, the best performance is obtained by LD-InfoMax due to its batch nature and dependent source separation capability. Finally, we notice that the biologically plausible WSM network is able to separate correlated sources whereas CorInfoMax performs better both visually and in terms of the PSNR metric. The hyperparameters used in this experiment for CorInfoMax are included in Appendix D.4, and the codes to reproduce each output are included in our supplementary material.

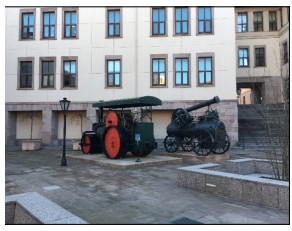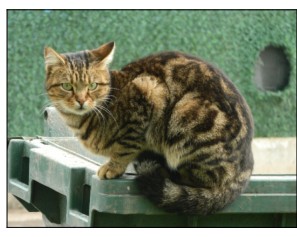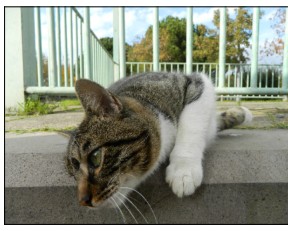

(a)

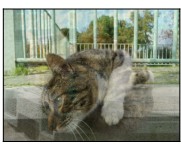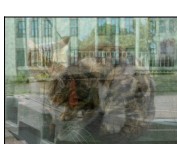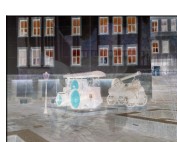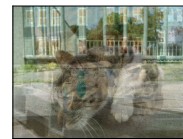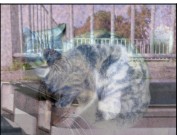

(b)

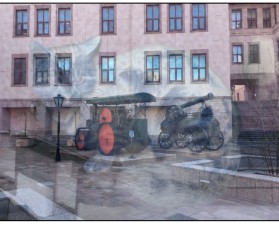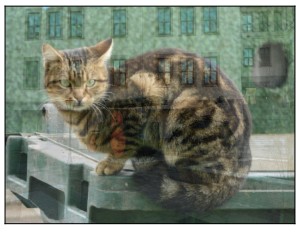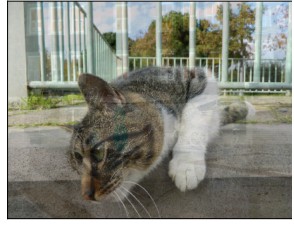

(c)

Figure 16: Image separation example: (a) Original RGB images, (b) mixture RGB images, (c) ICA outputs.

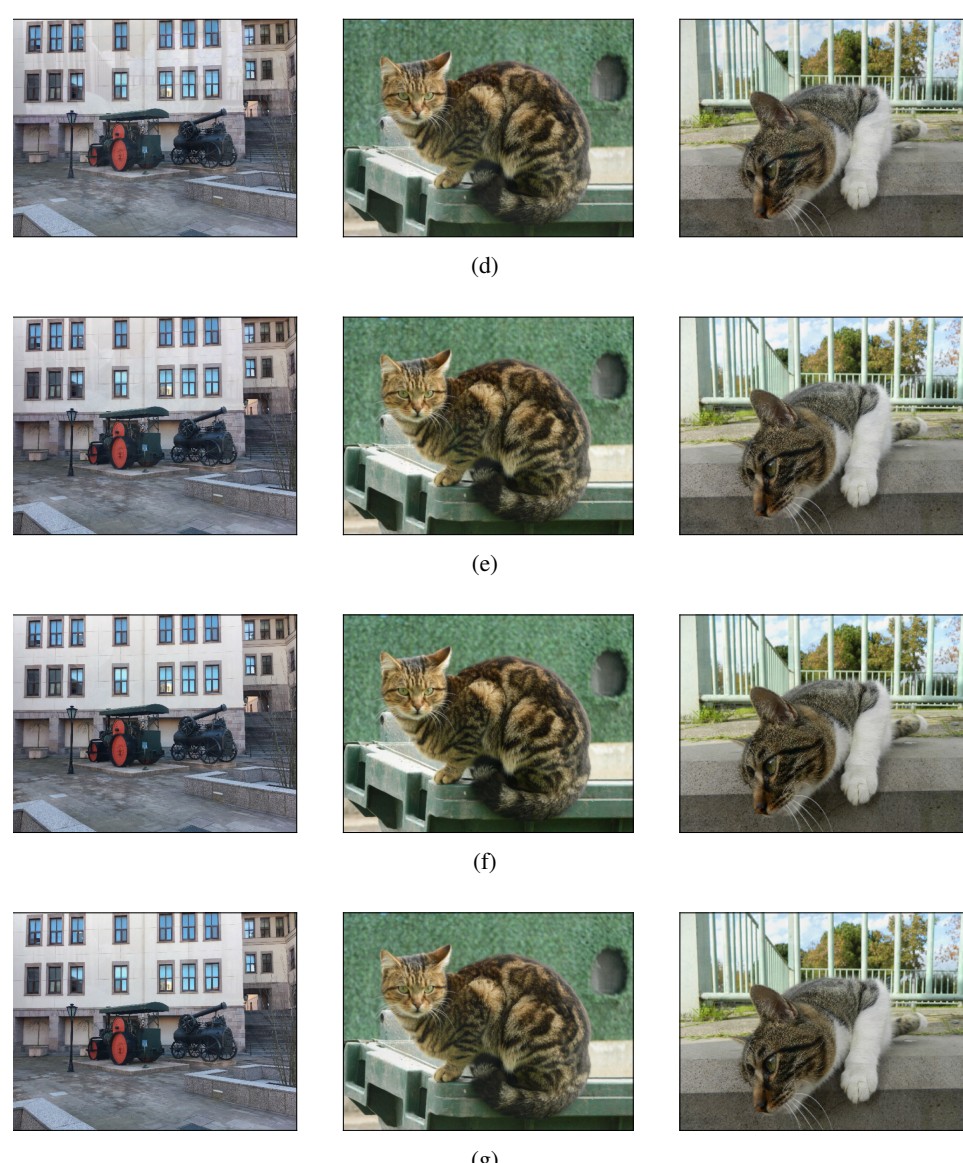

Figure 16: (d) NSM outputs, (e) WSM outputs, (f) LD-InfoMax Outputs, (g) CorInfoMax (ours) Outputs.

## D.4 HYPERPARAMETER SELECTIONS

Hyperparameter selection has a critical impact on the performance of the neural networks offered in this article. In this section, we provide the list of our hyperparameter selections for the experiments provided in the article. These parameters are selected through ablation studies and some trials, which are discussed in Appendix D.5.

Table 3 summarizes the hyperparameter selections for the special domain choices provided in Section 2.1. Based on this table, we can observe that the hyperparameter sets for different special source domains generally resemble each other. However, there are some noticable domain-specific changes in some parameters, such as the starting learning rate for neural dynamics ($\eta_{\boldsymbol{y}}(1)$) and the learning rate for the Lagrangian variable ($\eta_\lambda(\nu)$).

For the other experiments presented in the Appendices D.2.4 and D.3, Table 4 summarizes the corresponding hyperparameter selections.

Table 3: CorInfoMax network hyperparameter selections for special source domains in 2.1.

| Source Domain | Hyperparameters |
|---|---|
| $\mathcal{P} = \mathcal{B}_{\ell_\infty}$ | $\boldsymbol{W}(1) = \boldsymbol{I}, \quad \boldsymbol{B}_{\boldsymbol{y}}^{\zeta_y}(1) = 5\boldsymbol{I}, \quad \boldsymbol{B}_{\boldsymbol{e}}^{\zeta_e}(1) = 5000\boldsymbol{I}$ 
 $\zeta_{\boldsymbol{y}} = 1 - 10^{-2}, \quad \zeta_{\boldsymbol{e}} = 1 - 2 \times 10^{-2}, \quad \mu_{\boldsymbol{W}} = 3 \times 10^{-2}$ 
 $\nu_{\max} = 500, \quad \eta_{\boldsymbol{y}}(\nu) = 0.9/\nu, \quad \epsilon_t = 10^{-6}$ |
| $\mathcal{P} = \mathcal{B}_{\ell_\infty,+}$ | $\boldsymbol{W}(1) = \boldsymbol{I}, \quad \boldsymbol{B}_{\boldsymbol{y}}^{\zeta_y}(1) = 5\boldsymbol{I}, \quad \boldsymbol{B}_{\boldsymbol{e}}^{\zeta_e}(1) = 2000\boldsymbol{I}$ 
 $\zeta_{\boldsymbol{y}} = 1 - 10^{-2}, \quad \zeta_{\boldsymbol{e}} = 1 - 10^{-1}/3, \quad \mu_{\boldsymbol{W}} = 3 \times 10^{-2}$ 
 $\nu_{\max} = 500, \quad \eta_{\boldsymbol{y}}(\nu) = \max\{0.9/\nu, 10^{-3}\}, \quad \epsilon_t = 10^{-6}$ |
| $\mathcal{P} = \mathcal{B}_1$ | $\boldsymbol{W}(1) = \boldsymbol{I}, \quad \boldsymbol{B}_{\boldsymbol{y}}^{\zeta_y}(1) = \boldsymbol{I}, \quad \boldsymbol{B}_{\boldsymbol{e}}^{\zeta_e}(1) = 1000\boldsymbol{I}$ 
 $\zeta_{\boldsymbol{y}} = 1 - 10^{-2}, \quad \zeta_{\boldsymbol{e}} = 1 - 10^{-2}, \quad \mu_{\boldsymbol{W}} = 3 \times 10^{-2}$ 
 $\nu_{\max} = 500, \quad \eta_{\boldsymbol{y}}(\nu) = \max\{0.1/\nu, 10^{-3}\},$ 
 $\eta_\lambda(\nu) = 1, \quad \epsilon_t = 10^{-6}$ |
| $\mathcal{P} = \mathcal{B}_{1,+}$ | $\boldsymbol{W}(1) = \boldsymbol{I}, \quad \boldsymbol{B}_{\boldsymbol{y}}^{\zeta_y}(1) = 5\boldsymbol{I}, \quad \boldsymbol{B}_{\boldsymbol{e}}^{\zeta_e}(1) = 1000\boldsymbol{I}$ 
 $\zeta_{\boldsymbol{y}} = 1 - 10^{-2}, \quad \zeta_{\boldsymbol{e}} = 1 - 10^{-2}, \quad \mu_{\boldsymbol{W}} = 3 \times 10^{-2}$ 
 $\nu_{\max} = 500, \quad \eta_{\boldsymbol{y}}(\nu) = \max\{0.1/\nu, 10^{-3}\},$ 
 $\eta_\lambda(\nu) = 1, \quad \epsilon_t = 10^{-6}$ |
| $\mathcal{P} = \Delta$ | $\boldsymbol{W}(1) = \boldsymbol{I}, \quad \boldsymbol{B}_{\boldsymbol{y}}^{\zeta_y}(1) = 5\boldsymbol{I}, \quad \boldsymbol{B}_{\boldsymbol{e}}^{\zeta_e}(1) = 1000\boldsymbol{I}$ 
 $\zeta_{\boldsymbol{y}} = 1 - 10^{-2}, \quad \zeta_{\boldsymbol{e}} = 1 - 10^{-2}, \quad \mu_{\boldsymbol{W}} = 3 \times 10^{-2}$ 
 $\nu_{\max} = 500, \quad \eta_{\boldsymbol{y}}(\nu) = \max\{0.1/\nu, 10^{-3}\},$ 
 $\eta_\lambda(\nu) = 0.05, \quad \epsilon_t = 10^{-6}$ |

Table 4: CorInfoMax network hyperparameter selections for the polytope with mixed latent attributes in (A.19) and application examples in Appendix D.3.

| Experiment | Hyperparameters |
|---|---|
| Polytope in (A.19) with Mixed Attributes (Canonical) | $\boldsymbol{W}(1) = \boldsymbol{I}, \quad \boldsymbol{B}_{\boldsymbol{y}}^{\zeta_y}(1) = \boldsymbol{I}, \quad \boldsymbol{B}_{\boldsymbol{e}}^{\zeta_e}(1) = 1000\boldsymbol{I}$ 
 $\zeta_{\boldsymbol{y}} = 1 - 10^{-2}, \quad \zeta_{\boldsymbol{e}} = 1 - 10^{-2}, \quad \mu_{\boldsymbol{W}} = 5 \times 10^{-2}$ 
 $\nu_{\max} = 500, \quad \eta_{\boldsymbol{y}}(\nu) = \max\{0.25/\nu, 10^{-4}\},$ 
 $\eta_{\boldsymbol{\lambda}}(\nu) = 0.1, \quad \epsilon_t = 10^{-6}$ |
| Polytope in (A.19) with Mixed Attributes (Feature-based) | $\boldsymbol{W}(1) = \boldsymbol{I}, \quad \boldsymbol{B}_{\boldsymbol{y}}^{\zeta_y}(1) = 5\boldsymbol{I}, \quad \boldsymbol{B}_{\boldsymbol{e}}^{\zeta_e}(1) = 2500\boldsymbol{I}$ 
 $\zeta_{\boldsymbol{y}} = 1 - 10^{-2}, \quad \zeta_{\boldsymbol{e}} = 1 - 10^{-2}, \quad \mu_{\boldsymbol{W}} = 5 \times 10^{-2}$ 
 $\nu_{\max} = 500, \quad \eta_{\boldsymbol{y}}(\nu) = \max\{0.1/\nu, 10^{-10}\},$ 
 $\eta_{\lambda_1}(\nu) = \eta_{\lambda_2}(\nu) = 1, \quad \epsilon_t = 10^{-6}$ |
| Sparse Dictionary Learning | $\boldsymbol{W}(1) = \boldsymbol{I}, \quad \boldsymbol{B}_{\boldsymbol{y}}^{\zeta_y}(1) = \boldsymbol{I}, \quad \boldsymbol{B}_{\boldsymbol{e}}^{\zeta_e}(1) = 20000\boldsymbol{I}$ 
 $\zeta_{\boldsymbol{y}} = 1 - 10^{-3}/7, \quad \zeta_{\boldsymbol{e}} = 1 - 10^{-3}/7, \quad \mu_{\boldsymbol{W}} = 0.25 \times 10^{-3}$ 
 $\nu_{\max} = 500, \quad \eta_{\boldsymbol{y}}(\nu) = \max\{1.5/(\nu \times 0.01), 10^{-8}\},$ 
 $\eta_\lambda(\nu) = 3 \times 10^{-2}, \quad \epsilon_t = 10^{-6}$ |
| Digital Communications | $\boldsymbol{W}(1) = \boldsymbol{I}, \quad \boldsymbol{B}_{\boldsymbol{y}}^{\zeta_y}(1) = 5\boldsymbol{I}, \quad \boldsymbol{B}_{\boldsymbol{e}}^{\zeta_e}(1) = 1000\boldsymbol{I}$ 
 $\zeta_{\boldsymbol{y}} = 1 - 10^{-2}, \quad \zeta_{\boldsymbol{e}} = 1 - 10^{-2}, \quad \mu_{\boldsymbol{W}} = 3 \times 10^{-2}$ 
 $\nu_{\max} = 500, \quad \eta_{\boldsymbol{y}}(\nu) = \max\{0.9/\nu, 10^{-3}\}, \quad \epsilon_t = 10^{-6}$ |
| Video Separation | $\boldsymbol{W}(1) = \boldsymbol{I}, \quad \boldsymbol{B}_{\boldsymbol{y}}^{\zeta_y}(1) = \boldsymbol{I}, \quad \boldsymbol{B}_{\boldsymbol{e}}^{\zeta_e}(1) = 65\boldsymbol{I}$ 
 $\zeta_{\boldsymbol{y}} = 1 - 10^{-1}/5, \quad \zeta_{\boldsymbol{e}} = 0.3, \quad \mu_{\boldsymbol{W}} = 6 \times 10^{-2}$ 
 $\nu_{\max} = 500, \quad \eta_{\boldsymbol{y}}(\nu) = \max\{0.5/\nu, 10^{-3}\}, \quad \epsilon_t = 10^{-6}$ |
| Image Separation | $\boldsymbol{W}(1) = \boldsymbol{I}, \quad \boldsymbol{B}_{\boldsymbol{y}}^{\zeta_y}(1) = \boldsymbol{I}, \quad \boldsymbol{B}_{\boldsymbol{e}}^{\zeta_e}(1) = 100\boldsymbol{I}$ 
 $\zeta_{\boldsymbol{y}} = 1 - 10^{-1}/15, \quad \zeta_{\boldsymbol{e}} = 0.5, \quad \mu_{\boldsymbol{W}} = 5 \times 10^{-2}$ 
 $\nu_{\max} = 500, \quad \eta_{\boldsymbol{y}}(\nu) = \max\{0.5/\nu, 10^{-3}\}, \quad \epsilon_t = 10^{-6}$ |

## D.5    ABLATION STUDIES ON HYPERPARAMETER SELECTIONS

In this section, we illustrate an ablation study on selection of two hyperparameters for the non-negative antisparse CorInfoMax network: we experiment with selection for the learning rate of the

feedforward weights $\mu_{\boldsymbol{W}}$ and the initialization of the inverse error correlation matrix $\boldsymbol{B}_e^{\zeta_e}$. For these hyperparameters, we consider the following selections:

- $\mu_{\boldsymbol{W}} \in \{5 \times 10^{-3}, 10^{-2}, 3 \times 10^{-2}, 5 \times 10^{-2}\}$,
- $\boldsymbol{B}_e^{\zeta_e} \in \{10^3, 2 \times 10^3, 5 \times 10^3, 10^4\}$.

We consider the experimental setup in Section 4.1 for both uncorrelated sources and correlated sources with correlation parameter $\rho = 0.6$. For the ablation study for $\mu_{\boldsymbol{W}}$, we used fixed $\boldsymbol{B}_e^{\zeta_e}$ as indicated in Table 3, and vice versa.

Figures 17a and 17b illustrate the mean SINR with standard deviation envelope results with respect to $\mu_{\boldsymbol{W}}$ for uncorrelated and correlated source separation settings, respectively. Note that although selection $\mu_{\boldsymbol{W}} = 5 \times 10^{-2}$ seems to be better for uncorrelated sources, its performance degrades for correlated sources as for some of the realizations, the algorithm diverges. We conclude that the selection $\mu_{\boldsymbol{W}} = 3 \times 10^{-2}$ is near optimal in this setting and obtains good SINR results for both correlated and uncorrelated sources. For the initialization of $\boldsymbol{B}_e^{\zeta_e}$, Figures 17c and 17d illustrate the SINR performances of CorInfoMax for uncorrelated and correlated source separation experiments, respectively. Note that the selections $\boldsymbol{B}_e^{\zeta_e} = 2000\boldsymbol{I}$ and $\boldsymbol{B}_e^{\zeta_e} = 5000\boldsymbol{I}$ are suitable considering both settings.

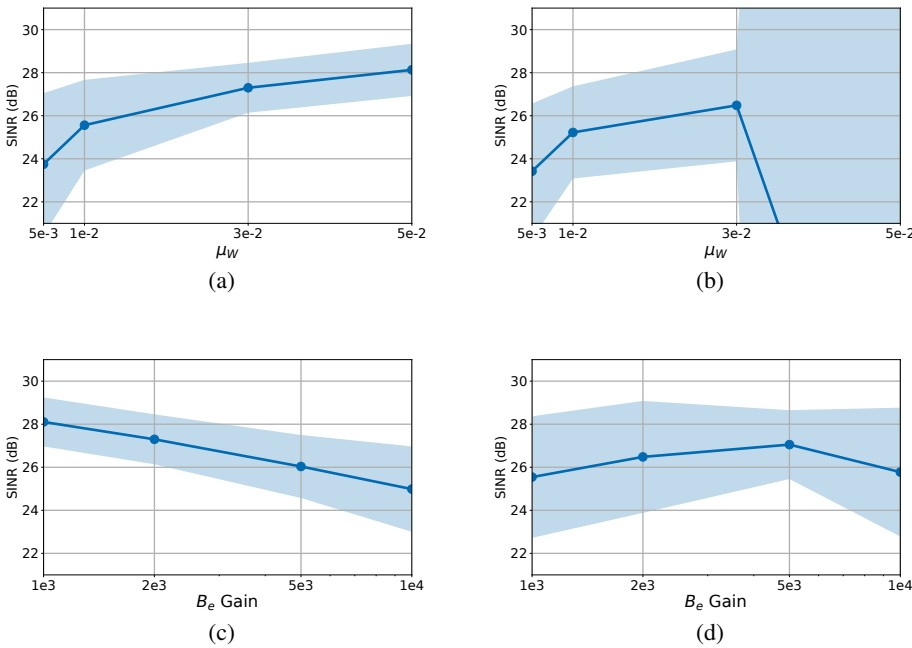

Figure 17: CorInfoMax ablation studies on the hyperparameter selections of $\mu_{\boldsymbol{W}}$ and $\boldsymbol{B}_e^{\zeta_e}$: mean SINR with standard deviation envelopes corresponding to 50 realizations. (a) ablation study of $\mu_{\boldsymbol{W}}$ on uncorrelated nonnegative antisparse sources, (b) ablation study of $\mu_{\boldsymbol{W}}$ on correlated (with $\rho = 0.6$) nonnegative antisparse sources, (c) ablation study of $\boldsymbol{B}_e^{\zeta_e}$ on uncorrelated nonnegative antisparse sources, (d) ablation study of $\boldsymbol{B}_e^{\zeta_e}$ on correlated (with $\rho = 0.6$) nonnegative antisparse sources,

### D.6 ABLATION STUDIES ON THE EFFECT OF NUMBER OF MIXTURES

The number of mixtures with respect to the number of sources might be crucial for blind source separation problem, and it can affect the overall performance of the proposed method. To explore the impact of the number of mixtures on the SINR performance of our proposed method, we performed experiments with varying number of mixtures and fixed number of sources for nonnegative antisparse, i.e., $\mathcal{P} = \mathcal{B}_{\infty,+}$, and sparse, i.e., $\mathcal{P} = \mathcal{B}_1$, source separation settings. In these experi-

mental settings, we consider 5 sources, and change the number of mixtures gradually from 5 to 10. Figure 18a and 18b illustrate the overall SINR performances with a standard deviation envelope for nonnegative antisparse CorInfoMax and sparse CorInfoMax networks with respect to the number of mixtures, which are averaged over 50 realizations, respectively. For each realization, we randomly generate a mixing matrix with i.i.d. standard normal entries. We observe that the performance of CorInfoMax networks monotonically improves as the number of mixtures increases. This aligns with the theoretical expectations that the condition of the random mixing matrix improve with increasing number of mixtures Chen & Dongarra (2005) which positively impacts the algorithm's numerical performance.

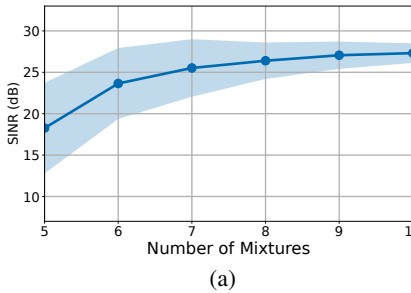
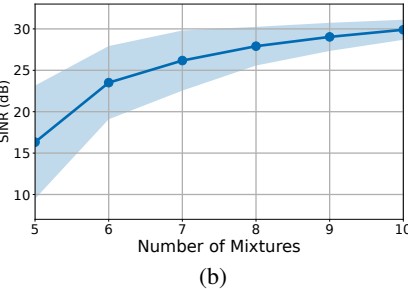

(a)                    (b)

Figure 18: CorInfoMax ablation studies on the impact of number of mixtures with fixed number of sources (5 sources). (a) illustrates the SINR performances of nonnegative antisparse CorInfoMax network as a function of number of mixtures, (b) illustrates the SINR performances of sparse CorInfoMax network as a function of number of mixtures.

### D.7    Computational Complexity of the Proposed Approach

The complexity of the proposed approach can be characterized based on the analysis of output dynamics and weight updates. For simplicity, we consider the complexity of the antisparse CorInfoMax network in Figure 4: Let $\nu_{max}$ represent the maximum count for the neural dynamic iterations.

*Neural Dynamics' Complexity:* For one neural dynamic iteration, we observe the following number of operations

- Error calculation $e(k; \nu) = y(k; \nu) - W(k)x(k)$ requires $nm$ multiplications,
- $B_y^{\zeta_y}(k)y(k; \nu)$ requires $n^2$ multiplications,
- $B_e^{\zeta_e}(k)e(k; \nu)$ requires $n$ multiplications as $B_e^{\zeta_e}(k) = \frac{1}{\epsilon}I$.

As a result, $\nu_{max}$ neural dynamics iterations require $\nu_{max}(mn + n^2 + n) \approx \nu_{max}mn$ multiplications per output computation.

*Weight Updates' Complexity:* For the weight updates, we note the following number of operations

- $W(k+1) = W(k) + \mu_W(k)e(k)x(k)^T$ requires $2mn$ multiplication,
- $B_y^{\zeta_y}(k+1) = \frac{1}{\zeta_y}(B_y^{\zeta_y}(k) - \frac{1-\zeta_y}{\zeta_y}B_y^{\zeta_y}(k)y(k)y(k)^T B_y^{\zeta_y}(k))$ requires $n^2 + n^2 + n^2 + 1 + \frac{n(n+1)}{2} = \frac{7n^2 + n + 2}{2}$ multiplications.

Therefore, weight updates for learning require $2mn + \frac{7n^2+n+2}{2}$ multiplications per input sample.

Taking into account both components of the computations, complexity is dominated by the neural dynamics iterations which require approximately $\mathcal{O}(\nu_{max}mn)$ operations. This is in the same order as the complexity reported in Bozkurt et al. (2022) for biologically plausible WSM, NSM, and BSM networks.

