# OpenReview forum: "Correlative Information Maximization Based Biologically Plausible Neural Networks for Correlated Source Separation"
_ICLR.cc/2023/Conference — ICLR 2023 poster_

### Official Review · Reviewer_C2ij · 2022-10-23

**Confidence:** 3
**Correctness:** 4
**Technical Novelty And Significance:** 2
**Empirical Novelty And Significance:** 2
**Recommendation:** 5

**Clarity, Quality, Novelty And Reproducibility:**


Clarity:
In the abstract, the understanding of domain and "their presumed sets" needs to be made more explicit.

Figure 1 is very detailed but it is presented without sufficient introduction to the notations, these should be in the caption.

Ambiguous 'they' in "or example, the lateral connections of the outputs in Bozkurt et al. (2022) are based on the output autocorrelation matrix, while they are based on the inverse of the output autocorrelatin matrix" (also note the typo).

Equation 3e can be interpreted as a surrogate for H(Y) - H(Y|X), which is discussed in the appendix but should be included in the body.

Section 4.2
"to exclude the negative frame effect"

Novelty:
The work builds on an existing objective and follows other biologically plausible networks. Thus, this is incremental novelty.

Quality:
The derivations appear sound and the overall presentation is logical.

Although dependent sources are discussed a real world example of dependent sources is not illustrated.

The case of separation with as many observations as sources is not discussed.   I.e. 5 mixtures of 3 sources is demonstrated. How well does it work for 3 mixtures of 3 sources?

Reproducibility:
There is a relatively large number of hyper parameters. It is not clear if the hyper parameter selection was done in a reproducible manner. I.e. if the result cannot be evaluated by a metric or qualitatively can the user practically tune the hyper-parameters? If so with what criteria?

**Strength And Weaknesses:**

Strengths:
The work builds off of a solid foundation for BSS using second-order statistics and is logically presented.
The method compares well to related work in ideal cases.
The local updates seem reasonable based on the assumptions.

Weaknesses:
The figures of networks are not clear from their captions.
The overloaded notation is quite cumbersome at times.
The method is not compared to other approaches on the real data.
Although dependent sources are discussed a real world example of dependent sources is not illustrated.
There is a relatively large number of hyper parameters. It is not clear if the hyper parameter selection was done in a reproducible manner. I.e. if the result cannot be evaluated by a metric or qualitatively can the user practically tune the hyper-parameters? If so with what criteria?

**Summary Of The Paper:**

The paper develops an online algorithm with local update rules for blind source separation using information maximization in terms of second-order statistics and knowledge of the domain of the sources.

Like previous biologically plausible algorithms, the online algorithm can be instantiated using only Hebbian and anti-Hebbian learning in two or three layer networks.

In particular, when the domain of the sources are known to be a polytope (L1-ball for sparse vectors), L$\infty$-ball for dense vectors, possibly non-negative (yielding simplex for the former), the Lagrangian of the constrained maximization problem is shown to induce different activation functions.

Results show that the method is competitive to other prior work on synthetic problems where the true domain of the sources is known.

**Summary Of The Review:**

The paper proposes an online algorithm for blind source separation using a log-det based information maximization principle under a second order statistic.  The method builds from previous work, and is logical and appears correct. Paper does thorough job in describing how this leads to biologically plausible cases. But the clarity of the presentation at times could be improved. Overall novelty is mostly incremental. Results are detailed but challenging real world cases are not covered. The knowledge of the domain of the sources is required, and it is unclear how a user would be able to blindly select hyper-parameters.

---

> ### Author Response · Authors · 2022-11-12
> **Response to Reviewer C2ij - Part 5 of 5**
>
> >*Summary of the paper*: ... Results are detailed but challenging real world cases are not covered.
>
> Thank you for this comment. In the revised article, we include several examples with real sources including
> * Video separation example (Section 4.2 and Appendix D.3.3).
> * Dictionary Learning with $12 \times 12$ natural image patches (Appendix D.3.1): Note that in this case, the scenario corresponds to the $144 \times 144$ separation system, which may be considered a challenging problem due to the input-output dimensions. The proposed Sparse CorInfoMax network for this application is a biologically plausible solution to the problem in the original Olshausen & Field (1997) article in which the recurrent weights are tightly connected to the feedback weights.
> * Image separation example (Appendix D.3.4).
>
> >*Summary of the paper*: ... The knowledge of the domain of the sources is required,
>
> This is true, however, to be able to solve the ill-possed source separation problem, we need to exploit a form of side information such as the mutual independence of sources. The source domain assumption can be used to eliminate the need for the strong and potentially irrelevant mutual independence assumption. The source domains are known in several applications as exemplified in Section 2.1: unit-simplex for hyperspectral unmixing and text mining, $\ell_\infty$-norm ball for digital communication signals, $\ell_1$-norm ball for sparse signals, nonnegative section of $\ell_\infty$-norm ball for natural images. The side information about sign and relational features (such as mutual sparsity), as illustrated by polytope construction in (2) and Appendices C.5 and D2.4, can be used to specify source domains.
>
> >*Summary of the paper*: ... and it is unclear how a user would be able to blindly select hyper-parameters.
>
> This is fair, however, proper selection of hyperparameters is a common challenge in the implementation of neural networks, especially for recurrent networks with feedback. As we responded earlier, we provide two detailed Appendix sections (Appendix D.4 (with Tables 3 and 4) and Appendix D.5) in the supplementary part of our article, which is a comprehensive reference for future implementations. Furthermore, we share our code that contains implementations for several source-mixing scenarios, which would be good starting points for any implementation.

---

> ### Author Response · Authors · 2022-11-12
> **Response to Reviewer C2ij - Part 4 of 5**
>
> >*Summary of the paper*: ... But the clarity of the presentation at times could be improved.
>
> We believe that by correcting/implementing the points suggested by the reviewers and with additional corrections and clarifications, our revised article is easier to understand and follow.
>
> >*Summary of the paper*: Overall novelty is mostly incremental.
>
> We appreciate this comment, but we kindly disagree. Our article offers a novel and comprehensive framework for generating biologically plausible neural networks to solve the BSS problem for potentially correlated sources based on the principle of correlative information maximization. To our knowledge, no such biologically plausible framework exists that makes direct of the correlative information maximization principle to solve the problem of blind separation of correlated sources. To clarify the novelty, we can provide the following high-level summary:
> * The Determinant Maximization criterion has been proposed for structured matrix factorization frameworks, including Nonnegative and Polytopic Matrix Factorization.
> * The reference, Erdogan (2022), provides a second-order statistics-based information maximization perspective for these determinant-maximization/minimization-based matrix factorization methods and for the connected BSS problems. However, the algorithm provided in this reference is a ***batch*** algorithm, i.e., no neural network implementations.
> * The framework proposed in our article offers an ***online formulation*** for the correlative information maximization criterion of Erdogan(2022), which enables implementations of biologically plausible neural networks.
> * The reference Bozkurt et al. (2022) also offers a biologically plausible BSS framework for determinant maximization. However, the main differences are the following: this reference is not directly based on the correlative information maximization objective. Furthermore, Bozkurt et al. (2022) uses a ***similarity matching criterion*** for generating biologically plausible networks, which we do not use in our article. The resulting neural network architectures and learning rules are completely different. The networks in Bozkurt et al. (2022) have neurons with learnable gains,  the lateral weights corresponding to the output correlation matrix, which results from the similarity matching criterion (in our case, the lateral weights correspond to the inverse of the correlation matrix, which results from logdeterminant optimization). The feedforward weights of the networks in Bozkurt et al. (2022) correspond to a cross-correlation matrix between the inputs and outputs of a layer,  whereas, for the proposed framework, the feedforward connections correspond to the linear predictor of the output from the input. Finally, the feedback weights of networks in Bozkurt et al. (2020) are the transpose of the feedforward weights, where in our framework the feedback weights are diagonal and not the transpose of feedforward weights.
>
> Therefore, our comprehensive framework is a totally novel approach that clearly distinguishes itself from existing biologically plausible and batch approaches to solving the correlated BSS problem. We will gladly provide further clarification if we are given more specific grounds for any remaining reservations about novelty.
>
> To clarify the novelty of our article, we revised the discussion in Section 1.1 (Related Work and Contributions) of the article.

---

> ### Author Response · Authors · 2022-11-12
> **Response to Reviewer C2ij - Part 3 of 5**
>
> >*Clarity*: Figure 1 is very detailed but it is presented without sufficient introduction to the notations, these should be in the caption.
>
> Based on your suggestion, we modified the caption of Figure 1 (along with other network figures) to provide description of all its components.
>
> >*Clarity*: Ambiguous 'they' in "or example, the lateral connections of the outputs in Bozkurt et al. (2022) are based on the output autocorrelation matrix, while they are based on the inverse of the output autocorrelatin matrix" (also note the typo).
>
> Thanks for pointing out this typo. In the revised article, we modified this sentence as
> * “For example, the lateral connections of the outputs in Bozkurt et al. (2022) are based on the output autocorrelation matrix, while the lateral connections for our proposed framework are based on the inverse of the output autocorrelation matrix.”
>
> >*Clarity*: Equation 3e can be interpreted as a surrogate for $H(Y) - H(Y|X)$, which is discussed in the appendix but should be included in the body.
>
> Thank you for this clarification suggestion. If we understand your point correctly, Equation 3a is not a surrogate for Shannon mutual information $H(Y)-H(Y|X)$, rather it is the sample-based estimate of log-determinant mutual information $H_{LD}(Y)-H_{LD}(Y|_{L} X)$ which is a measure of correlation. (Shannon mutual information is a measure of dependence). Note that LD-mutual information is second-order-statistics-based only and does not require Gaussian distribution assumption. (LD-Mutual Information and Shannon Mutual Information overlap in the Gaussian case.)
>
> >*Clarity*:  Section 4.2 "to exclude the negative frame effect"
>
> We replaced this note with
> * “ to ensure nonnegative mixtures so that they can be displayed as proper images without loss of generality ”
>
> >*Quality*: The case of separation with as many observations as sources is not discussed. I.e. 5 mixtures of 3 sources is demonstrated. How well does it work for 3 mixtures of 3 sources?
>
> Thank you for this suggestion. In fact, we considered the case with as many mixtures as sources in the original submission. In Appendix D.3.1, for the application on sparse dictionary learning from $12 \times 12$ image patches, both the inputs and the outputs are $144$ dimensional vectors corresponding to square $144 \times 144$ mixing/separator systems.
>
> To address your specific request on the $3\times 3$ mixing scenario, we performed a new experiment for the video separation example with $3$ sources and $3$ mixtures. For this experiment, we applied the following minimal hyperparameter changes: 1) we increased the total number of iterations of the network by using more pixels per frame in each source, as the algorithm requires more samples, in the reduced mixture scenario, to recover the original videos, and 2) we decreased the gain of the initial error correlation matrix to make the convergence of neural dynamics more smooth. You can view the corresponding video that illustrates the $3\times 3$ mixing experiment at the following link:
> * [https://figshare.com/s/683ab5f1a9bd7d0c1caa](https://figshare.com/s/683ab5f1a9bd7d0c1caa)
>
> In this experiment, CorInfoMax is trained to a stage with PSNR performances of $33.81 \ dB$, $35.02 \ dB$, $28.98 \ dB$ for each source. We also share our code for this experiment in the supplementary zip file.
>
> In addition, to address your question in a more systematic way, in Appendix D.6 of the revised article,  we include an ablation study on the effect of the number of mixtures on the resulting SINR of the proposed method. In this study, we performed experiments for both nonnegative antisparse $\mathcal{P}=\mathcal{B}_{\infty,+}$ and sparse $\mathcal{P}=\mathcal{B}_1$ sources. We fixed the number of sources as $5$ and obtained the SINR performance of the CorInfoMax algorithm for a different number of mixtures. As the condition of the mixing matrix and the numerical stability of the algorithm improve with increasing number of mixtures ( Chen and Dongarra (2005)), we observe a monotonic increase in the performance, as expected. The following are the corresponding SINR vs. number of mixtures plots for nonnegative antisparse and sparse sources  provided in Appendix D.6, respectively:
> * [Figure 18(a)](https://figshare.com/s/39702521dbb77478b5f4) in Appendix D.6
> * [Figure 18(b)](https://figshare.com/s/dbe04b07455c51f9ecc1) in Appendix D.6

---

> ### Author Response · Authors · 2022-11-12
> **Response to Reviewer C2ij - Part 2 of 5**
>
> >*Weaknesses, Quality*: Although dependent sources are discussed a real world example of dependent sources is not illustrated.
>
> We apologize for not being clear with the setting of our numerical experiments. Regarding numerical experiments, we included synthetic examples as proof-of-concept demonstrations for robustness against varying levels of source correlation, input noise power, and mixing system. The **video separation** application provided in the Numerical Experiments (Section 4) section in Appendix D.3.3 is, in fact, a perfect visual example of **naturally correlated sources**. However, *we did not report their correlation levels* between the frames in the original article. To make this point clear, in the revised article, we report the correlation levels between the frames of different video sources. According to Section 4.2 of the revised article, the average (across frames) and maximum correlation levels for the video sources (across frames) are $\rho_{12}^{\text{average}}=-0.1597$, $\rho_{13}^{\text{average}}=-0.1549$, $\rho_{23}^{\text{average}}=0.3811$ and $\rho_{12}^{\text{maximum}}=0.3139$, $\rho_{13}^{\text{maximum}}=0.2587$, $\rho_{23}^{\text{maximum}}=0.5173$, respectively. Therefore, the video example serves as a nice illustration to demonstrate the successful separation capability for naturally correlated sources.
>
> Furthermore, as we noted in our earlier response on the comparison with other approaches, we have included an experiment with naturally correlated image sources in Appendix D.3.4 of the revised article. The pairwise Pearson correlation coefficients for three images used in this experiment are $\rho_{12}=0.076, \rho_{13}=0.262, \rho_{23}=0.240$, respectively. In this example, we compare the proposed CorInfoMax neural network framework with selected biologically plausible neural networks (NSM and WSM) and batch algorithms for independent/dependent component analysis (ICA-InfoMax and LD-InfoMax). The results of this experiment are provided in Appendix D.3.4 and summarized in our response to your question about the comparison with other approaches.
>
> >*Weaknesses*: There is a relatively large number of hyper parameters. It is not clear if the hyper parameter selection was done in a reproducible manner. I.e. if the result cannot be evaluated by a metric or qualitatively can the user practically tune the hyper-parameters? If so with what criteria?
>
> Thank you for this comment. We agree that there is a large number of hyperparameters due to the degrees of freedom in the formulation and optimization of the online normative criterion. To provide guidance to ease the implementation of CorInfoMax networks, we provided two Appendix sections related to the selection of these hyperparameters: Appendix D.4 provides details about our hyperparameter choices. Based on Table $3$ in this appendix, we observe that the hyperparameter choices for different source domains resemble each other. We also provide information about the ablation studies that we performed in search of parameters in Appendix D.5 (and Figure 16) in detail. In addition, our detailed codes that are available in the supplementary documents also serve as useful references for future implementations.
>
> >*Clarity*: In the abstract, the understanding of domain and "their presumed sets" needs to be made more explicit.
>
> Thank you for this suggestion. We modified the sentence containing “their presumed sets” in the abstract as
> * “To derive this network, we choose the maximum correlative information transfer from inputs to outputs as the separation objective under the constraint that the output vectors are restricted to the set where the source vectors are assumed to be located.”
>
> In other words, “their presumed sets” is replaced with “ the set where source vectors are assumed to be located.”. In addition, we replaced the “infinitely many source domains” in the following sentence with the “infinitely many set choices for the source domain” for further clarification on “domain”.

---

> ### Author Response · Authors · 2022-11-12
> **Response to Reviewer C2ij - Part 1 of 5**
>
> We thank you for your time and constructive comments. We address your comments and questions in the revised article and in our response below.
>
> >*Weaknesses*: The figures of networks are not clear from their captions. The overloaded notation is quite cumbersome at times.
>
> We appreciate your feedback on the captions of the figures. Based on your suggestions and Reviewer hwJs's suggestions, we extended all figure captions. In particular, in the revised article, we included more thorough explanations of the CorInfoMax networks in the figure captions, describing all of their components.  The notational complexity is mainly attributable to the coexistence of time frames for both input/output samples time and algorithm iterations. That is also partly due to our desire to provide a precise and implementable description in sufficient detail. In the revised article, we improved the presentation and readability of mathematical expressions.
>
> >*Weaknesses*: The method is not compared to other approaches on the real data.
>
> This is fair. To address your point, we have added a new example in Appendix D.3.4, in which the sources are photographs, which serve as excellent examples of naturally correlated sources. In fact, the pairwise Pearson correlation coefficients for three images used in this experiment are $\rho_{12}=0.076, \rho_{13}=0.262, \rho_{23}=0.240$, respectively. We compared the proposed CorInfoMax networks with selected existing **biologically plausible neural network solutions** (Nonnegative Similarity Matching (NSM) and Weighted Similarity Matching (WSM)) with correlated source separation capability, and **batch algorithms for independent and dependent component analysis** (ICA-InfoMax and  LD-InfoMax).  As illustrated by this example, the proposed CorInfoMax approach achieves a performance level superior to other biologically plausible network approaches and batch ICA-InfoMax algorithm and close to the batch LD-InfoMax algorithm, which illustrates its potential for naturally correlated sources.
>
> Following are source, mixture and algorithm output images that are available in Appendix D.3.4 of the revised article.
> * Original Sources : [Figure 16(a)](https://figshare.com/s/cb386db6287a56da53af)
> * Mixtures : [Figure 16(b)](https://figshare.com/s/f4c818c0034a1fcd5cc3)
> * Proposed CorInfoMax Outputs (PSNR values are $\displaystyle 32.45 \ dB$, $\displaystyle 29.72 \ dB$, and $\displaystyle 32.37 \ dB$) : [Figure 16(g)](https://figshare.com/s/1a54b10d6fc6cea3f7d7)
> * Batch ICA-InfoMax Outputs (PSNR values are $\displaystyle18.56 \ dB$, $\displaystyle20.52 \ dB$, and $\displaystyle21.06 \ dB$) : [Figure 16(c)](https://figshare.com/s/abc2981b8ce5eee8b94a)
> * Biologically Plausible NSM Outputs (PSNR values are $\displaystyle25.30 \ dB$, $\displaystyle26.49 \ dB$, $\displaystyle26.45 \ dB$) : [Figure 16(d)](https://figshare.com/s/284ec76fcbc979219a81)
> * Biologically Plausible WSM Outputs (PSNR values of $\displaystyle 27.99 \ dB$, $\displaystyle 29.71 \ dB$, and $ \displaystyle 31.92 \ dB$ ) : [Figure 16(e)](https://figshare.com/s/b75481f373688db92005)
> * Batch LD-InfoMax Outputs (PSNR values are $\displaystyle33.60 \ dB$, $\displaystyle31.99 \ dB$, and $\displaystyle33.62 \ dB$) : [Figure 16(f)](https://figshare.com/s/ec1aa6f6ad232d85597e)

---

### Official Review · Reviewer_LCHG · 2022-10-25

**Confidence:** 4
**Correctness:** 4
**Technical Novelty And Significance:** 3
**Empirical Novelty And Significance:** 3
**Recommendation:** 8

**Clarity, Quality, Novelty And Reproducibility:**

### Clarity

Overall, the paper is decently written. However, some of the results/derivations in my opinion should be moved to the appendix to leave more space.

This space can be used to, for instance, write down dynamics of all weights and activations for the whole network in one place. Right now the dynamics are presented one by one, which is somewhat hard to follow.

Minor comments:

\hat R in Eq.3 should be defined in the main text.

(personal preference) Adding Eq. when referencing equations is more readable.

Labelling your methods as (ours) in tables/plots should improve readability.

### Quality

The paper is technically solid. Moreover, the experiments are very extensive and cover several source separation setups, showing decent performance of the proposed algorithms.

### Novelty

The work is, to my knowledge, novel.

The discussion around Eqs. 14-15 reminds me of the FORCE rule [Sussillo and Abbott, 2009], which ended up with the same issue of approximating recursive least squares (RLS) to make it more plausible. The RLS derivation is similar to what you have in Appendix B. Probably it’s worth mentioning as a background citation? I think they ended up with a similar more plausible approximation to RLS, so some approximations in that or the follow-up work could improve your algorithm too, although it might be out of the scope of this work. See, for instance,

Emergence of complex computational structures from chaotic neural networks through reward-modulated Hebbian learning
GM Hoerzer, R Legenstein, W Maass, 2014

### Reproducibility
The code is provided but not checked. Additionally, the hyperparameters are listed in the appendix.


**Strength And Weaknesses:**

### Strengths

Theoretically grounded algorithm for blind source separation.

Derivation of network dynamics that can implement this algorithm online.

Good performance despite the online setting and correlation matrix approximations.

### Weaknesses

Writing is lacking in some parts (see below), but contribution-wise I don't see any issues.

**Summary Of The Paper:**

The paper proposes an online algorithm for blind source separation that is represented by a multi-layer network with biologically plausible weight/activity dynamics.

**Summary Of The Review:**

Good paper with a solid technical and experimental contributions. I recommend acceptance.

---

> ### Author Response · Authors · 2022-11-12
> **Response to Reviewer LCHG**
>
> We thank you for your useful feedback and positive reviews. We appreciate your rating our paper as solid. In the revised article and the comments below, we address your comments and concerns.
>
> > *Clarity*: Overall, the paper is decently written. However, some of the results/derivations in my opinion should be moved to the appendix to leave more space.
> This space can be used to, for instance, write down dynamics of all weights and activations for the whole network in one place. Right now the dynamics are presented one by one, which is somewhat hard to follow.
>
> Thanks a lot for this constructive suggestion.  We moved the table containing different source domain dynamics/activations to the appendix.  Based on your suggestion, we inserted an Algorithm box summarizing output and learning dynamics for the Sparse CorInfoMax network.
>
> >*Minor Comments*: $\hat{\mathbf{R}}$ in Eq.3 should be defined in the main text.
>
> Following your suggestion, we included the definitions of $\hat{\mathbf{R}}_\mathbf{y}$ and $\hat{\mathbf{R}}_\mathbf{xy}$ after the optimization setting in (3).
>
> >*Minor Comments*: (personal preference) Adding Eq. when referencing equations is more readable.
>
> Thanks for this suggestion. We tried placing Eq. before equations, but we could  not fit to the page limits. Therefore,  we decided to keep our equation referencing in the initial draft.
>
> >*Minor Comments*: Labelling your methods as (ours) in tables/plots should improve readability.
>
> Thanks for this suggestion. In the revised article,  we include (ours) labels in the captions of figures and tables.
>
> >*Quality*: The paper is technically solid. Moreover, the experiments are very extensive and cover several source separation setups, showing decent performance of the proposed algorithms.
>
> We appreciate your positive feedback.
>
> >The work is, to my knowledge, novel.
>
> >The discussion around Eqs. 14-15 reminds me of the FORCE rule [Sussillo and Abbott, 2009], which ended up with the same issue of approximating recursive least squares (RLS) to make it more plausible. The RLS derivation is similar to what you have in Appendix B. Probably it’s worth mentioning as a background citation? I think they ended up with a similar more plausible approximation to RLS, so some approximations in that or the follow-up work could improve your algorithm too, although it might be out of the scope of this work. See, for instance,
>
> >Emergence of complex computational structures from chaotic neural networks through reward-modulated Hebbian learning GM Hoerzer, R Legenstein, W Maass, 2014
>
> Thank you for your comment. Both inverse covariance updates in (13)-(14) ( (14)-(15) in the initial submission) in our algorithm and RLS  are based on the use of Matrix Inversion Lemma (MIL) or Woodburry Identity to convert the rank-one update for the covariance/correlation to the rank-one update for its inverse. However, we do not really use it to minimize the least squares cost, but to maximize $\log-\det$ entropy objective. Following your suggestion, we included the reference Kailath et al. (2000), right before the application of MIL (before equation (A.10)) in Appendix B, to mention the connection to the RLS derivation. We appreciate your reference recommendations for potential extensions. The FORCE rule indeed uses RLS and, therefore, the partial resemblance in expressions.

---

> > ### Comment · Reviewer_LCHG · 2022-11-15
> > **Good response, keeping the score of 8**
> >
> > Thank you for the response! I'm happy with the paper/current comments, so I'm leaving the same score (8).
> >
> > > We tried placing Eq. before equations, but we could not fit to the page limits.
> >
> > Understandable!
> >
> > > The FORCE rule indeed uses RLS and, therefore, the partial resemblance in expressions.
> >
> > Yes, the resemblance is indeed due to the RLS derivation rather than the similarity to FORCE.

---

> > > ### Author Response · Authors · 2022-11-18
> > > **Thank you**
> > >
> > > We would like to thank you again for your efforts and for your feedback about our response.

---

### Official Review · Reviewer_hwJs · 2022-10-26

**Confidence:** 2
**Correctness:** 4
**Technical Novelty And Significance:** 2
**Empirical Novelty And Significance:** 2
**Recommendation:** 6

**Clarity, Quality, Novelty And Reproducibility:**

It is not clear how important the biological plausibility is and that does not seem be further explore by the paper.

Nits:

Both the title and the abstract are very hard to understand and one is left wondering what the paper is about. Latent causes are not defined, domains are not defined, what are presumed sets?

Second paragraph: The fact that biological receptive fields look like ICA filters should not be confused with the much stronger claim that the brain is doing BSS.

All figures need more explanation in the caption.

1.1.1 second to last sentence: ‘while’ ? ‘they’; autocorrelation

2.1.iii superfluous dot?

2.1 after list the “” quotation marks are wrongly formatted

Fig3 define the axis labels in the caption

**Strength And Weaknesses:**

The paper is well written, with a thorough mathematical derivation. It also cites many relevant existing works.

The paper is extremely dense, possibly incremental on previous work, somewhat niche.



**Summary Of The Paper:**

I thank the authors for providing a thoughtful reply addressing most of my concerns. Together with the clarifications I am now convinced that this work is a good contribution. I have adjusted my score accordingly.

The paper proposes a biologically plausible source separation method with local updates and batched online learning.

I can see that the paper is based on a recent publication by Erdogan (2022). Generally, this work cites a large number of previous works by that author. I cannot judge how novel the contribution in this paper is compared to those previous works. The framework and mathematical formulation are extremely complex and I was not able, within reasonable time investment, to digest and asses the full breadth of it. The applications seem somewhat niche. And the only real data application does, as far as I can tell, not include the interesting setting of correlated sources. By the way, the same setting (in more interesting nonlinear scenarios) is also investigated in this (https://proceedings.mlr.press/v139/trauble21a.html) work.

**Summary Of The Review:**

The work is hard to digest and it is not clear how incremental it is compared to previous work.

---

> ### Author Response · Authors · 2022-11-12
> **Response to Reviewer hwJs - Part 4 of 4**
>
> >*Clarity, Nits*: Second paragraph: The fact that biological receptive fields look like ICA filters should not be confused with the much stronger claim that the brain is doing BSS.
>
> Thank you for this point. We agree. Therefore, we had used soft claims in that paragraph, i.e. “potential ubiquity of BSS in the brain”.
>
> To expand on our view, the BSS problem is essentially the unsupervised linear inverse problem, where we assume a generative model that poses measurements as a linear transformation of some latent factors. Therefore, we can potentially look at some of the existing methods to model the receptive fields of visual system (V1) neurons from this perspective, in which we model natural image patches as linear transformations of independent or sparse latent factors.  Through the solution of this unsupervised linear inverse problem,  we obtain both the latent factors and the linear transformation matrix that can be used to extract potential models for the receptive fields. In fact,  in Appendix D.3.1, we use this perspective and apply the sparse CorInfoMax BSS solver to obtain receptive field models shown in Figure 14.
>
> >*Clarity, Nits*: All figures need more explanation in the caption.
>
> We appreciate your feedback on the captions of the figures. Based on your suggestions and Reviewer C2ij's suggestions, we extended all figure captions. In particular, in the revised article, we included more thorough explanations of the CorInfoMax networks in the figure captions, describing all of their components.
>
> >*Clarity, Nits*: 1.1.1 second to last sentence: ‘while’ ? ‘they’; autocorrelation
>
> Thank you for pointing out this typo. In the revised article, we  corrected and modified this sentence as:
>
> “For example, the lateral connections of the outputs in Bozkurt et al. (2022) are based
> on the output autocorrelation matrix, while the lateral connections for our proposed framework are
> based on the inverse of the output autocorrelation matrix.”
>
> >*Clarity, Nits*:
> >* 2.1.iii superfluous dot?
> >* 2.1 after list the “” quotation marks are wrongly formatted
> >* Fig3 define the axis labels in the caption
>
> Thanks for pointing out these. In the revised article, we implemented these corrections/suggestions.
>
> >*Summary of the Review*:The work is hard to digest and it is not clear how incremental it is compared to previous work.
>
> Thank you for this summary.
>
> Regarding clarity: We believe that by correcting/implementing the points suggested by the reviewers and with additional corrections and clarifications, our revised article is easier to understand and follow.
>
>
> Regarding Novelty: In answering your first comment, we hope that we adequately address your concerns about the novelty of our article in relation to the existing literature. Our article offers a fresh framework for generating biologically plausible neural networks to solve the BSS problem for potentially correlated sources based on the principle of correlative information maximization. To the best of our knowledge, there is no biologically plausible framework that makes direct use of the correlative information maximization principle to solve the problem of blind separation of correlated sources. Therefore, our comprehensive framework is a totally novel approach that clearly distinguishes itself from existing biologically plausible and batch approaches to solving the correlated BSS problem. We will gladly provide further clarification if we are given more specific grounds for any remaining reservations about novelty.

---

> ### Author Response · Authors · 2022-11-12
> **Response to Reviewer hwJs - Part 3 of 4**
>
> >*Clarity*:It is not clear how important the biological plausibility is and that does not seem be further explore by the paper.
>
> Thank you for this comment. We had to keep this discussion brief in the article due to page-length constraints. We have three major arguments for the significance of biological plausibility and the local learning rule:
> * *biological linear inverse solvers*: The blind source separation problem is ubiquitous in nature, and, in essence, it is the “unsupervised linear inverse problem”. Therefore, it is natural to expect that brains are potentially employing circuits to implement this fundamental system block. Our article offers a general framework with a local learning rule constraint, which can be considered as a potential explanation of how brains might implement BSS circuits that are robust against potential source correlations.
> * *biological representation learners*: From an alternative angle, the neural circuits of the proposed framework can also work as feature generators, where the selection of the source domain determines the feature characteristics, which can be selected as a mixture of attributes such as negativity, sparsity, antisparsity, etc. These circuits with local learning rules aim to preserve the information content of the inputs in the outputs while restricting the outputs to the desired representation form. We can view the sparse dictionary learning experiment in Appendix D.3.1 as an example of this perspective. As a future extension of this perspective, we can consider $\mathbf{A}_\mathcal{P}$ of the canonical form in Figure 1.(a), which determines the constraint set for the output feature vectors and, therefore, their attributes, to be adaptable (by a chosen normative criterion). In such a setting, networks correspond to representation learning circuits with maximum information transfer from their inputs to their outputs, where the domains of outputs are adaptively adjusted.
> * *neuromorphic system implementations*: The key property of the proposed framework is its foundation on the local learning rule, which is considered a fundamental property that brain circuits are believed to possess. The local learning rule is also a hard constraint for low-power adaptable neuromorphic systems. Therefore, our framework and its potential supervised extensions are potentially useful for developing low-power machine learning architectures.
>
> In the revised article, we include a brief summary of these points in the conclusion section ( Section 5).
>
> >*Clarity, Nits*: Both the title and the abstract are very hard to understand and one is left wondering what the paper is about. Latent causes are not defined, domains are not defined, what are presumed sets?
>
> Thank you for these suggestions. Regarding the title, we like the current version because it is complete in the sense that it includes the normative approach (information maximization), the implementation feature (biological plausibility) and the performance feature (correlated source separation capability). We are open to suggestions to improve our title.
>
> Regarding the abstract: we agree that the term “presumed sets” is not clear, and in the revised article, we replaced the corresponding sentence as
>
> * “To derive this network, we choose the maximum correlative information transfer from inputs to outputs as the separation objective under the constraint that the output vectors are restricted to the set where the source vectors are assumed to be located.”
>
> In other words, we replaced “their presumed sets” by “the set where source vectors are assumed to be located.”  In addition, we replaced the “infinitely many source domains” in the following sentence with the “infinitely many set choices for the source domain” for further clarification on “domain”.
>
> The word *latent* is a common term in the unsupervised learning literature used in place of “hidden, unaccesible”. We do not refer to any specific sources that generate stimulations. However, we can replace it with a better alternative if needed.

---

> ### Author Response · Authors · 2022-11-12
> **Response to Reviewer hwJs - Part 2 of 4**
>
> >*Summary of the Paper*: The applications seem somewhat niche. And the only real data application does, as far as I can tell, not include the interesting setting of correlated sources. By the way, the same setting (in more interesting nonlinear scenarios) is also investigated in this (https://proceedings.mlr.press/v139/trauble21a.html) work.
>
> We apologize for not being clear with the setting of our numerical experiments. We included synthetic examples as proof-of-concept demonstrations for robustness against varying levels of source correlation, input noise power, and mixing system.
>
> The **video separation** application provided in the Numerical Experiments (Section 4) section in Appendix D.3.3 is actually an example of **naturally correlated sources**. However, *we did not report the correlation levels* between the frames in the original article. In order to make this point clear, in the revised article, we report the correlation levels between the frames of different video sources. According to the revised article, the average (across frames) and maximum correlation levels for the video sources (across frames) are $\rho_{12}^{\text{average}}=-0.1597$, $\rho_{13}^{\text{average}}=-0.1549$, $\rho_{23}^{\text{average}}=0.3811$ and $\rho_{12}^{\text{maximum}}=0.3139$, $\rho_{13}^{\text{maximum}}=0.2587$, $\rho_{23}^{\text{maximum}}=0.5173$, respectively. Therefore, the video example serves as a nice illustration to demonstrate the successful separation capability for naturally correlated sources.
>
> To address the same point, we added a new example in Appendix D.3.4, in which the sources are photographs, which serve as excellent examples of naturally correlated sources. In fact, the pairwise Pearson correlation coefficients for the three images used in this experiment are $\rho_{12}=0.076, \rho_{13}=0.262, \rho_{23}=0.240$, respectively. We compared the proposed CorInfoMax networks with selected existing **biologically plausible neural network solutions** (Nonnegative Similarity Matching (NSM) and Weighted Similarity Matching (WSM)) with correlated source separation capability, and **batch algorithms for independent and dependent component analysis** (ICA-InfoMax and LD-InfoMax).  As illustrated by this example, the proposed CorInfoMax approach achieves a performance level superior to other biologically plausible network approaches and batch ICA-InfoMax algorithm and close to the batch LD-InfoMax algorithm, which illustrates its potential for naturally correlated sources.
>
> Following are source, mixture, and algorithm output images that are available in Appendix D.3.4 of the revised article.
>
> * Original Sources : [Figure 16(a)](https://figshare.com/s/cb386db6287a56da53af)
> * Mixtures : [Figure 16(b)](https://figshare.com/s/f4c818c0034a1fcd5cc3)
> * Proposed CorInfoMax Outputs (PSNR values are $\displaystyle 32.45 \ dB$, $\displaystyle 29.72 \ dB$, and $\displaystyle 32.37 \ dB$) : [Figure 16(g)](https://figshare.com/s/1a54b10d6fc6cea3f7d7)
> * Batch ICA-InfoMax Outputs (PSNR values are $\displaystyle18.56 \ dB$, $\displaystyle20.52 \ dB$, and $\displaystyle21.06 \ dB$) : [Figure 16(c)](https://figshare.com/s/abc2981b8ce5eee8b94a)
> * Biologically Plausible NSM Outputs (PSNR values are $\displaystyle25.30 \ dB$, $\displaystyle26.49 \ dB$, $\displaystyle26.45 \ dB$) : [Figure 16(d)](https://figshare.com/s/284ec76fcbc979219a81)
> * Biologically Plausible WSM Outputs (PSNR values of $\displaystyle 27.99 \ dB$, $\displaystyle 29.71 \ dB$, and $ \displaystyle 31.92 \ dB$ ) : [Figure 16(e)](https://figshare.com/s/b75481f373688db92005)
> * Batch LD-InfoMax Outputs (PSNR values are $\displaystyle33.60 \ dB$, $\displaystyle31.99 \ dB$, and $\displaystyle33.62 \ dB$): [Figure 16(f)](https://figshare.com/s/ec1aa6f6ad232d85597e)
>
> Finally, thank you for pointing to Trauble et al. (2021) as a relevant reference. Indeed, they address the key problem of investigating the impact of correlation in latent factors, and we cited this reference in our revised article. In our article, we focus on constructing a biologically plausible solution to the unsupervised linear inverse problem in which the desired latent variables are potentially correlated. Therefore, our article is related to Trauble et al. (2021) only from a broader perspective but different in our specific contributions and objectives.

---

> ### Author Response · Authors · 2022-11-12
> **Response to Reviewer hwJs - Part 1 of 4**
>
> We thank you for your time and constructive comments. We address your comments and questions in the revised article and in our response below.
>
> > *Summary of the Paper*: I can see that the paper is based on a recent publication by Erdogan (2022). Generally, this work cites a large number of previous works by that author. I cannot judge how novel the contribution in this paper is compared to those previous works. The framework and mathematical formulation are extremely complex and I was not able, within reasonable time investment, to digest and asses the full breadth of it.
>
> Thank you for this comment and incentivizing us to clarify our contributions. In Section 1.1 of the article, we provide the relation of the proposed *Biologically Plausible CorInfoMax Blind Source Separation* framework to the existing literature, constrained by the page limitations of the article.  We can provide the following high-level summary to clarify the novelty of our article:
> * The Determinant Maximization criterion has been proposed for structured matrix factorization frameworks, including Nonnegative Matrix Factorization and Polytopic Matrix Factorization.
> * The reference, Erdogan (2022), provides a second-order statistics-based information maximization perspective for these determinant-maximization/minimization-based matrix factorization methods and for the connected BSS problems. However, the algorithm provided in this reference is a ***batch*** algorithm, i.e., no neural network implementations.
> * The framework proposed in our current article offers an ***online formulation*** for the correlative information maximization criterion of Erdogan (2022), which enables implementations of biologically plausible neural networks.
> * The reference Bozkurt et al. (2022) also offers a biologically plausible BSS framework for determinant maximization. However, the main differences are the following: this reference is not directly based on the correlative information maximization objective. Furthermore, Bozkurt et al. (2022) uses a ***similarity matching criterion*** for generating biologically plausible networks, which we do not use in our article. The resulting neural network architectures and learning rules are completely different. The networks in Bozkurt et al. (2022) have neurons with learnable gains,  the lateral weights corresponding to the output correlation matrix, which results from the similarity matching criterion (in our case, the lateral weights correspond to the inverse of the correlation matrix, which results from logdeterminant optimization). The feedforward weights of the networks in Bozkurt et al. (2022) correspond to a cross-correlation matrix between the inputs and outputs of a layer,  whereas, for the proposed framework, the feedforward connections correspond to the linear predictor of the output from the input. Finally, the feedback weights of networks in Bozkurt et al. (2020) are the transpose of the feedforward weights, whereas in our framework the feedback weights are diagonal and not the transpose of feedforward weights.
>
> To address your concerns and better clarify the novelty of our article, we revised the discussion in Section 1.1 (Related Work and Contributions) of the article.

---

> ### Author Response · Authors · 2022-11-18
> **Thank you**
>
> We would like to thank you again for your efforts and for your feedback about our response.

---

### Decision · Program_Chairs · 2023-01-20

**Decision:**

Accept: poster

**Justification For Why Not Higher Score:**

Though the reviewers were generally positive, and hence I think it should be accepted, the paper is very dense and is arguably not incredibly novel. Thus, I don't think it is appropriate for a spotlight.

**Justification For Why Not Lower Score:**

The work is technically sound and does make a clear contribution, so rejection would be inappropriate.

**Metareview: Summary, Strengths And Weaknesses:**

This paper propose an information-theoretic framework for creating neural networks that separate latent sources in a dataset, including for correlated sources. The authors use the maximum correlative information transfer from inputs to outputs as the separation objective, with additional constraints. This leads to local learning rules for the networks that can learn good separation of sources.

The reviewers agreed that the paper is technically sound and provides a novel contribution.

There were some consistent concerns regarding the density of the paper and lack of clarity in parts, as well as a few more reviewer specific concerns about whether the results are incremental or whether the connections to biology are all that strong. However, after rebuttal, the final scores moved up a bit to provide a final average score of 6.33. Based on these scores, as well as the general consensus that the work is technically sound and provides at least some novel contributions, a decision for accept was reached.

**Note From Pc:**

if the above contains the word "oral" or "spotlight" please see: "oral" presentation means -> notable-top-5% and "spotlight" means -> notable-top-25%. As stated in our emails, we are disassociating presentation type from AC recommendations

**Summary Of Ac-Reviewer Meeting:**

N/A